# Epigenome-wide meta-analysis of DNA methylation differences in prefrontal cortex implicates the immune processes in Alzheimer's disease

Lanyu Zhang[1,5], Tiago C. Silva [1,5], Juan I. Young[2,3], Lissette Gomez[3], Michael A. Schmidt [2,3], Kara L. Hamilton-Nelson[3], Brian W. Kunkle[2,3], Xi Chen[1,4], Eden R. Martin[2,3] & Lily Wang [1,2,3,4 ✉]

DNA methylation differences in Alzheimer's disease (AD) have been reported. Here, we conducted a meta-analysis of more than 1000 prefrontal cortex brain samples to prioritize the most consistent methylation differences in multiple cohorts. Using a uniform analysis pipeline, we identified 3751 CpGs and 119 differentially methylated regions (DMRs) significantly associated with Braak stage. Our analysis identified differentially methylated genes such as *MAMSTR*, *AGAP2*, and *AZU1*. The most significant DMR identified is located on the *MAMSTR* gene, which encodes a cofactor that stimulates MEF2C. Notably, MEF2C cooperates with another transcription factor, PU.1, a central hub in the AD gene network. Our enrichment analysis highlighted the potential roles of the immune system and polycomb repressive complex 2 in pathological AD. These results may help facilitate future mechanistic and biomarker discovery studies in AD.

[1] Division of Biostatistics, Department of Public Health Sciences, Miller School of Medicine, University of Miami, Miami, FL 33136, USA. [2] Dr. John T Macdonald Foundation Department of Human Genetics, Miller School of Medicine, University of Miami, Miami, FL 33136, USA. [3] John P. Hussman Institute for Human Genomics, Miller School of Medicine, University of Miami, Miami, FL 33136, USA. [4] Sylvester Comprehensive Cancer Center, Miller School of Medicine, University of Miami, Miami, FL 33136, USA. [5]These authors contributed equally: Lanyu Zhang, Tiago C. Silva. ✉email: lily.wang@miami.edu

Late-onset Alzheimer's disease (LOAD) is the most common cause of dementia, affecting about 10% of people 65 years and older in the US[1]. The causes of Alzheimer's disease (AD) are complex, with the disease likely resulting from a complicated interplay of genetic factors and environmental factors. While a number of AD-associated genetic variants have been identified[2,3], they do not completely explain an individual's risk for developing AD. Epigenetic studies investigate the mechanisms that modify the expression levels of genes without changes to the underlying DNA sequence. In particular, there is growing evidence for the prominent role of DNA methylation in AD. Several reviews[4–7] have provided comprehensive details on recent findings of DNA methylation and other epigenetic alterations in AD.

For neurological disorders such as AD, the use of disease-relevant tissue is often preferred for epigenetic studies. However, obtaining sufficient sample sizes for brain studies is challenging because of the difficulty in procuring postmortem human brain tissue. This makes it difficult to detect the DNA methylation differences observed in the brains of AD subjects, because these differences are often of small magnitude. For example, in the Lunnon et al. epigenome-wide association study (EWAS)[8], which examined postmortem brain tissues in pathological AD subjects and controls, the absolute difference in corrected DNA methylation between individuals with the lowest (score 0) and highest (score VI) Braak score ranged from 1 to 5% change even for the most significant CpGs in the prefrontal cortex region, a region that shows considerable vulnerability to AD. Furthermore, because of methodological differences used for analyzing different methylation datasets, inconsistencies are often seen across multiple studies[9,10].

To address these challenges, we conducted a meta-analysis of 1030 prefrontal cortex samples. All the samples included in this meta-analysis were measured on the same Infinium Human-Methylation450 BeadChip platform, targeted the same prefrontal cortex region, and included the same pathological variable, Braak stage, which is a standardized measure of neurofibrillary tangle burden determined at autopsy. Moreover, we reanalyzed each dataset using a uniform analytical pipeline. In addition to meta-analyzing individual CpGs, we also performed a meta-analysis of genomic regions, as methylation levels are often strongly correlated in closely located CpGs[11]. Methods for identifying differentially methylated regions (DMRs) can be classified into supervised methods, which look for regions in the genome with consecutive small P-values, or unsupervised methods which group CpGs probes into clusters first and then test the clusters against phenotype[10]. We performed a meta-analysis of DMRs using two complementary analysis tools, a supervised method comb-p[12] and an unsupervised method coMethDMR[13]. Relevant to this meta-analysis, the coMethDMR-based meta-analysis strategy allowed us to assess between cohort heterogeneities in genomic regions. We identified 119 DMRs and 3751 significant CpGs that are consistently associated with AD Braak stage in multiple cohorts. In addition to corroborating previous findings, our analysis also nominated a number of differentially methylated genes. Enrichment analysis of differentially methylated genes highlighted multiple immune processes epigenetically associated with pathological AD as well as polycomb repressed regions.

## Results

**Study cohort characteristics.** Our meta-analysis included 1030 prefrontal cortex brain samples from four independent cohorts (Table 1), previously described in the ROSMAP[14], Mt. Sinai[15], London[8], and Gasparoni[16] methylation studies. To assess the diagnostic utility of DNA methylation as clinical biomarkers, we also compared methylation differences in brain samples with premortem whole blood samples from a subset of subjects in the London cohort. Among the four cohorts, the mean age at death ranged from 73.6 years to 86.3 years, and the percentage of females ranged from 51.8 to 63.5%.

**Meta-analysis identified methylation differences significantly associated with AD Braak stage at individual CpGs and co-methylated genomic regions.** Adjusting for estimated cell-type proportions (i.e., the proportion of neurons), age at death, sex, and batch effects, our meta-analysis of single CpGs in the four cohorts identified 3979 statistically significant individual CpGs at 5% false discovery rate (FDR). After eliminating CpGs associated with smoking[17] or overlapping with cross-reactive probes[18], we obtained 3751 significant CpGs (Supplementary Data 1), of which 47.8% is located in noncoding regions. This proportion is lower than the proportion of noncoding probes (53.2%) on the array (P-value = $2.18 \times 10^{-11}$). Among the 3751 CpGs, 339 also reached genome-wide significance level (see details "Genomic inflation and sensitivity analysis" section).

The DMRs were identified by both coMethDMR[13] and comb-p[12] software. In the coMethDMR approach, we tested 40,010 predefined genomic regions to identify co-methylated and differentially methylated regions associated with Braak stage, adjusting for estimated neuron proportions, age at death, sex, and batch effects for each cohort separately. Next, we combined the cohort-specific P-values for these genomic regions using inverse-variance weighted regression models for meta-analysis. Alternatively, in the comb-p approach, we used meta-analysis P-values of individual CpGs as input, and comb-p was then used to scan the genome for regions enriched with a series of adjacent low P-values (Fig. 1).

The coMethDMR and comb-p based meta-analysis approaches identified 478 and 187 significant DMRs associated with AD Braak stage, respectively, with 143 being identified by both methods. After eliminating those DMRs containing cross-reactive probes[18] or smoking-associated probes[17], we obtained 119 co-methylated DMRs at 5% FDR (Supplementary Data 2). The average number of CpGs per DMR is 5.16 ± 2.80 CpGs. Notably,

**Table 1 Sample characteristics of the brain and blood cohorts included in the meta-analysis.**

| Dataset | Tissue | Sample size | Women N (%) | Cases N (%) | Age at death mean (SD) | Accession |
|---|---|---|---|---|---|---|
| *Brain samples cohort* | | | | | | |
| (1) ROSMAP cohort | PFC | 726 | 461 (63.5%) | 581 (80.0%) | 86.3 (4.8) | Synapse: syn3157275 |
| (2) Mt. Sinai cohort | PFC | 141 | 88 (62.4%) | 85 (60.3%) | 85.8 (7.8) | GEO: GSE80970 |
| (3) London cohort | PFC | 107 | 64 (59.8%) | 80 (74.8%) | 84.6 (9.0) | GEO: GSE59685 |
| (4) Gasparoni cohort | PFC | 56 | 29 (51.8%) | 36 (64.3%) | 73.6 (14.7) | GEO: GSE66351 |
| (5) London cohort | whole blood | 69 | 44 (63.8%) | 59 (85.5%) | 83.6 (6.2)* | GEO: GSE59685 |

Shown are numbers (and percentages) of samples after quality control.
*PFC* prefrontal cortex.
*Age at blood draw.

118 out of the 119 DMRs included FDR significant individual CpGs. On the other hand, only 421 out of the 3751 FDR significant individual CpGs overlapped with the FDR significant DMRs. Therefore, methylation differences at individual CpGs and DMRs did not completely overlap, so analyzing both individual CpGs and DMRs provided a more complete picture of the Braak-associated methylome in the prefrontal cortex. Our final set of DNA methylation differences included these 3751 individual CpGs along with the 119 DMRs (Fig. 2). The top 20 most significant CpGs and DMRs are shown in Tables 2 and 3, respectively. As previous studies have noted[8,14,19,20], in both individual CpG and DMR analyses, we observed that the majority of the significant methylation differences were hyper-methylated in AD, for which methylation levels were increased as AD stage increased. More specifically, 58.6% of significant CpGs and 73.9% of significant DMRs were hyper-methylated in AD (Supplementary Data 1 and 2).

**Enrichment analysis of significant DNA methylation differences in pathological AD highlights immune-related processes and polycomb repressive complex 2 (PRC2).** The probes on the Illumina 450k array are annotated according to their locations with respect to genes (TSS1500, TSS200, 5′UTR, 1st Exon, gene body, 3′UTR, and intergenic) or to CpG islands (island, shelf, and open sea). We tested enrichment of the significant methylation differences associated with pathological AD in these different types of genomic features by analyzing individual CpGs and DMRs separately using Fisher's exact test. Interestingly, hypermethylation in AD at individual CpGs and DMRs was enriched in different features of genes across the genome (Fig. 3a, b and Supplementary Data 3). Significant hypermethylated individual CpGs were over-represented in CpG island and shore but under-represented in open sea and shelf (Fig. 3a). In contrast, the hypermethylated DMRs were under-represented in CpG island but enriched in open sea and shelf (Fig. 3b). In terms of genic features, the hypermethylated individual CpGs were enriched in 1st Exon, 5′UTR, and gene body but under-represented in intergenic regions and TSS200 (Fig. 3a). On the other hand, the hyper-methylated DMRs were only slightly under-represented in intergenic regions (Fig. 3b). In contrast, there was more agreement between hypomethylated changes at individual CpGs and DMRs. Both significant hypomethylated individual CpGs and DMRs were enriched in open sea, but depleted in CpG islands.

In addition, we also compared our results with epigenomic annotations including chromatin states and transcription factor binding sites. Using combinations of histone modification marks, computational algorithms such as ChromHMM[21] segment and annotate the genome with different chromatin states (repressed, poised and active promoters, strong and weak enhancers, putative insulators, transcribed regions, and large-scale repressed and inactive domains), which were shown to vary across sex, tissue type, and developmental age[22]. For our analysis, we used the 15-state ChromHMM annotation of the Roadmap Epigenomics project[23]. Our enrichment analysis with respect to chromatin states showed that significant hypermethylated DMRs and CpGs were both enriched in flanking active promoter regions (TssAFlnk), enhancers (Enh), Transcr. at gene 5′ and 3′ (TxFlnk), and polycomb repressed regions (ReprPC), but under-represented in promoter regions (TssA), strongly transcribed regions (Tx), and repressed regions (Quies, ReprPCWk). In contrast, hypomethylated DMRs and CpGs were enriched in repressed regions (Quies, ReprPCWk) and weakly transcribed regions (TxWk), but under-represented in promoter regions (TssA, TssBiv) (Fig. 3c, d and Supplementary Data 4). Notably, among the 151 significant CpGs located in the group of Hox genes on chromosome 7, the majority (135 out of 151) were located in polycomb repressed regions, and the rest in bivalent enhancer and bivalent/poised TSS regions.

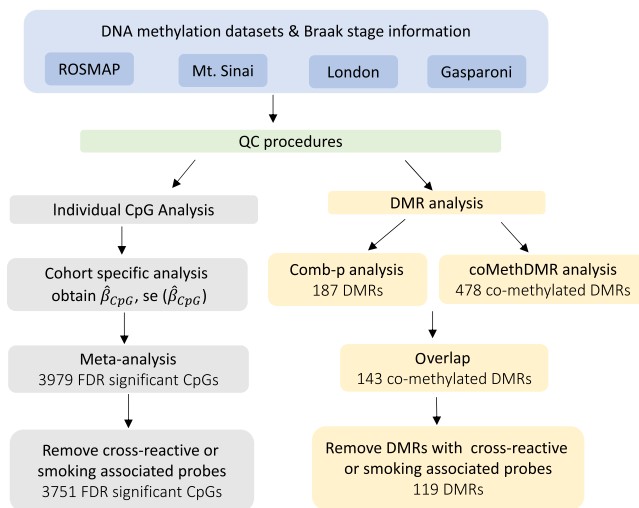

**Fig. 1 Workflow of meta-analysis for individual CpGs and DMRs.**

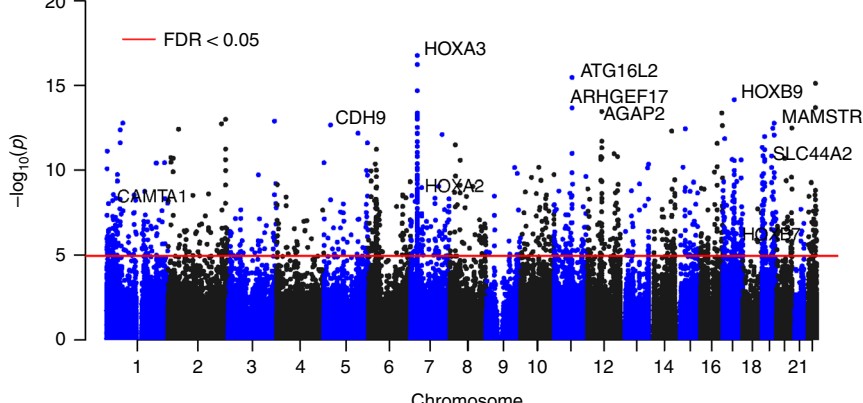

**Fig. 2 Manhattan plot of significant methylation differences in individual CpGs and DMRs identified in meta-analysis using inverse-variance weighted regression models.** The X-axis indicates chromosomes 1–22 and the Y-axis indicates $-\log_{10}$ (P-value), with the horizontal red line indicating a 5% FDR (false discovery rate) adjusting for multiple comparisons.

**Table 2 Top 20 most significant differentially methylated CpGs associated with Braak stage in meta-analysis.**

| CpG | Chr | Position | GREAT_annotation | Illumina annotation | Estimate | P-value | FDR | Estimate_direction |
|---|---|---|---|---|---|---|---|---|
| cg22962123 | 7 | 27,153,605 | HOXA2 (−11176); HOXA3 (+5608) | HOXA3 | 0.080 | 1.71E−17 | 7.73E−12 | ++++ |
| cg01301319 | 7 | 27,153,580 | HOXA2 (−11151); HOXA3 (+5633) | HOXA3 | 0.056 | 5.86E−17 | 1.32E−11 | ++++ |
| cg21806242 | 11 | 72,532,891 | ATG16L2 (+7539); FCHSD2 (+320414) | ATG16L2 | 0.082 | 3.42E−16 | 5.14E−11 | ++++ |
| cg06635946 | 22 | 46,470,016 | WNT7B (−97008); PPARA (−76482) | NA | 0.083 | 7.55E−16 | 8.50E−11 | ++++ |
| cg07061298 | 7 | 27,153,847 | HOXA2 (−11418); HOXA3 (+5366) | HOXA3 | 0.092 | 2.03E−15 | 1.80E−10 | ++++ |
| cg04917446 | 17 | 46,699,073 | HOXB8 (−6773); HOXB9 (+4765) | HOXB9 | 0.058 | 6.93E−15 | 5.20E−10 | ++++ |
| cg20864214 | 11 | 73,054,121 | RELT (−33187); ARHGEF17 (+34788) | ARHGEF17 | 0.073 | 2.12E−14 | 1.19E−09 | ++++ |
| cg03672272 | 22 | 46,470,191 | WNT7B (−97183); PPARA (−76307) | NA | 0.059 | 2.04E−14 | 1.19E−09 | ++++ |
| cg09596958 | 12 | 58,132,105 | AGAP2 (−77) | AGAP2 | 0.093 | 3.45E−14 | 1.73E−09 | ++++ |
| cg17775899 | 7 | 27,179,426 | HOXA4 (−9009); HOXA5 (+3860) | NA | 0.050 | 4.23E−14 | 1.73E−09 | ++++ |
| cg04874795 | 16 | 86,477,638 | FOXF1 (−66494); IRF8 (+545230) | NA | −0.071 | 4.23E−14 | 1.73E−09 | −−−− |
| cg18751141 | 7 | 27,138,173 | HOXA1 (−2581) | NA | 0.066 | 4.69E−14 | 1.76E−09 | ++++ |
| cg14216068 | 7 | 27,146,445 | HOXA2 (−4016) | HOXA3 | 0.072 | 6.40E−14 | 2.22E−09 | ++++ |
| cg01964852 | 7 | 27,146,262 | HOXA2 (−3833) | HOXA3 | 0.065 | 6.98E−14 | 2.25E−09 | ++++ |
| cg00921266 | 7 | 27,153,663 | HOXA2 (−11234); HOXA3 (+5550) | HOXA3 | 0.098 | 7.72E−14 | 2.32E−09 | ++++ |
| cg09490371 | 2 | 233,253,024 | ALPPL2 (−18528); ALPP (+9781) | ECEL1P2 | 0.071 | 1.01E−13 | 2.67E−09 | ++++ |
| cg09144964 | 7 | 27,150,262 | HOXA2 (−7833); HOXA3 (+8951) | HOXA3 | 0.059 | 9.79E−14 | 2.67E−09 | ++++ |
| cg12307200 | 3 | 188,664,632 | TPRG1 (−225130); LPP (+733912) | NA | −0.057 | 1.28E−13 | 3.21E−09 | −−−− |
| cg13390284 | 1 | 65,531,864 | JAK1 (−99678); AK4 (−81648) | NA | 0.050 | 1.67E−13 | 3.81E−09 | ++++ |
| cg14103343 | 19 | 49,220,223 | MAMSTR (+2754); FUT2 (+20992) | MAMSTR | 0.063 | 1.69E−13 | 3.81E−09 | ++++ |

For each CpG, annotations include the location of the CpG based on hg19/GRCh37 genomic annotation (Chr, Position), nearby genes based on GREAT, and Illumina gene annotations. The inverse-variance weighted meta-analysis regression models results include estimated effect size (Estimate) where CpGs that are hyper-methylated in AD have positive values, P-value, and false discovery rate (FDR) for multiple comparison corrections. The last column (Estimate_Direction) indicates the direction of effects in Gasparoni, London, Mount Sinai, and ROSMAP cohorts where + indicates hyper-methylation in AD and − indicates hypomethylation in AD in an individual cohort. All P-values are two-sided.

Similarly, enrichment tests for regulatory elements using the LOLA software[24] also supported the potential functional relevance of these significant changes in DNA methylation. In particular, the significant CpGs were enriched in the binding sites of 26 transcription factors and chromatin proteins assayed by the ENCODE project[25] (Supplementary Data 5). Notably, the top hits included EZH2 and SUZ12, both are subunits of polycomb repressive complex 2 (PRC2), consistent with the observed enrichment of methylation differences in PRC2 repressed regions (Fig. 3c, d) and previous observations that DNA methylation often interact with PRC2 binding[26–28]. Another top enriched TF is PU.1, which is critical for the differentiation, proliferation, and survival of microglia, the resident macrophages of the brain[29]. Evidence from a recent GWAS study also suggested PU.1 as a master regulator for a number of genes associated with delayed onset of AD, including TREM2, CD33, and ABCA7[30].

As pathological AD-associated genes can harbor both significant individual CpGs and significant DMRs, we performed a pathway analysis by considering the significant CpGs and DMRs jointly. The test of KEGG pathways showed that hematopoietic cell lineage, phagosome, Cytokine–cytokine receptor interaction, and chemokine signaling pathways were significantly enriched with methylation differences in pathological AD at 5% FDR (Table 4). Similarly, gene ontology (GO) analysis showed strong enrichment in biological processes involving inflammatory response, immune cell differentiation, and cytokine production, recapitulating the prominence of immune processes in AD[31,32]. Other significant GO terms involved cellular processes previously shown to be important in AD including cell adhesion, phagocytosis, cell migration, and synapse pruning.

**Prioritizing significant DNA methylation differences with sample matching**. DNA methylation levels are known to be influenced by aging, which is also the strongest risk factor for AD. To prioritize significant methylation differences in pathological AD and minimize confounding effects due to aging, we also explored an alternative strategy by matching each pathological AD case with a control subject of the same sex and age at death in the same cohort. We obtained a total of 346 subjects (173 cases and 173 controls) for the matched sample set from the London ($n = 46$), Mount Sinai ($n = 56$), and ROSMAP ($n = 244$) cohorts. We did not include the Gasparoni cohort for this analysis because too few age and sex-matched samples ($n = 12$) were present in this dataset. The age and sex matched samples were then analyzed in the same way as described above, except for removing age at death and sex effects in the linear models. We identified a total of 151 CpGs and 32 DMRs that were significantly different after matching cases and control samples by sex and age at death (Supplementary Datas 6 and 7). Among them, 85% ($n = 129$) of CpGs and 50% ($n = 16$) of genomic regions overlapped with the significant CpGs and DMRs in our main analysis described above with the same direction of change. In particular, we found that methylation differences at a number of AD-related genes such as HOXA3, SLC44A2, AGAP2, CDH9, and MAMSTR were significant in both analyses.

**Correlation of AD-associated CpGs and DMRs methylation levels in blood and brain samples**. To evaluate the diagnostic potential of the identified methylation differences, we computed Spearman rank correlations between inter-individual variations of the DNA methylation levels in the brain and blood using the London cohort dataset[8], which included 69 pairs of matched brain and blood samples passing quality control (Table 1). The difference between age at pre-mortem blood draw and age at death ranged from 0 to 10 years, with an average of 3.81 ± 2.61

**Table 3 Top 20 most significant differentially methylated regions (DMRs) associated with Braak stage identified by both coMethDMR and comb-p in meta-analysis.**

| Region | GREAT annotation | Illumina annotation | No. probes | Estimate | coMethDMR P-value | FDR | comb-p Sidak P-value | Estimate_direction |
|---|---|---|---|---|---|---|---|---|
| chr19:49220102-49220485 | MAMSTR (+2684); FUT2 (+21062) | MAMSTR | 4 | 0.061 | 6.02E-16 | 2.41E-11 | 1.32E-28 | +++++ |
| chr7:27153580-27153944 | HOXA2 (-11332); HOXA3 (+5452) | HOXA3 | 6 | 0.084 | 1.16E-14 | 2.32E-10 | 1.59E-47 | +++++ |
| chr7:27146237-27146445 | HOXA2 (-391l) | HOXA3 | 4 | 0.066 | 2.23E-14 | 2.37E-10 | 2.06E-32 | +++++ |
| chr7:27154262-27155548 | HOXA2 (-12475); HOXA3 (+4309) | HOXA3 | 10 | 0.047 | 2.37E-14 | 2.37E-10 | 6.19E-27 | +++++ |
| chr7:27179161-27179432 | HOXA4 (-8879); HOXA5 (+3990) | NA | 3 | 0.045 | 1.35E-12 | 1.08E-08 | 2.60E-21 | +++++ |
| chr5:27038605-27038836 | CDH9 (-28) | CDH9 | 3 | 0.059 | 2.14E-12 | 1.22E-08 | 1.00E-11 | +++++ |
| chr7:27140797-27141139 | HOXA1 (-5375); HOXA2 (-1462) | HOXA2 | 5 | 0.048 | 3.44E-12 | 1.72E-08 | 2.79E-19 | +++++ |
| chr19:10736006-10736448 | SLC44A2 (+293) | SLC44A2 | 7 | 0.058 | 7.79E-12 | 3.11E-08 | 2.13E-38 | +++++ |
| chr7:692321-7692367 | VAMP3 (-138,985); CAMTA1 (+846,960) | CAMTA1 | 3 | -0.052 | 9.58E-12 | 3.19E-08 | 2.13E-12 | ----- |
| chr17:46685292-46685448 | HOXB6 (-3016) | HOXB7 | 3 | 0.046 | 1.49E-11 | 4.59E-08 | 2.58E-09 | +++++ |
| chr17:46698881-46699155 | HOXB8 (-6717); HOXB9 (+482l) | HOXB9 | 4 | 0.038 | 1.95E-11 | 5.56E-08 | 2.40E-20 | +++++ |
| chr6:1388668865-138867125 | ECT2L (-250068); HEBP2 (+141639) | NHSL1 | 7 | 0.025 | 4.12E-11 | 1.10E-07 | 1.82E-22 | +++++ |
| chr6:10556147-10556523 | GNT6 (-77658); GCNT2 (+27746) | GCNT2 | 3 | 0.070 | 7.84E-11 | 1.96E-07 | 4.69E-20 | +++++ |
| chr7:27143046-27143806 | HOXA2 (-996) | HOXA2 | 11 | 0.042 | 1.04E-10 | 2.45E-07 | 6.42E-21 | +++++ |
| chr9:98739496-98739782 | SMURF1 (+2084); TRRAP (+263526) | SMURF1 | 4 | 0.044 | 1.22E-10 | 2.71E-07 | 4.04E-11 | +++++ |
| chr19:827715-827843 | AZU1 (-47) | AZU1 | 3 | -0.033 | 2.10E-10 | 4.20E-07 | 5.87E-16 | ----- |
| chr19:6476756-6477198 | DENND1C (+4842); CRB3 (+12683) | DENND1C | 3 | 0.034 | 3.78E-10 | 6.86E-07 | 3.60E-13 | +++++ |
| chr11:72533295-72533664 | ATG16L2 (-8127); FCHSD2 (+319826) | ATG16L2 | 3 | 0.064 | 8.07E-10 | 1.29E-06 | 1.20E-20 | +++++ |
| chr20:57582581-57583709 | CTSZ (-843) | CTSZ | 16 | -0.037 | 8.03E-10 | 1.29E-06 | 2.49E-18 | ----- |
| chr5:172175604-172175855 | DUSP1 (+22468); NEURL1B (+107461) | NA | 4 | 0.027 | 1.07E-09 | 1.62E-06 | 2.22E-11 | +++++ |

For each DMR, annotations include location of the DMR based on hg19/GRCh37 genomic annotation (Region), nearby genes based on GREAT, and Illumina gene annotations. The annotations include the number of probes in the DMR (No. Probes), estimated effect size (Estimate) where DMRs that are hypermethylated in AD have positive values, P-value, and false discovery rate (FDR) for multiple comparison corrections. Meta-analysis results based on comb-p include multiple comparison corrected P-value based on Sidak method (Sidak P-value). The last column (Estimate_Direction) indicates the direction of DMR effects estimated by coMethDMR in Gasparoni, London, Mount Sinai, and ROSMAP cohorts where + indicates hypermethylation in AD and − indicates hypo-methylation in AD in an individual cohort. All P-values are two-sided.

years. We performed both an adjusted correlation analysis based on methylation residuals ($r_{resid}$), in which we adjusted estimated neuron proportions for brain samples (or estimated blood cell-type proportions), array, age at death (for brain samples) or at blood draw (for blood samples), and sex, and an unadjusted correlation analysis based on beta values ($r_{beta}$) (Online Methods). The correlation between methylation levels in brain and blood were modest at the majority of CpG sites (mean Pearson $r = 0.069$, SD $= 0.165$), which is similar to those reported in Yu et al.[33] for CD4+ lymphocytes and other previous reports[33,34]. Among CpGs mapped within the 119 significant DMRs ($n = 728$), only 11 showed moderate to strong association in brain and blood in both adjusted and unadjusted analyses (absolute $r_{beta} \geq 0.5$, $FDR_{beta} < 0.05$, absolute $r_{resid} \geq 0.5$, $FDR_{resid} < 0.05$). Similarly, among the 3751 significant individual CpGs, only 39 showed moderate to strong associations (absolute $r_{beta} \geq 0.5$, $FDR_{beta} < 0.05$, absolute $r_{resid} \geq 0.5$, $FDR_{resid} < 0.05$) in brain and blood. Remarkably, all 50 CpGs showed significant positive correlations, corroborating previous analyses[34] that also observed a significant negative correlation between brain and blood is relatively rare.

To further validate these brain-blood correlations, we performed an additional analysis using the BeCon software[35], which computed correlations between brain and blood methylation levels in an independent cohort with 16 subjects. We compared our results from the London cohort with correlation results between whole blood and Brodmann area 10 (anterior PFC) in BeCon. Among the 50 CpGs described above, 5 CpGs (cg03765423, cg16106427, cg18776287, cg22595230, cg00445443) mapped to *HOXA2, intergenic, STK32C, MRPS2,* and *CENPB* genes also exhibited moderate to strong correlation (absolute $R > 0.5$) in BA10 (Supplementary Data 8). Again, all these BA10-blood correlations were positive.

**Correlation of methylation levels of significant CpGs and DMRs in AD with expressions of nearby genes.** Among the four cohorts of brain samples, the ROSMAP study had matched RNA-seq gene expression data and DNA methylation data available for 529 samples (428 cases and 101 controls). We therefore evaluated the role of significant DMRs or CpGs by correlating methylation levels of the significant DMRs or CpGs with the expression values of genes found in the vicinity (±250 kb from the start or end of the DMR, or location of CpG). To reduce the effect of potential confounding effects, when testing for methylation-gene expression associations, we first adjusted for age at death, sex, cell-type proportions, and batch effects in both DNA methylation and gene expression levels separately and extracted residuals from the linear models. Then we tested for association between methylation residuals and gene expression residuals, adjusting for Braak stage.

We found that out of the 118 DMRs that were linked to a gene transcript (+/−250 kb), 73 (62%) DMRs were associated with gene expression levels at 5% FDR (Supplementary Data 9). Similarly, out of the 3642 CpGs that were linked to a nearby gene (+/−250 kb), we found 652 (18%) CpGs were associated with gene expression levels at 5% FDR (Supplementary Data 10). Among the significant DMR–RNA and CpG–RNA associations, 41.1% and 49.2% were negative associations, respectively. We next compared the strengths of DMR–RNA with CpG–RNA associations using a generalized estimating equations (GEE) model where values for $-\log_{10}$ (P-value) from each DMR or CpG were treated as clusters (Online Methods). In general, we found the effects of DMRs on gene expression to be larger than those for single CpGs ($P = 2.25 \times 10^{-10}$), consistent with the notion that DMRs often have a more relevant biological role than isolated CpGs.

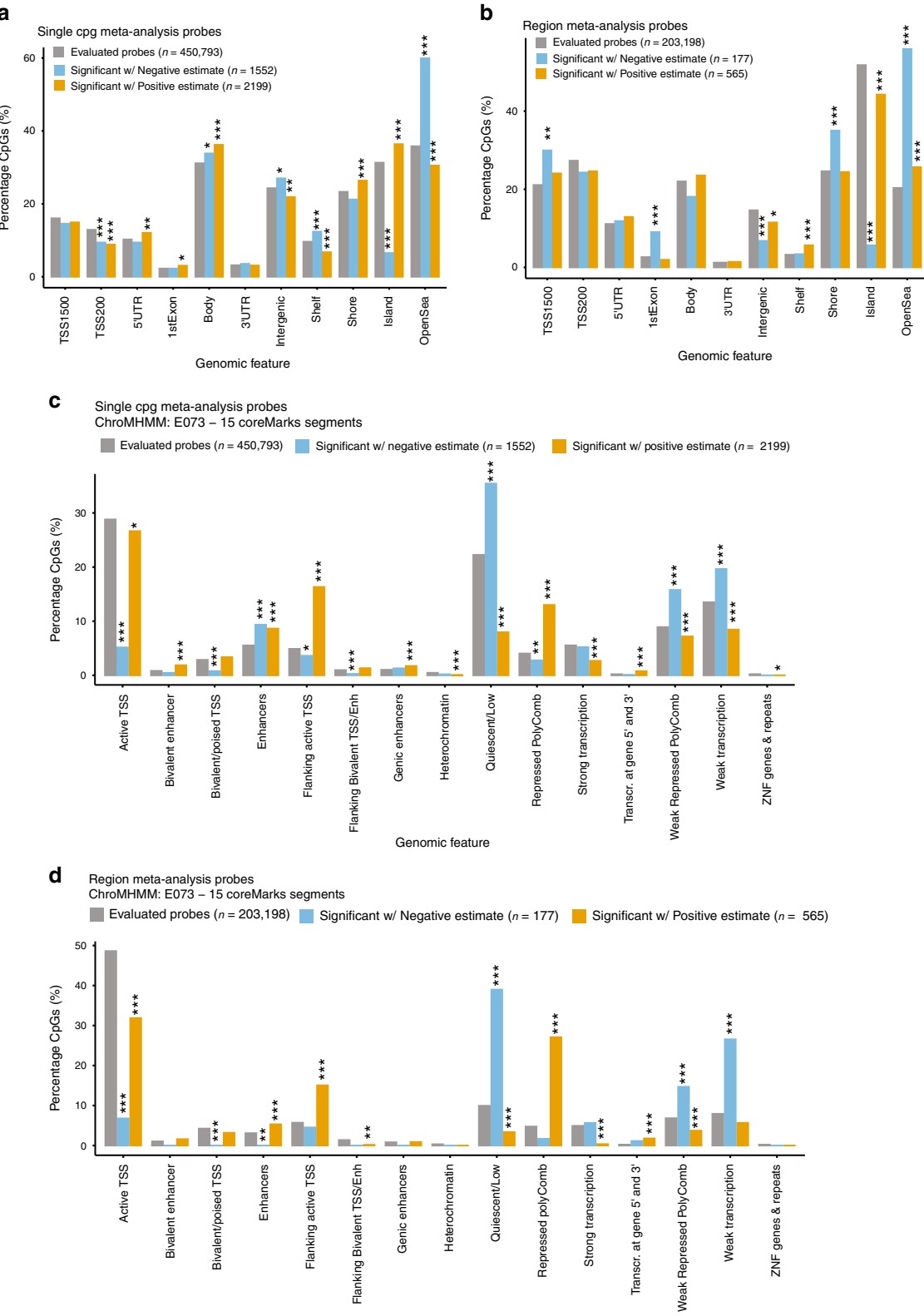

**Fig. 3 Enrichment of CpGs significantly associated with AD Braak stage in meta-analysis of individual CpGs and DMRs at 5% FDR.** A two-sided Fisher's test was used to determine over or under-representation of the significant CpGs in individual CpGs analysis and CpGs mapped within significant DMRs in various **a**, **b** genomic features and **c**, **d** chromatin states. ***P-value < 0.001, **P-value < 0.01, *P-value < 0.05, uncorrected for multiple comparisons.

**Table 4 Gene set enrichment analysis of significant methylation differences associated with AD Braak stage identified in meta-analysis using Wallenius' noncentral hypergeometric test which adjusted for different number of CpGs associated with each gene.**

| Gene Set | Description | P-value | FDR |
|---|---|---|---|
| GO:0002684 | Positive regulation of immune system process | 4.00E−07 | 2.94E−04 |
| GO:0022409 | Positive regulation of cell–cell adhesion | 3.88E−10 | 1.96E−06 |
| GO:0030217 | T cell differentiation | 3.32E−06 | 1.99E−03 |
| GO:0050863 | Regulation of T cell activation | 1.35E−07 | 1.62E−04 |
| GO:0050870 | Positive regulation of T cell activation | 4.89E−07 | 3.27E−04 |
| GO:0043312 | Neutrophil degranulation | 5.28E−05 | 1.97E−02 |
| GO:0030097 | Hemopoiesis | 1.10E−07 | 1.39E−04 |
| GO:0006952 | Defense response | 1.57E−07 | 1.78E−04 |
| GO:0098609 | Cell–cell adhesion | 3.23E−07 | 2.72E−04 |
| GO:0097530 | Granulocyte migration | 7.42E−05 | 2.68E−02 |
| GO:0002523 | Leukocyte migration involved in inflammatory response | 2.54E−06 | 1.56E−03 |
| GO:0002573 | Myeloid leukocyte differentiation | 3.76E−06 | 2.05E−03 |
| GO:0002429 | Immune response-activating cell surface receptor signaling pathway | 1.55E−05 | 7.21E−03 |
| GO:0006909 | Phagocytosis | 5.32E−06 | 2.75E−03 |
| GO:0071706 | Tumor necrosis factor superfamily cytokine production | 1.07E−04 | 3.69E−02 |
| GO:0098883 | Synapse pruning | 3.42E−05 | 1.41E−02 |
| GO:0045123 | Cellular extravasation | 5.37E−05 | 1.97E−02 |
| GO:0016477 | Cell migration | 7.73E−05 | 2.75E−02 |
| GO:0050854 | Regulation of antigen receptor-mediated signaling pathway | 1.29E−04 | 4.38E−02 |
| KEGG:hsa04640 | Hematopoietic cell lineage | 1.55E−04 | 3.96E−02 |
| KEGG:hsa04145 | Phagosome | 3.01E−04 | 3.96E−02 |
| KEGG:hsa04060 | Cytokine–cytokine receptor interaction | 3.53E−04 | 3.96E−02 |
| KEGG:hsa04062 | Chemokine signaling pathway | 5.87E−04 | 4.95E−02 |

Shown are gene ontology or KEGG database ID (Gene Set), a description of the pathway (Description), and significance assessment (P-value, FDR).

**Correlation and co-localization with genetic susceptibility loci.** To identify methylation quantitative trait loci (mQTLs) for the significant DMRs and CpGs, we tested associations between the methylation levels with nearby SNPs, using the ROSMAP study dataset (imputed to the Haplotype Reference Consortium r1.1 reference panel)[36], which had matched genotype data and DNA methylation data for 688 samples. To reduce the number of tests, we focused on identifying *cis* mQTLs located within 500 kb from the start or end of the DMR (or position of the significant CpG)[37]. Among 166,797 SNPs that are associated with AD, 11,670 were also significantly associated with methylation levels, after correcting for confounding effects age, sex, cell type, batch effects and PCs in methylation data. Among the 119 DMRs and 3751 CpGs significantly associated with Braak stage, 37 DMRs and 1010 CpGs had at least one corresponding mQTL in brain samples, respectively (Supplementary Data 11 and 12).

To evaluate if the significant methylation differences overlap with genetic risk loci implicated in AD, we compared enrichment of significant CpGs and DMRs identified in this study with the 24 LD blocks of genetic variants reaching genome-wide significance in a recent AD meta-analysis[3]. We found that while no DMRs overlapped with the 24 LD blocks, 24 FDR significant individual CpGs overlapped with genetic variants mapped to the *HLA-DRB1, TREM2, NYAP1, SPI1, MS4A2, ADAM10, ACE*, and *ABCA7* genes (Supplementary Data 13).

Given the observed overlap between AD pathology associated CpGs and AD genetic risk loci, we next sought to determine whether the association signals at the GWAS loci (variant to AD status as determined by clinical consensus diagnosis of cognitive status, and variant to CpG methylation levels) are due to a single shared causal variant or to distinct causal variants close to each other. To this end, we performed a co-localization analysis using the method described in Giambartolomei et al.[38]. The results of this co-localization analysis strongly suggested[39] (i.e., PP3 + PP4 > 0.90, PP4 > 0.8 and PP4/PP3 > 5, Online Methods) that 2 out of

the 24 regions included a single causal variant common to both phenotypes (i.e., AD status and CpG methylation levels). The CpGs associated with these causal variants are located on the *SPI1*[30] and *ADAM10*[40] genes (Supplementary Data 14).

**Genomic inflation and sensitivity analysis.** In this study, we chose to use the false discovery rate, instead of the more stringent genome-wide significance threshold, to select significant DNA methylation differences for enrichment analysis. This was motivated by the observation that in brain disorders such as AD, the DNA methylation differences in the epigenome are often found to be modest, so they might be missed by using the more conventional genome-wide significance threshold. To assess the potential inflation in our results, we estimated genomic inflation factors using both the conventional and the *bacon* method[41], specifically proposed for EWAS. As shown by simulation studies[41], real datasets[41], and theory[42], the conventional genomic inflation factor (lambda or $\lambda$ used interchangeably below) is dependent on the expected number of true associations. Because in a typical EWAS it is expected that small effects from many CpGs might be associated with the phenotype, the genomic inflation factor would overestimate actual test–statistic inflation. To estimate genomic inflations more accurately in EWAS, Iterson et al.[41] developed a Bayesian method that estimates inflation in EWAS based on empirical null distributions, which is implemented in the Bioconductor package bacon. The estimated genomic inflation factors for individual cohorts in our study were modest and comparable to other recent EWAS profiling brain tissues (for example, Supplementary Fig. 10 in Viana et al.[43] showed that lambdas for different brain regions ranged from 1.02 to 1.23). In our study, lambdas ($\lambda$) by conventional approach ranged from 1.002 to 1.249, and lambdas based on the *bacon* approach ($\lambda.bacon$) ranged from 0.986 to 1.082 (Supplementary Fig. 1). In particular, for the ROSMAP cohort with the largest sample size,

genomic inflation factors were close to 1 by both approaches ($\lambda = 1.016$, $\lambda.bacon = 1.002$) Genomic inflation factors for the meta-analysis were $\lambda = 1.264$ and $\lambda.bacon = 1.086$.

In addition, we conducted sensitivity analyses to evaluate the impact of inflation on our enrichment analysis results. To this end, we performed inflation correction for single cohort effect sizes and standard errors, and then meta-analyzed the *bacon*-corrected effect sizes and standard errors. Enrichment analysis was then conducted for the 2767 CpGs that reached FDR significance and the 339 CpGs that reached the $2.4 \times 10^{-7}$ genome-wide significance level[44] (Supplementary Data 15). Our results showed that the functional enrichment results in these sensitivity analyses were largely congruent with our main analysis results (Supplementary Figs. 2–3 and Supplementary Data 16–18). In particular, we still observed significant enrichment in polycomb repressed regions for hyper-methylated CpGs in AD, as well as in binding sites of polycomb repressive complex 2 subunits EZH2 and SUZ12. Moreover, pathway analysis still showed significant enrichment in immune system related pathways (Supplementary Data 19–20) for both FDR significant and genome-wide significant CpGs after *bacon* correction.

## Discussion

We conducted a comprehensive meta-analysis of four cohorts of prefrontal cortex brain samples to prioritize consistent DNA methylation differences involved in pathological AD. Our study illustrates the power of meta-analysis for EWAS. In individual cohort analysis, we obtained between 0 and 40 FDR significant CpGs per cohort. In comparison, we obtained 3751 FDR significant CpGs in our meta-analysis. With the larger sample size utilized by this meta-analysis, in addition to replicating previous cohort-specific analysis results, we were able to identify many new genomic regions and sites associated with AD pathology.

To reduce the concerns about false positives, we employed several strategies in addition to the sensitivity analysis described above. First, we made sure that our analyses were robust to different pre-processing pipelines. To this end, we compared our single-CpG analysis results for the London and Mt. Sinai cohorts using our preprocessing pipeline with those used in a previous study[15] and we found the use of different preprocessing methods did not influence the results of statistical analysis (Supplementary Data 21). Second, in all our analyses, including the integrative analysis of DNA methylation with gene expression or genetic variants, we adjusted for potential confounding effects including age at death, sex, estimated cell-type proportions, and batch effects. Third, as CpG sites located in gene regulatory regions often act as functional units when regulating gene expression, they usually have a high degree of co-methylation. Therefore, in addition to identifying CpGs associated with AD, we also identified differentially methylated regions by intersecting two DMR analysis methods, coMethDMR and comb-p, which allowed us to borrow information across multiple CpGs within a region to improve both sensitivity and specificity. While a recent review[9] commented that various DMR tools often agreed poorly in single cohort analysis, here we reported robust findings for 119 DMRs that were identified by two methods that are vastly different in methodology, which is probably due to the larger sample size utilized by our meta-analysis. Fourth, as an alternative strategy to reduce confounding by sex and age at death, we also employed a matching design that matched each case with a control sample of the same sex and age at death. While this greatly reduced the sample sizes, we still were able to identify 32 DMRs and 151 CpGs at FDR significance with substantial overlap with our main meta-analysis results. Together, these strategies enabled us to identify a list of informative and unbiased regulatory changes in DNA methylation associated with AD pathology. Remarkably, we found a substantial number of our top results were consistent with recent AD DNA methylation literature (Supplementary Data 1 and 2).

Our meta-analysis of individual CpGs identified many of the loci previously reported in single cohort DNA methylation analysis[8,14–16] include genes such as *HOXA3*, *ANK1*, *RHBDF2*, *SLC44A2*, and *BIN1*. Notably, among the 3751 FDR significant CpGs (Supplementary Data 1), 151 CpGs were mapped to the group of HOX genes on chromosome 7, where aberrant methylation in a 48 kb region near the *HOXA* gene cluster has been shown to be associated with AD neuropathology in multiple AD EWAS datasets[14–16]. Two CpGs in the *HOXA3* gene (cg22962123 and cg01301319) also ranked as the most significant association in our meta-analysis of individual CpGs. Another noteworthy gene was *AGAP2*, also known as *PIKE*, which mediates the neuroprotective effects of BDNF in response to amyloid beta-induced toxicity[46]. Interestingly, this gene is observed to be expressed mainly in neurons[46]. Our meta-analysis result of significant promoter hyper-methylation in *AGAP2* is consistent with previous studies that suggested this gene might be involved in reduced functionality of BDNF and subsequent neuronal death in AD[46].

The significant DMRs were identified by both comb-p and coMethDMR software, which allowed us to determine high confidence regions with consistent changes in multiple CpGs across cohorts. In addition, coMethDMR highlighted co-methylated regions within each cohort (Supplementary Fig. 4). The most significant DMR was in the promoter region of the *MAMSTR* gene, which was consistently hypermethylated in AD across the four cohorts (Supplementary Fig. 5). *MAMSTR* is a transcriptional coactivator that stimulates Myocyte Enhancer Factor-2C (MEF2C), which was recently found to modulate microglial responses to promote homeostasis under proinflammatory conditions in the aging brain[47]. Genetic variants in the *MEF2C* locus were found to be associated with LOAD in a recent meta-analysis[2]. Our result is consistent with previous studies that suggested loss of *MEF2C* function might contribute to the increased sensitivity of microglia to immune stimuli, as commonly observed in neurodegenerative diseases such as AD[47].

In addition to corroborating previous findings in single-cohort analysis using the same methylation datasets (Supplementary Data 1 and 2), our meta-analysis also uncovered a number of novel differentially methylated genes. For example, among genes that overlapped with the top 20 DMRs (Table 3), is *CAMTA1* (calmodulin binding transcription activator 1), for which variants were shown to be associated with episodic memory[48] and more recently with immediate recall in GWAS studies[49]. Given memory impairment is an early feature of AD, DNA methylation differences affecting this gene are particularly relevant. Another cardinal feature of AD is chronic neuroinflammation. In pathological AD samples, our meta-analysis identified significant hypomethylation in the promoter region of the *AZU1* gene, also known as *CAP37*, which is a neutrophil granule protein that helps defend the host against microbial pathogens and regulate inflammation[50]. Previously, mRNA expression levels of *CAP37* were observed to be upregulated in AD patients[51,52]. It has been suggested that in AD, elevated levels of amyloid beta and proinflammatory cytokines might trigger upregulation of *CAP37*, which then activates microglial cells[50], whose function has been observed to be dysregulated in AD. Taken together, these results demonstrated our meta-analysis replicated methylation differences in a number of genes previously implicated in pathological AD, as well as nominated additional genes likely to be involved in AD pathogenesis.

As in previous single cohort studies[8,14,19,20], we observed that the majority of the significant DNA methylation differences were hypermethylated in pathological AD. Our enrichment analysis brought to light the potential roles of these hypermethylations in regulating active and repressive elements in pathological AD. Comparison of the DNA methylation differences with genomic annotations and additional epigenomic features such as chromatin states showed these hyper-methylated changes were enriched in CpG islands, gene body, flanking active TSS, and polycomb repressed regions where many developmental genes such as the Hox genes are located. An integrative analysis of the DNA methylation differences with ChIPSeq performed by the ENCODE project showed these hypermethylations were also enriched in the binding sites of two chromatin proteins, EZH2 and SUZ12, both of which are core subunits of the polycomb repressive complex 2 (PRC2) protein complex.

PRC2 is a type of polycomb group (PcG) protein and plays important roles in multiple biological processes including proliferation and differentiation as well as maintenance of cellular identity through regulation of gene expression. The PRC2 protein complex is composed of three core component proteins: SUZ12, EED, and either EZH1or EZH12, which along with RBBP4 or RBBP7 forms distinct subcomplexes by associating with different interaction partners. It is highly conserved and was originally discovered in Drosophila studies as a suppressor of Hox genes[53,54]. It is well known that PRC2 mediates the methylation of Histone 3 lysine 27 (H3K27). Methylation of H3K27 negatively regulates gene expression via chromatin compaction and shows a highly dynamic profile during developmental transitions[55–57]. Of particular relevance to AD, PRC2-mediated gene silencing has also been observed in adult brains, long after the completion of neuronal differentiation[58]. Furthermore, PRC2 was recently shown to regulate neuronal lineage specification and to maintain neuronal functions. Most importantly, PRC2 silences genes involved in neurodegeneration and its deficiency leads to the de-repression of developmental regulators such as the Hox gene clusters, which manifest in progressive and fatal neurodegeneration in mice[58].

Previously, substantial cross-talk has been observed between PRC2 and DNA methylation, two key epigenetics mechanisms for gene repression[26–28]. Although the targeting of PRC2 to methylated CpGs has also been shown in vitro[59], PRC2 typically binds to unmethylated CpG islands[60] at the promoters of inactive developmental genes. Dysfunction of PRC2 is associated with alterations of DNA methylation of CpGs in promoters of developmental genes[61]. Conversely, it has also been shown that loss of DNA methylation results in enhanced H3K27me3, suggesting that PRC2 can serve as a back-up repressive complex for newly hypomethylated CpGs. In particular, epigenetic switching has been observed in development[62] and cancer cell lines[63,64], wherein CG-rich regions of genes silenced by PRC2 lose their polycomb marks but gain DNA methylation and remain repressed. Our meta-analysis revealed substantial hypermethylation in AD, which was significantly over-represented in polycomb repressed regions (Fig. 3 and Supplementary Data 4, 5). These observations prompted us to hypothesize that epigenetic switching might also be involved in AD, where defects in PRC2 functionality might have influenced the observed DNA hypermethylation changes, to continue to repress the PRC2 target genes. When we examined PRC2 target genes[58], we found many of these genes had low normalized gene expression levels in both pathological AD cases (Braak stage 3–6) and controls (Braak stage 0–2) in the ROSMAP samples (Supplementary Data 22), consistent with recent studies that have suggested DNA methylation and PRC2 often complement each other in gene silencing[26,27,63]. Although epigenetic switching do not cause de novo repression, it

might potentially participate in AD pathogenesis by modifying epigenetic plasticity[64]. Future studies that perform high throughput sequencing on chromatin modification, DNA methylation, and gene expression changes in parallel would help elucidate the complex interactions between these epigenetic mechanisms underlying AD.

The results of our pathway analysis overwhelmingly point to immune system alterations in pathological AD. A main hypothesis in AD pathogenesis states that accumulation of amyloid beta in the brain triggers a cascade of events that culminate with neuronal death and brain atrophy, and in response the brain activates innate immune responses that include microglia and astrocyte changes[31,32]. Indeed, genetic variants that increase AD risk include astrocyte and microglia expressing genes involved in the innate immune system and inflammation[2,65–67]. While previous studies have associated AD with inflammatory responses through genetics, gene expression, or proteins[68–70], our study provides strong support that these changes can also be observed in the epigenome, in particular in DNA methylation.

Although overall brain and blood DNA methylations did not correlate well[8,34], we did observe robust brain-blood correlations for a few significant methylation differences in both the London[8] and Edger et al.[35] cohorts, possibly reflecting the systemic inflammation associated with AD[32]. Future studies are needed to further evaluate and validate these potential biomarkers before their adoption in clinics.

Traditionally neuroinflammation in AD has been viewed as a reactive process, however more recent studies supported the notion that immune actions occur early in the disease course and can, at least at a given time point, drive and sustain AD pathology[31,71,72]. A recent gene expression network analysis nominated the immune-specific and microglia-specific co-expression module as the most significant module associated with AD pathology[68]. Moreover, recent GWAS studies have demonstrated that AD-associated common and rare genetic variants are linked to genes that regulate immune processes[2,65–67] including *ABCA7, CLU, CR1, MS4A4E/MS4A6A, CD33, EPHA1, HLA-DRB5, HLA-DRB1, INPP5D, MEF2C, SORL1*, and *TREM2*. Notably, several of these AD-associated genes (*CD33, MS4A4E/MS4A6A, TREM2*, and *ABCA7*) contained binding sites for the transcription factor PU.1, a central hub in the AD gene network that is critical for regulating microglial gene expression[73]. Previously, PU.1 binding sites were shown to be enriched in epigenomic signals in AD mouse models using ChIPseq experiments[74]. Consistent with these previous studies, we observed significant differential methylation at several immune-related genes with PU.1 binding sites in our meta-analysis, including *MS4A6A, MS4A4A*, and *TREM2* genes as well as the *SPI1* gene that encodes PU.1. Interestingly, the most significant DMR identified in this meta-analysis is located on the *MAMSTR* gene, which encodes a cofactor that interacts with MEF2[75], a transcription factor that cooperates with PU.1[76,77].

As amyloid beta might be deposited in the brain decades before the onset of clinical symptoms, DNA methylation differences that reflect inflammatory responses might provide a useful source of biomarkers for early detection of AD. It has also been proposed that targeting glia cells and reducing neuroinflammation may be a viable approach for delaying the onset and progression of AD[78]. For example, the HDAC-inhibitor Vorinostat was recently shown to be effective at reducing transcription factor PU.1 expression in human microglia[73]. On the other hand, epigenetic therapies that leverage the plastic nature of DNA methylation might also be an alternative strategy to modulate inflammation[79].

Several limitations of the current study are in order. First, the 450k arrays used by the studies analyzed in this meta-analysis cannot distinguish 5-methylcytosine (5mC) from 5-

hydroxymethylcytosine (5hmC). This complicates the biological interpretation of the observed methylation differences. Efforts to identify 5hmC in AD are underway and they provide an enhanced understanding of the biological role of DNA modifications[80–83]. Meta-analysis strategies such as the one employed here could be similarly applied to large datasets that discriminate between 5mC and further oxidized DNA modifications. Also, although using target tissues to study brain disorders is often preferred, there are still issues with using postmortem brain tissue in such studies. For example, it typically represents the end stages of the disease, and it is still unclear if there are postmortem changes in methylation patterns and how postmortem intervals might affect them. In this study, we did not adjust for postmortem interval (PMI), because in the ROSMAP cohort PMI was not significantly associated with Braak stage (Spearman correlation $r = -0.018$, $P$-value $= 0.6156$), so is unlikely to be a confounder for it. Also, PMI was not available for the other three public datasets (London, Mount Sinai, Gasparoni) we analyzed. Furthermore, the methylation levels in the studies used here were measured on the bulk prefrontal cortex, which contains a complex mixture of cell types. To reduce confounding effects due to different cell types, we included estimated cell-type proportions of each brain sample as a covariate variable in all our analyses. Currently, a challenge with cell-type specific studies is that they are often limited to smaller sample sizes due to labor-intensive sample preparation procedures and therefore have limited statistical power. Finally, the associations we identified do not necessarily reflect causal relationships. Future studies that employ longitudinal designs with AD endophenotypes are needed to identify causal changes in DNA methylation as AD initiates and progresses.

In summary, we have identified numerous methylation differences at DMRs and CpGs consistently associated with AD Braak stage in multiple cohorts. Enrichment analysis of these significant methylation differences highlights the particularly relevant roles of PRC2 and immune processes in AD pathology. Our analysis results suggest a meta-analysis that synthesizes information from multiple large cohorts might be a useful strategy for uncovering the epigenetic architecture underlying AD. These findings will be valuable for designing future studies that more precisely map AD-associated changes in the epigenome.

## Methods

**Study cohorts**. Our meta-analysis included a total of 1030 brain samples in four independent cohorts, collected from four different brain banks. The ROSMAP cohort included samples from the Religious Order Study (ROS) and the Memory and Aging Project (MAP)[84]. Samples of the Mount Sinai cohort were obtained from the Mount Sinai Alzheimer's Disease and Schizophrenia Brain Bank, previously described in Smith et al.[15]. The London cohort included samples obtained from the MRC London Brain Bank for Neurodegenerative Disease, previously described in Lunnon et al.[8]. Samples in the fourth cohort were obtained from the Gasparoni et al.[16] study. In all these datasets, brain samples were classified according to Braak stage[85], with scores ranging from 0 (control) to VI (late stage tau pathology AD), indicating different levels of severity of the disease. In addition, to assess the diagnostic utility of DNA methylation as clinical biomarkers, we also studied methylation changes in premortem whole blood samples from a subset of subjects in the London cohort.

**Preprocessing of DNA methylation data**. All methylation datasets were measured by the same Illumina HumanMethylation 450k beadchip, which included more than 450,000 methylation sites primarily at genic regions and CpG islands[86]. Supplementary Data 23 shows the number of CpGs and samples removed at each quality control step. Quality control for CpG probes included several steps: First, when raw.idat files were available for the cohort, we selected probes with detection $P$-value $< 0.01$ for all the samples in the cohort. A small detection $P$-value corresponds to significant difference between signals in the probes compared to background noise. Next, probes on the X and Y chromosomes were removed, as were those in which a single nucleotide polymorphism (SNP) with minor allele frequency (MAF) $\geq 0.01$ was present in the last five base pairs of the probe. We did not remove cross-reactive probes[18] or probes associated with cigarette smoking[17], but chose to examine them posthoc in the list of significant DMRs and CpGs, as was previously done in other large scale meta-analyses[87]. Quality control for samples included restricting our analysis to samples with good bisulfite conversion efficiency (i.e., ≥88%) and principal component analysis (PCA). More specifically, PCA was performed using the 50,000 most variable CpGs for each cohort. Samples that were within ±3 standard deviations from the mean of PC1 and PC2 were selected to be included in the final sample set.

The quality controlled methylation datasets were next subjected to the QN. BMIQ normalization procedure as recommended by a recent systematic study of different normalization methods[88]. More specifically, we first applied quantile normalization as implemented in the lumi R package to remove systematic effects between samples. Next, we applied the β-mixture quantile normalization (BMIQ) procedure[45] as implemented in the wateRmelon R package[89] to normalize beta values of type 1 and type 2 design probes within the Illumina arrays. To benchmark our pre-processing pipeline, we compared our single CpG analysis results for the London and Mount Sinai cohorts with published results[15] using alternative *dasen* pre-processing pipeline[89] (Supplementary Data 21).

**Meta-analysis**. First, we performed cohort specific analyses for individual CpGs. The association between CpG methylation levels and Braak stage was assessed using linear statistical models in each cohort. Given that methylation $M$-values (logit transformation of methylation beta values) has better statistical properties (i.e., homoscedasticity) for linear regression models[90], we used the $M$-values as the outcome variable in our statistical models. We adjusted for potential confounding factors including age at death, sex, methylation slide effects, and cell-type proportions (i.e., proportions of neurons) in the samples estimated by the CETS R package[91]. For the ROSMAP cohort, we also included the variable "batch" that was available in the dataset to adjust for technical batches which occurred during data generation.

To meta-analyze individual CpG results across different cohorts, we used the meta R package. The evidence for heterogeneity of study effects was tested using Cochran's Q statistic[92]. The inverse-variance weighted fixed effects model was applied to synthesize statistical significance from individual cohorts. Although the fixed effects model for meta-analysis does not require the assumption of homogeneity[93], for those regions with nominal evidence for heterogeneity (nominal $P_{heterogeneity} < 0.05$), we also applied random effects meta-analysis[94] and assigned final meta-analysis $P$-value based on the random effects model.

For region based meta-analysis, we used two analytical pipelines, the comb-p[12] approach and the coMethDMR[13] approach, and selected significant DMRs identified by both methods. Briefly, comb-p takes single CpG $P$-values and locations of CpG sites to scan the genome for regions enriched with a series of adjacent low $P$-values. In our analysis, we used meta-analysis $P$-values of the four brain samples cohorts as input for comb-p. As comb-p uses the Sidak method[95] to account for multiple comparisons, we considered DMRs with Sidak $P$-values less than 0.05 to be significant based on comb-p. We used the default setting for our comb-p analysis, with parameters --seed 1e−3 and --dist 200, which required a $P$-value of $10^{-3}$ to start a region and extend the region if another $P$-value was within 200 base pairs.

In the coMethDMR approach, we performed cohort specific analyses for genomic regions first. We define "contiguous genomic regions" to be genomic regions on the Illumina array covered with clusters of contiguous CpGs, where the maximum separation between any two consecutive probes is 200 base pairs. First, coMethDMR selects co-methylated subregions within the contiguous genomic regions. Next, we summarized methylation M values within these co-methylated subregions using medians and tested them against AD Braak stage. In the same way as in single CpG analyses, we adjusted for potential confounding factors including age at death, sex, methylation slide, and cell-type proportions in the samples estimated by the CETS R package[91]. For the ROSMAP cohort, we also included the variable "batch" that was available in the dataset to adjust for technical batches which occurred during data generation. The cohort specific $P$-values for each contiguous genomic region were then combined across cohorts using the inverse-variance weighted fixed effects meta-analysis model (or inverse-variance weighted random effects model if test of heterogeneity had a $P$-value less than 0.05) as described above. Note that the coMethDMR approach allowed us to assess among cohort heterogeneities for genomic regions. DMRs with less than 5% false discovery rate (FDR) were considered to be significant based on coMethDMR. We then selected DMRs that are significant by both comb-p and coMethDMR. Finally, we excluded CpGs and DMRs that overlapped with any of the 2623 CpGs associated with smoking identified in Joehanes et al.[17] or any of the cross-reactive probes identified in Chen et al.[18].

To prioritize methylation changes most likely to be affected by the AD pathogenesis process, we also performed an analysis using an alternative strategy to control for confounding effects for age and sex. More specifically, we first matched each case with a control sample with the same age at death (in years) and sex in the same cohort using the matchControls function in the e1071 R package. When there are multiple matched control samples, the control sample with the most similar age as the case sample is selected. The age and sex matched samples were then analyzed in the same way as described above, except for removing age at death and sex effects in the linear models.

To assess inflation of test statistics in this study, we used Quantile–quantile (QQ) plots of observed and expected distributions of P-values for each cohort. In addition, we also computed genomic inflation factors for each cohort and meta-analysis, using both the conventional approach and the more recently proposed *bacon* method as described in Iterson et al.[41] and implemented in the bacon R package.

**Functional annotation of significant methylation changes.** The identified methylation changes at individual CpGs and DMRs were annotated using both the Illumina (UCSC) gene annotation and the GREAT (Genomic Regions Enrichment of Annotations Tool) annotation[96] with the default "Basal plus method" that associates genomic regions to regulatory domain of genes.

To test for over-representation and under-representation of significant DMRs in different types of genomic regions with respect to CpG islands or genes, we used Fisher's exact test, which compared the proportion of CpGs within significant DMRs that mapped to a particular type of genomic region (e.g., CpG islands) (foreground) to the proportion of CpGs in contiguous genomic regions covered by CpGs on the array that mapped to the same type of genomic region (background). Similarly, we also used Fisher's test to assess enrichment of significant DMRs in different chromatin states by comparing with the 15-chromatin state data estimated with ChromHMM[21] using a DLPFC tissue sample (E073) from the Roadmap Epigenomics Project[23]. The enrichment of significant CpGs, or DMSs (differentially methylated sites), were tested in the same way, except for replacing foreground with significant DMSs and background with all probes on the array. In addition, we also explored an alternative enrichment analysis that accounts for correlations between CpGs using a logistic mixed effects regression model. More specifically, for each type of genomic feature (e.g., CpG island), we tested for an association between the type of genomic region (e.g., isCpGisland = "yes" or "no") and significance of the CpG (e.g., isSignificant = "yes" or "no"). Random effects for each chromosome were also included in this model to account for correlations between CpGs within the same chromosome. This analysis was performed using SAS procedures GLIMMIX and HPMIXED, which implemented specialized high-performance techniques designed to cope with estimation problems in mixed effects models with large datasets. The enrichment analysis results based on the mixed model were similar to those based on Fisher's test (Supplementary Data 24).

To identify biological pathways enriched with significant methylation changes, because significant DMRs and DMSs can co-localize to the same gene, we combined CpGs from DMRs and DMSs and tested for enrichment using the missMethyl R package[97]. We grouped the FDR significant GO terms into several clusters based on similarity of their member genes, using the Jaccard similarity index. A total of 19 clusters of GO terms was obtained and we selected one term to represent each resulting cluster.

**Correlation of significant DMRs with expression of nearby genes.** The ROS-MAP study also generated RNA-seq data for a subset of samples with available DNA methylation data. We used 529 samples with matched DNA methylation and gene expression data for this analysis. More specifically, normalized FPKM (Fragments Per Kilobase of transcript per Million mapped reads) gene expression values for the ROSMAP study were downloaded from the AMP-AD Knowledge Portal (Synapse ID: syn3388564). Next, for each significant DMR identified in the meta-analysis, we first removed confounding effects in DNA methylation data by fitting the model median methylation M value ~ neuron.proportions + batch + sample.plate array + ageAtDeath + sex and extracting residuals from this model, which are the methylation residuals. Similarly, we also removed potential confounding effects in RNA-seq data by fitting model log2(normalized FPKM values + 1) ~ ageAtDeath + sex + markers for cell types. The last term, "markers for cell types," included multiple covariate variables to adjust for the multiple types of cells in the brain samples. More specifically, we estimated expression levels of genes that are specific for the main five cell types present in the CNS: ENO2 for neurons, GFAP for astrocytes, CD68 for microglia, OLIG2 for oligodendrocytes, and CD34 for endothelial cells, and included these as variables in the above linear regression model, as was done in a previous large study of AD samples[14]. The residuals extracted from this model are the gene expression residuals. For each gene expression and DMR pair, we then tested the association between gene expression residuals and methylation residuals using a linear model: gene expression residuals ~ methylation residuals + Braak stage. For significant CpGs this analysis was repeated, except for replacing median methylation level in the DMR with methylation level of the CpG, and correlating with expression values of genes found ±250 kb away from the CpG. To compare the strengths of DMR–RNA with CpG–RNA associations, we used a generalized estimating equations (GEE) model where $-\log_{10}$ (P-value) from each DMR or CpG were treated as clusters. The GEE model included $-\log_{10}$ (P-value) of the DNA methylation to RNA associations as the outcome variable, and is DMR (yes/no) as the independent variable. We assumed an exchangeable working correlation structure for the clusters of correlated observations (i.e., values of $-\log_{10}$ (P-value) for the same DMR or CpG), along with log link and gamma distribution for the outcome variable.

**Correlation and co-localization with genetic susceptibility loci.** The GWAS regions associated with AD were obtained from Supplementary Data 8 of the recent AD meta-analysis described in Kunkle et al.[3], which identified 24 LD blocks with genetic variants reaching genome-wide significance. For the methylation quantitative trait loci (mQTLs) analysis, we used the ROSMAP study dataset, which also had matched genotype data and DNA methylation data for 688 samples. More specifically, ROSMAP genotype data was downloaded from AMP-AD (syn3157325) and imputed to the Haplotype Reference Consortium r1.1 reference panel[36]. To reduce the number of tests, we focused on *cis* mQTLs located within 500 kb from the start or end of the DMR as previously done[37]. We additionally required SNPs to (1) have minor allele frequency of at least 1%, (2) be imputed with good certainty: information metric (info score) $\geq 0.4$, and (3) be associated with AD case–control status (as determined by clinical consensus diagnosis of cognitive status), after adjusting for age, sex, batch, and the first three PCs estimated from genotype data, at nominal P-value less than 0.05. We then fit the linear model methylation residual ~ SNP dosage + batch + PC1 + PC2 + PC3, where PC1, PC2, and PC3 are the first three PCs estimated from genotype data, to test the association between methylation residuals in CpGs and the imputed allele dosages for SNPs to identify mQTLs. The analysis for DMRs is the same except for replacing methylation residual with median (methylation residuals) of all CpGs located within the DMR. For co-localization analysis, we used the R package *coloc* to compare association signals in the AD GWAS meta-analysis[3], which associated genetic variants with AD status (trait 1), with results from a mQTL analysis which associated genetic variants with methylation levels at CpGs within 500 kb of the 24 LD blocks (trait 2) computed using the ROSMAP dataset. Given observed data for trait 1 and trait 2, the co-localization analysis computes Bayesian posterior probabilities (*PPi*) for each of the following five hypotheses[38]: (a) $H_0$: No association with either trait; (b) $H_1$: Association with trait 1, not with trait 2; (c) $H_2$: Association with trait 2, not with trait 1; (d) $H_3$: Association with trait 1 and trait 2, two independent SNPs; and (e) $H_4$: Association with trait 1 and trait 2, one shared SNP. We set prior probabilities at their default values. The summary statistics obtained in Kunkle et al.[3] were downloaded from https://www.niagads.org/igap-rv-summary-stats-kunkle-p-value-data (file "Kunkle_etal_Stage1_results.txt").

**Correlation of methylation changes in brain and blood samples.** Using the London cohort[8], which consisted of 69 samples with matched PFC and blood samples, we compared brain-blood methylation levels in significant CpGs and those CpGs mapped within significant DMRs using Spearman correlations. Two approaches were used to quantify methylation levels: beta values or corrected methylation levels (i.e., methylation residuals adjusted for estimated neuron proportions for brain samples (or estimated blood cell-type proportions), array, age at death (for brain samples) or age at blood draw (for blood samples), and sex as described above). In addition, we also conducted look up analysis using the BeCon tool[35], which compared brain-blood methylation levels of Broadmann areas 7, 10, and 20 in postmortem samples of 16 subjects.

In all analyses, to account for multiple comparisons, we computed FDR using the method of Benjamini and Hochberg[98]. Associations with 5% or less FDR were considered to be FDR significant. All analyses were performed using the R software (https://www.r-project.org/; version 3.6), Python software (version 2.7.12), PLINK (version 2), and SAS (version 9.4).

**Reporting summary.** Further information on experimental design is available in the Nature Research Reporting Summary linked to this paper.

## Data availability

All datasets analyzed in this study are publicly available as described in Table 1 and "Methods" section. The Mt. Sinai, London, Gasparoni and ROSMAP datasets can be accessed from GEO (accessions GSE80970, GSE59685, GSE66351) and Synapse (https://doi.org/10.7303/syn3157275). Source data are provided with this paper.

## Code availability

The scripts for the analysis performed in this study can be accessed at https://github.com/TransBioInfoLab/ad-meta-analysis. The version number of the software used can be accessed at end of each individual script under "Session information".

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

## Acknowledgements

This research was supported by US National Institutes of Health grants R21AG060459 (L.W), R01AG062634 (E.R.M, B.W.K, L.W.), and 1R01AG060472 (E.R.M). The ROSMAP study data were provided by the Rush Alzheimer's Disease Center, Rush University Medical Center, Chicago. Data collection was supported through funding by NIA grants P30AG10161, R01AG15819, R01AG17917, R01AG30146, R01AG36836, U01AG32984, and U01AG46152, the Illinois Department of Public Health, and the Translational Genomics Research Institute.

## Author contributions

L.W., J.Y., E.R.M., L.Z., T.C.S., and L.G. designed the computational analysis. L.Z., T.C.S., L.G., M.S., K.L.H.-N., and L.W. analyzed the data. L.W., J.Y., E.R.M, B.K., and X.C. contributed to interpretation of the results. L.Z. and L.W. wrote the paper, and all authors participated in the review and revision of the manuscript. L.W. conceived the original idea and supervised the project.

## Competing interests

The authors declare no competing interests.
