## [Peer Review File · Nature Communications]

REVIEWER COMMENTS

Reviewer #1 (Remarks to the Author):

Zhang et al. examined epigenome-wide associations with Braak stages in several brain tissue samples. They identified a range of CpG sites as well as DMRs within a meta-analytical framework. The authors followed up on these to assess the relationship with gene expression and blood-based methylation patterns.

The paper is very well written and the analyses thorough. This manuscript is actually (!) interesting with noteworthy findings and excellent method descriptions. I also very much appreciate that the authors published their analysis protocols on github.

I have the following comments:

1) The authors identified a rather large amount of individual CpGs. I wonder if the authors observed an inflation of test results, but I couldn't find a QQ plot / lambda. Please add and discuss, if indeed there is evidence of inflation.

2) I found the overlap results between individual sites and DMR really interesting. However, I'm not sure I agree with the authors that these analyses identified "complementary regions". Isn't it rather the case that the DMRs are a subset of the individual sites? I.e. the authors identified 3k individual sites, of which a subset extended into regions? (and related to this, what was the average number of CpGs per DMR?)

3) I really liked the alternative approach of matching by age, in addition to controlling for age. It wasn't very clear if samples were matched within or across cohorts? I.e. is it possible that differences in the age-matched analysis merely reflect cohort effects (if matched across cohorts)?

4) I'm not sure the conclusion that "significant findings in this small sample analysis are still valid". Reduced power can also increase the number of false positives. Rephrase?

5) The authors say that 75% and 44% of the age-matched results matched those of the main analysis in terms of significance. Is the direction of effect also consistent? Please add.

6) Re blood-brain correlation results: the authors report for example 39 of 3,751 probes showing strong correlations. Isn't that just even lower than the 5% level you would expect to observe by chance? I.e. evidence for cross-tissue correlation wasn't actually that convincing overall?

7) Re methylation – expression results: the authors argue that this was done separately in cases and controls as this association might be different in these groups. However, wouldn't a more sensible approach be a regression with group as a covariate?

8) Did I read this correctly that in the methylation analysis, Braak stage was included as an ordinal/continuous variable, but dichotomized in the gene expression analysis? Why was that?

Minor comments:

1) In the blood-brain correlation section, the authors report on "Rresid" and "Rbeta". Are these correlation coefficients? I.e. should this be a lower-case "r"?

2) The quality threshold of bisulfite conversion efficiency was set to 88%. That seems to be rather arbitrary. What was the justification for doing so?

3) The authors equate a fixed-effect meta-analysis with an inverse-variance weighted one. I'm not sure that's correct (e.g. random effect MA could also be IVW). Can the authors explain and/or provide a reference?

4) In the methods section: "More specifically, we first matched each case with a control sample using matchControls function in e1071 R" – please add what the pairs were match by. E.g. "age-matched each case..."

Reviewer #2 (Remarks to the Author):

This paper analyses multiple brain-based DNA methylation datasets to identify epigenetic correlates of Alzheimer's disease in the prefrontal cortex. The study applies multiple approaches (EWAS, region-based methylation analyses, mQTL overlap and enrichment), identifying 3,751 significant CpG sites and 119 differentially methylated regions from 1,030 brain samples (from four cohorts). The analyses are thorough but I think could be improved with some minor adjustments. Most notably, there seems to be very high genomic inflation – the number of DMPs is extremely large for a complex trait EWAS – and I think sensitivity analyses and more conservative approaches are required to help determine the likely true positive signals.

Main comments:

- I don't think use of the FDR threshold for significance is the best approach here. The authors state that FDR is 'widely used in DNA methylation analysis' but this is not true in epigenome-wide association studies in which genome-wide significant thresholds are well established and employed (Saffari et al. Genetic Epidemiology 2018; Mansell et al. BMC Genomics 2019). While the authors are trying to avoid missing true positives by using a less stringent significance threshold, it seems likely that use of the FDR threshold introduces false positives - this is reflected in the extremely high number of significant CpGs identified (n=3,751 – presumably correlated sites?) and evidence of inflation. Use of a conventional genome-wide significance threshold or additional sensitivity analyses would be more informative. Please report genomic inflation factors.

- Linked to the point above, the use of analysis tools such as OSCA (Zhang et al. Genome Biology 2019) should be used to account for CpG correlation structure and hidden confounders that might drive inflation. This would allow for a more accurate identification of the main signals and aid in distinguishing the true- from the false-positives.

- The analyses for overlap with genetic loci ("Correlation and enrichment with genetic susceptibility loci") are informal (as are the CAMTA1 analyses earlier in the paper). I think colocalization and Mendelian Randomisation approaches would offer greater insight.

Minor comments:

Introduction

- 'AD' is used throughout the first paragraph of the introduction but only introduced - 'Alzheimer's disease (AD)' - in the second paragraph

- 'the use of a disease relevant tissue is critical for informative epigenetic studies' – while it may be preferable to investigate methylation changes in the target tissue, use of peripheral and accessible tissue can still be informative

- 'brain tissues provide the ideal source for study samples' – the authors should make it clear that there are still issues with using post-mortem brain tissue in such studies eg. it typically represents the end stages of disease and it's still unclear if there are post-mortem changes in methylation patterns and how post-mortem intervals (which aren't corrected for here) may effect these

- 'the difference in corrected DNA methylation (%) between individuals with the lowest (score 0)

and highest (score VI) Braak score ranged from 1% to 5%' – are these differences relative to the mean and standard deviation of the probe? For example, a difference of 1% is much greater if the mean is 10% (SD 0.1%) than if it is 50% (SD 15%)

- The authors refer to methylation 'changes' (here and throughout the manuscript) – this should be changed to 'differences'

- 'in addition to corroborating previous findings...' – it would help to clarify which findings are from basic re-analysis of the same data and which are novel

- 'suggesting their potential value as AD biomarkers' – this is quite a bold statement and would need justification

Results

- Study cohort characteristics

o 'women' should be changed to 'female'

- Meta-analysis identified previously reported as well as novel DNA methylation changes significantly associated with AD Braak stage at individual CpGs and co-methylated genomic regions

- o 'Adjusting for [estimated] cell type proportions' – authors should also make it clear that it was the proportion of neurons that was controlled for

- o 'Therefore, methylation changes at individual CpGs and DMRs are located at different regions in the genome complementing each other, which provide a more complete picture of the Braak associated methylome.' – this sentence is unclear

- o Rather than removing probes associated with smoking it would be preferable to re-run the analysis with smoking as a covariate

- o 'among which 24% are located in non-coding regions' – how does this compare to the representation of non-coding regions on the arrays themselves?

- o 'Next, we combined the cohort specific p-values for these genomic regions using meta-analysis' – I know this is mentioned in the methods but I would restate here that this is an inverse-variance fixed effects meta

- o 'which provide a more complete picture of the Braak associated methylome [in the prefrontal cortex]'

- o 'Interestingly, this gene is observed to be expressed mainly in neurons' – needs a reference

- o 'In addition to corroborating previous findings, our meta-analysis also uncovered a number of novel differentially methylated genes' – the 'corroboration of previous findings' should be rephrased to reflect that results are from the same data

- Enrichment analysis of significant DNA methylation changes highlighted immune-related processes and Polycomb Repressive Complex 2 (PRC2) in AD

- o I think this section would benefit from a more general introduction to the biological features being investigated, particularly for a general journal such as Nature Comms.

- Prioritizing significant DNA methylation changes with sample matching

- o 'too few age-matched samples' – this should be quantified – how many is too few?

o 'note that the significant findings in this small sample analysis are still valid' – what is meant by 'valid'?

o 'Among them, 75% (n = 218) CpGs and 44% (n = 17) genomic regions overlapped with the significant CpGs and DMRs in our main analysis described above' – Was there any attenuation in the effect sizes?

- Correlation of AD associated methylation changes in blood and brain samples

o Authors refer to 'adjusted correlation analysis' – it would be beneficial to include information about the covariates included in this analysis here and a signpost to the relevant methods section

- Correlation of AD associated methylation changes with expressions of nearby genes

o 'we first removed age, sex...' – this would be clearer if it read 'adjusted for' rather than 'removed'

Discussion

- 'Our study illustrates the power of meta-analysis for epigenome-wide association studies' – it would be helpful if the authors added how the study illustrates this

- inconsistent use of PRC2/polycomb repressive complex 2

- unclear what is meant by 'substantial cross-talk has been observed between PRC2 and DNA methylation, two key epigenetics mechanisms for gene repression'. Again some more general comments about PRC2 would be very helpful here

- 'Plasticity nature' – plastic nature?

- use of 'robust' in the first sentence of the final paragraph is perhaps not warranted seeing as a lenient significance threshold was used

- 'a new computational framework' – this is a bit of an overstatement - the tools used for analysis are pre-existing so don't represent a new computational framework

- 'Facilitating development of effective therapeutic strategies for AD' – again this is a bold statement – how will it do this?

Methods

- Study cohorts

o '(late stage AD)' – it should be noted this is late stage tau pathology – there is no information about symptomatology or amyloid staging

- Pre-processing of DNA methylation Data -

o misspelling of Illumina in the first sentence

- Meta-analysis

o Post-mortem interval should also be considered as a covariate in analyses

o 'Fixed effects model (also referred to as inverse variance-weighted) meta-analysis was applied to synthesize statistical significance from individual cohorts' – I think you can simplify this to "An inverse variance-weighted fixed effects meta-analysis.."

o 'More specifically, we first matched each case with a control sample using matchControls function in e1071 R package' – more detail on how this package performs the matches and how

closely cases and controls were matched would be beneficial

- Functional annotation of significant methylation changes

o 'The enrichment of significant CpGs, or DMS (differentially methylated sites)...' – does this account for the correlation structure among CpGs?

- Correlation and overlap of significant methylation changes with genetic susceptibility loci

o 'We then fit the linear model median (Methylation_resid) ~ SNP to test association between methylation residuals in DMRs and CpGs described above and the imputed allele dosages for SNPs to identify mQTLs.' – There does not appear to be any adjustments for population stratification here (genetic PCs). I think these analyses should be rerun with PC adjustments.

Reviewer #3 (Remarks to the Author):

In this manuscript, Zhang et al. conducted a meta-analysis using data from four cohorts with more than one thousand prefrontal cortex brain samples to examine DNA methylation changes in association with Braak stage. The authors did very extensive analyses. I have the following comments:

1. Throughout the manuscript, I think the authors need to differentiate between AD dementia, pathological AD, Braak and cases. They are different concepts, and seem to be used interchangeably. It is confusing sometimes. Please check the whole manuscript and make changes.
2. Why did the authors only consider Braak? Why not consider other phenotypes such as CERAD?
3. As the authors analyzed brain samples as well as blood sample, it is better to be specific about age. Please point out whether age means age at death or age at blood draw. Please check the main text and the tables.
4. For the meta-analysis, I think it is better to use 'fixed-effect' and 'random-effects'. Please check your results section. Also, why don't you just use random-effects analysis, no matter whether there is heterogeneity or not because you lose little if there is no heterogeneity.
5. More information is needed about the blood sample of the London cohort. For example, when was the blood drew? How long was it from the blood draw to death?
6. A relevant comment is that the relationship between methylation in blood and brain has already been studied using data from ROSMAP (Yu et al., Methylation profiles in peripheral blood CD4+ lymphocytes versus brain: the relation to Alzheimer's disease pathology, 2016). Can the author compare the findings?
7. Another question I have is why the authors didn't exclude CpGs that are associated with smoking or overlapping with cross-reactive probes before the analysis? How did you determine smoking-related CpGs?
8. It seems that the authors did not consider postmortem interval (PMI) in the analysis. PMI can affect DNA methylation.
9. In the section 'Prioritizing significant DNA methylation changes with sample matching', the authors matched based on age (I think they mean age at death). Is this one-to-one matching? What if there are multiple matches for a person with a specific age at death in the case group? The authors need to be specific about how they handled matching. Also, matching based solely on age (at death) does not seem to be sufficient. I think sex and education should be at least considered.
10. I think some of the results should not be included in the results section. They are more appropriate for the discussions section. For example, from paragraph 'Our meta-analysis of individual CpGs identified many of the loci...' to the paragraph 'In addition to corroborating previous findings, ...' Please consider revision.
11. The paragraph before the discussion section. More information about genetic data from ROSMAP is needed. For example, are the genetic data imputed data? If so, were they imputed based on 1K genome or HRC? There are two batches of genetic data from ROSMAP. Did you use

both batches? If so, you should control for the batch effect (of the genetic data). Also, did you control for any principal components?

12. For many tables, more tables notes are needed. For example, what does the '++++' and '----' symbol mean in the column 'direction'?

13. There are occasionally grammar issues throughout the manuscript. Please carefully check the whole manuscript.

REVIEWER COMMENTS

Reviewer #1 (Remarks to the Author):

Zhang et al. examined epigenome-wide associations with Braak stages in several brain tissue samples. They identified a range of CpG sites as well as DMRs within a meta-analytical framework. The authors followed up on these to assess the relationship with gene expression and blood-based methylation patterns.

The paper is very well written and the analyses thorough. This manuscript is actually (!) interesting with noteworthy findings and excellent method descriptions. I also very much appreciate that the authors published their analysis protocols on github.

We thank this reviewer for the enthusiasm and encouragement for our study. We have made substantial changes in response to this reviewer's helpful comments, which we will discuss in detail below.

I have the following comments:

1) The authors identified a rather large amount of individual CpGs. I wonder if the authors observed an inflation of test results, but I couldn't find a QQ plot / lambda. Please add and discuss, if indeed there is evidence of inflation.

We appreciate this reviewer's concern about inflation of the test statistics. In response, we studied recent literature on genomic inflation factor. We found it was recently shown that the genomic inflation factor (lambda) is not suitable to measure inflation in epigenome-wide association studies (EWAS), because it often overestimates the amount of true inflation (see details in Iterson et al. (2017); PMID: 28129774). As shown by simulation studies (PMID: 28129774), real datasets (PMID: 28129774) as well as theory (PMID: 11315092), the genomic inflation factor is dependent on the expected number of true associations. Because in a typically EWAS, it is expected that many CpGs have small effects and might be truly associated with the phenotype, the genomic inflation factor would overestimate actual test-statistic inflation. To estimate genomic inflation more accurately in EWAS, Iterson et al. (2017) (PMID: 28129774) developed a Bayesian method that estimates bias and inflation in EWAS based on empirical null distributions, which is implemented in the Bioconductor package *bacon*.

In response to this reviewer's comment, we have now added Supplementary Figure 1 which includes Quantile-quantile (QQ) plots as well as genomic inflation factors estimated by both conventional and the *bacon* approach. Our results showed that the estimated genomic inflation factors for individual cohorts were modest and comparable to other recent EWAS profiling brain tissues (see for example Supp. Figure 10 in Viana et al. (2017) PMID: 28011714 shows lambdas for different brain regions: A) prefrontal cortex ($\lambda = 1.18$), B) striatum ($\lambda = 1.02$), C) hippocampus ($\lambda = 1.13$), and D) cerebellum ($\lambda = 1.23$)). In our study, lambdas by conventional approach ranged from 1.002 to 1.249, and lambdas based on *bacon* approach ranged from 0.987 to 1.082 (See items (1) below). In particular, for the ROSMAP cohort with the largest sample size, genomic inflation factors were close to 1 by both approaches ($\lambda = 1.016$, λ . *bacon* = 1.002). The genomic inflation factors for meta-analysis were $\lambda = 1.264$ and λ . *bacon* = 1.086.

In addition, we also performed sensitivity analysis using the 2,767 FDR significant CpGs after *bacon* correction. Our results showed the functional enrichment results were largely congruent with our previous results. In particular, using *bacon*-corrected FDR significant CpGs, we still observed significant enrichment in polycomb repressed regions ($P = 4.53 \times 10^{-65}$, OR = 4.37, Supplementary Table 17,

Supplementary Figure 2) as well as binding sites of polycomb repressive complex 2 subunits EZH2 ($P = 9.27 \times 10^{-16}$, OR = 1.90, Supplementary Table 18) and SUZ12 ($P = 9.44 \times 10^{-8}$, OR = 2.26, Supplementary Table 18). In addition, pathway analysis also showed significant enrichment in many immune system related pathways (e.g. immune system process, $P = 4.14 \times 10^{-11}$, FDR = 1.97×10^{-7} , Supplementary Table 19). We have added discussions of these additional results to the Discussion section. Please see items (1) – (5) below.

Revisions

(1) Supplementary Figure 1 Quantile-quantile (QQ) plots of observed and expected distributions of p-values in Gasparoni, London, Mount Sinai, and ROSMAP cohorts. λ is the genomic inflation factor, λ_{bacon} is genomic inflation factor estimated using the method of Iterson et al. (2017) (PMID: 28129774), as implemented in bacon R package. Shading indicates 95% confidence intervals.

Sensitivity analysis using FDR significant CpGs after bacon correction

(2) Supplementary Figure 2B – Enrichment of chromatin state for bacon corrected significant CpGs

B Single cpG meta-analysis probes
ChroM3MM: E073 – 15 coreMarks segments

Compare to Figure 3c - Enrichment of chromatin state using FDR significant CpGs (presented in initial version of manuscript)

C Single cpG meta-analysis probes
ChroM3MM: E073 – 15 coreMarks segments

(3) Supplementary Table 17 – Enrichment of 2,767 bacon-corrected FDR significant CpGs from meta-analysis in different chromatin states.

hyper-methylated differences in AD	After bacon correction			
	fdr significant CpGs		genome-wide significant CpGs	
	p-value	OR	p-value	OR
Active TSS	2.58E-01	0.93	2.72E-01	1.17
Bivalent Enhancer	5.60E-05	2.37	1.00E+00	0.93
Bivalent/Poised TSS	2.67E-02	1.37	9.12E-02	1.65
Enhancers	6.67E-05	1.50	2.77E-01	0.69
Flanking Active TSS	2.46E-61	3.91	4.11E-18	4.90
Flanking Bivalent TSS/Enh	4.17E-01	1.22	3.23E-01	1.61
Genic enhancers	3.59E-03	1.84	1.00E+00	0.75
Heterochromatin	2.66E-03	0.00	6.37E-01	0.00
Quiescent/Low	2.05E-63	0.23	2.38E-25	0.03
Repressed PolyComb	4.53E-65	4.37	8.52E-32	7.84
Strong transcription	3.51E-09	0.41	6.33E-03	0.34
Transcr. at gene 5' and 3'	1.34E-05	4.46	1.66E-02	5.73
Weak Repressed PolyComb	3.13E-05	0.64	2.63E-05	0.24
Weak transcription	2.14E-09	0.59	3.58E-05	0.37
ZNF genes & repeats	8.30E-02	0.00	1.00E+00	0.00

Compare to Supplementary Table 4 – enrichment of 3,751 FDR significant CpGs from previous submission

Supplementary Table 4 Enrichment of 119 FDR significant DMRs and 3751 FDR significant CpGs from meta-analysis in different chromatin states.

hyper-methylated changes in AD	DMR		CpG	
	p-value	OR	p-value	OR
	Active TSS	8.87E-16	0.49	2.49E-02
Bivalent Enhancer	2.04E-01	1.55	2.71E-06	2.29
Bivalent/Poised TSS	2.49E-01	0.74	1.58E-01	1.18
Enhancers	4.74E-03	1.76	2.33E-09	1.62
Flanking Active TSS	5.36E-16	2.94	1.71E-88	3.84
Flanking Bivalent TSS/Enh	6.08E-03	0.13	9.96E-02	1.37
Genic enhancers	8.11E-01	1.09	2.66E-03	1.71
Heterochromatin	4.34E-01	0.00	7.52E-04	0.10
Quiescent/Low	5.00E-09	0.31	3.02E-72	0.30
Repressed PolyComb	2.32E-70	7.56	1.29E-66	3.60
Strong transcription	3.08E-10	0.07	1.59E-10	0.47
Transcr. at gene 5' and 3'	7.42E-07	8.26	4.73E-06	3.85
Weak Repressed PolyComb	1.94E-03	0.53	3.84E-03	0.79
Weak transcription	5.07E-02	0.70	2.56E-13	0.59
ZNF genes & repeats	6.43E-01	0.00	1.77E-02	0.00

(4) Supplementary Table 18 After bacon correction, LOLA analysis of 2,767 FDR significant CpGs from meta-analysis

Antibody	CellType	Filename	OR	pValue	FDR
IKZF1_(iKN)_ (UCLA)	GM12878	wgEncodeAwgTfbsS	4.28	6.66E-21	2.01E-18
BCL11A	GM12878	wgEncodeAwgTfbsH	3.15	2.85E-18	7.81E-16
EZH2_(39875)	H1-hESC	wgEncodeAwgTfbsB	1.90	9.27E-16	1.55E-13
PU.1	GM12891	wgEncodeAwgTfbsH	2.01	1.57E-14	2.15E-12
p300_(SC-584)	GM12878	wgEncodeAwgTfbsS	4.59	7.75E-14	9.92E-12
BATF	GM12878	wgEncodeAwgTfbsH	2.55	1.41E-12	1.42E-10
PU.1	GM12878	wgEncodeAwgTfbsH	1.92	1.99E-12	1.94E-10
c-Fos	MCF10A-Er-Src	wgEncodeAwgTfbsS	2.03	6.67E-11	5.05E-09
eGFP-FOS	K562	wgEncodeAwgTfbsU	2.80	1.02E-10	7.13E-09
c-Fos	MCF10A-Er-Src	wgEncodeAwgTfbsS	1.89	5.48E-10	3.59E-08
EBF1_(SC-137065)	GM12878	wgEncodeAwgTfbsS	1.87	6.29E-10	4.03E-08
p300_(SC-584)	HeLa-S3	wgEncodeAwgTfbsS	2.39	1.96E-09	1.18E-07
SUZ12	H1-hESC	wgEncodeAwgTfbsS	2.26	9.44E-08	4.99E-06
EZH2_(39875)	NH-A	wgEncodeAwgTfbsB	1.55	3.68E-07	1.85E-05
SUZ12	NT2-D1	wgEncodeAwgTfbsS	1.77	4.61E-07	2.17E-05
p300	GM12878	wgEncodeAwgTfbsH	2.20	1.13E-06	4.81E-05
GR	ECC-1	wgEncodeAwgTfbsH	2.36	1.19E-06	4.99E-05
EBF1_(SC-137065)	GM12878	wgEncodeAwgTfbsH	1.53	2.13E-06	8.43E-05
CtBP2	H1-hESC	wgEncodeAwgTfbsS	1.49	3.30E-06	1.27E-04
GR	A549	wgEncodeAwgTfbsH	1.90	3.49E-06	1.33E-04

Compare to Supplementary Table 5 from previous submission – enrichment of 3,751 FDR significant CpGs

Supplementary Table 5 Enrichment of FDR significant CpGs in binding sites of ENCODE transcription factors and chromatin proteins.

Antibody	CellType	Description	Filename	OR	pValue	FDR
EZH2_(39875)	H1-hESC	CHIP H1-hESC EZH2_(39875)	wgEncodeAwgTfbs	1.89	3.41E-20	7.34E-18
IKZF1_(IKN)_UCLA	GM12878	CHIP GM12878 IKZF1_(IKN)_	wgEncodeAwgTfbs	3.48	2.99E-18	5.64E-16
BCL11A	GM12878	CHIP GM12878 BCL11A	wgEncodeAwgTfbs	2.74	2.37E-17	3.57E-15
p300_(SC-584)	GM12878	CHIP GM12878 p300_(SC-58	wgEncodeAwgTfbs	4.39	1.38E-16	1.89E-14
PU.1	GM12891	CHIP GM12891 PU.1	wgEncodeAwgTfbs	1.87	5.23E-15	6.06E-13
PU.1	GM12878	CHIP GM12878 PU.1	wgEncodeAwgTfbs	1.77	2.02E-12	1.65E-10
SUZ12	H1-hESC	CHIP H1-hESC SUZ12	wgEncodeAwgTfbs	2.40	1.23E-11	9.01E-10
BATF	GM12878	CHIP GM12878 BATF	wgEncodeAwgTfbs	2.22	2.26E-11	1.58E-09
c-Fos	MCF10A-Er-Src	CHIP MCF10A-Er-Src c-Fos	wgEncodeAwgTfbs	1.87	5.96E-11	3.90E-09
c-Fos	MCF10A-Er-Src	CHIP MCF10A-Er-Src c-Fos	wgEncodeAwgTfbs	1.77	2.53E-10	1.60E-08
EBF1_(SC-137065)	GM12878	CHIP GM12878 EBF1_(SC-13	wgEncodeAwgTfbs	1.75	4.24E-10	2.55E-08
p300_(SC-584)	HeLa-S3	CHIP HeLa-S3 p300_(SC-584)	wgEncodeAwgTfbs	2.22	5.20E-10	3.07E-08
eGFP-FOS	K562	CHIP K562 eGFP-FOS	wgEncodeAwgTfbs	2.37	2.75E-09	1.51E-07
EZH2_(39875)	NH-A	CHIP NH-A EZH2_(39875)	wgEncodeAwgTfbs	1.54	8.35E-09	4.41E-07
GR	A549	CHIP A549 GR	wgEncodeAwgTfbs	1.91	5.02E-08	2.48E-06
GR	ECC-1	CHIP ECC-1 GR	wgEncodeAwgTfbs	2.21	3.60E-07	1.49E-05
GATA-2	HUVEC	CHIP HUVEC GATA-2	wgEncodeAwgTfbs	1.74	4.20E-07	1.71E-05
SUZ12	NT2-D1	CHIP NT2-D1 SUZ12	wgEncodeAwgTfbs	1.65	6.12E-07	2.46E-05
c-Fos	HUVEC	CHIP HUVEC c-Fos	wgEncodeAwgTfbs	1.49	1.50E-06	5.46E-05

(5) Supplementary Table 19 Top 20 most significant pathways associated with AD Braak stage identified in meta-analysis. Significant single CpGs were selected as FDR significant CpGs after bacon correction.

	Description	P-value	FDR
GO:0045321	leukocyte activation	7.91E-15	1.73E-10
GO:0001775	cell activation	1.53E-14	1.73E-10
GO:0007159	leukocyte cell-cell adhesion	1.63E-11	1.23E-07
GO:0002376	immune system process	4.14E-11	1.97E-07
GO:0022409	positive regulation of cell-cell adhesion	4.89E-11	1.97E-07
GO:0046649	lymphocyte activation	5.23E-11	1.97E-07
GO:1903037	regulation of leukocyte cell-cell adhesion	7.53E-11	2.43E-07
GO:0042110	T cell activation	2.97E-10	8.37E-07
GO:0022407	regulation of cell-cell adhesion	4.40E-10	1.10E-06
GO:0006955	immune response	4.94E-10	1.12E-06
GO:0002694	regulation of leukocyte activation	5.83E-10	1.20E-06
GO:0050865	regulation of cell activation	1.01E-09	1.89E-06
GO:0002682	regulation of immune system process	1.25E-09	2.17E-06
GO:0002521	leukocyte differentiation	1.93E-09	3.11E-06
GO:0051249	regulation of lymphocyte activation	2.38E-09	3.58E-06
GO:1903039	positive regulation of leukocyte cell-cell adhesion	3.24E-09	4.35E-06
GO:0002252	immune effector process	3.28E-09	4.35E-06
GO:0050867	positive regulation of cell activation	7.24E-09	9.09E-06
GO:0006954	inflammatory response	1.19E-08	1.41E-05
GO:0050863	regulation of T cell activation	1.54E-08	1.73E-05

(6) In Results – under “Genomic inflation and sensitivity analysis”

To assess the potential inflation in our results, we estimated genomic inflation factors using both the conventional and the *bacon* method¹, specifically proposed for EWAS. As shown by simulation studies¹, real datasets¹, and theory², the conventional genomic inflation factor (lambda or λ used interchangeably below) is dependent on the expected number of true associations. Because in a typical EWAS it is expected that small effects from many CpGs might be associated with the phenotype, the genomic inflation factor would overestimate actual test-statistic inflation. To estimate genomic inflations more accurately in EWAS, Iterson et al. (2017)¹ developed a Bayesian method that estimates inflation in EWAS based on empirical null distributions, which is implemented in the Bioconductor package *bacon*. The estimated genomic

inflation factors for individual cohorts in our study were modest and comparable to other recent EWAS profiling brain tissues (for example, Supplementary Fig. 10 in Viana et al. (2017)³ showed that lambdas for different brain regions ranged from 1.02 to 1.23). In our study, lambdas (λ) by conventional approach ranged from 1.002 to 1.249, and lambdas based on the *bacon* approach ($\lambda.bacon$) ranged from 0.986 to 1.082 (Supplementary Fig. 1). In particular, for the ROSMAP cohort with the largest sample size, genomic inflation factors were close to 1 by both approaches ($\lambda = 1.016, \lambda.bacon = 1.002$). Genomic inflation factors for the meta-analysis were $\lambda = 1.264$ and $\lambda.bacon = 1.086$.

In addition, we conducted sensitivity analyses to evaluate the impact of inflation on our enrichment analysis results. To this end, we performed inflation correction for single cohort effect sizes and standard errors, and then meta-analyzed the *bacon*-corrected effect sizes and standard errors. Enrichment analysis was then conducted for the 2,767 CpGs that reached FDR significance and the 339 CpGs that reached the 2.4×10^{-7} genome-wide significance level⁴ (Supplementary Table 15). Our results showed that the functional enrichment results in these sensitivity analyses were largely congruent with our main analysis results (Supplementary Fig. 2-3, Supplementary Tables 16-18). In particular, we still observed significant enrichment in polycomb repressed regions for hyper-methylated CpGs in AD, as well as in binding sites of polycomb repressive complex 2 subunits EZH2 and SUZ12. Moreover, pathway analysis still showed significant enrichment in immune system related pathways (Supplementary Table 19-20) for both FDR significant and genome-wide significant CpGs after *bacon* correction.

2) I found the overlap results between individual sites and DMR really interesting. However, I'm not sure I agree with the authors that these analyses identified "complementary regions". Isn't it rather the case that the DMRs are a subset of the individual sites? I.e. the authors identified 3k individual sites, of which a subset extended into regions? (and related to this, what was the average number of CpGs per DMR?)

This reviewer raised an important point. To clarify, in the DMR analysis, a less stringent criteria was used to select CpGs (10^{-3} in *comb-p* software), a DMR is declared as long as the consecutive CpGs in the region has a small joint p-value, therefore the significant DMRs also include some CpGs not reaching FDR significance in the analysis of individual CpGs. The figure on the right shows the overlap between CpGs in DMRs and individual CpGs analysis. "Single CpG analysis probes" refers to the 3,751 CpGs that reached FDR significance in single CpGs analysis and "DMR analysis probes" refer to the 742 probes that are included in the 119 FDR significant DMRs.

In response to this reviewer's comment, we have now revised the description of enrichment analysis to clarify that enrichment of significant individual CpGs and CpGs within DMRs are at different genomic locations in the genome. In addition, we've also clarified parameter setting for *comb-p* DMR analysis to indicate an individual CpG p-value of 10^{-3} was used to start a region and the region is extended if another p-value is within 200 base pairs. Also, we added to the Results section that the average number of CpGs per DMR is 5.16.

Revision

(1) In Results section, under section “Enrichment analysis of significant DNA methylation differences in pathological AD highlights immune-related processes and polycomb repressive complex 2 (PRC2)”, first paragraph

Interestingly, hyper-methylation in AD at individual CpGs and DMRs was enriched in different features of genes across the genome (Fig. 3a-b, Supplementary Table 3).

(2) In Results section, 4th paragraph

The coMethDMR and comb-p based meta-analysis approaches identified 478 and 187 significant DMRs associated with AD Braak stage, respectively, with 143 being identified by both methods. After eliminating those DMRs containing cross-reactive probes⁵ or smoking-associated probes⁶, we obtained 119 co-methylated DMRs at 5% FDR (Supplementary Table 2). The average number of CpGs per DMR is 5.16 ± 2.80 CpGs. Notably, 118 out of the 119 DMRs included FDR significant individual CpGs. On the other hand, only 421 out of the 3,751 FDR significant individual CpGs overlapped with the FDR significant DMRs. Therefore, methylation differences at individual CpGs and DMRs did not completely overlap, so analyzing both individual CpGs and DMRs provided a more complete picture of the Braak-associated methylome in the prefrontal cortex. Our final set of DNA methylation differences included these 3,751 individual CpGs along with the 119 DMRs (Fig. 2).

(3) In Methods section, section “Meta-analysis”, third paragraph

We added the following sentence: “We used the default setting for our comb-p analysis, with parameters --seed 1e-3 and --dist 200, which required a P-value of 10^{-3} to start a region and extend the region if another P-value was within 200 base pairs.”

3) I really liked the alternative approach of matching by age, in addition to controlling for age. It wasn't very clear if samples were matched within or across cohorts? I.e. is it possible that differences in the age-matched analysis merely reflect cohort effects (if matched across cohorts)?

To clarify, we matched samples within each cohort. In response, we've added additional clarification on this.

Revision – In Results, under section “Prioritizing significant DNA methylation differences with sample matching”

To prioritize significant methylation differences in pathological AD and minimize confounding effects due to aging, we also explored an alternative strategy by matching each pathological AD case with a control subject of the same sex and age at death in the same cohort.

4) I'm not sure the conclusion that “significant findings in this small sample analysis are still valid”. Reduced power can also increase the number of false positives. Rephrase?

In response to this reviewer's comment, we removed this sentence from the section “Prioritizing significant DNA methylation changes with sample matching”.

5) The authors say that 75% and 44% of the age-matched results matched those of the main analysis in terms of significance. Is the direction of effect also consistent? Please add.

We thank this reviewer for this good suggestion. As another reviewer suggested, we re-ran the matched samples analysis by matching each AD case with a control subject with both the same sex and age at death within the same cohort. For the results, we identified 151 significant CpGs and 32 significant DMR with less than 5% *fdr*. Among them, 85% (*n* = 129) CpGs and 50% (*n* = 16) genomic regions overlapped with the significant CpGs and DMRs in our main analysis with the same direction of changes.

In response, we added clarifications to the manuscript to indicate the overlap between main analysis and matched analysis considered directions of the effects.

Revision – under “Prioritizing significant DNA methylation differences with sample matching”

Among them, 85% (*n* = 129) of CpGs and 50% (*n* = 16) of genomic regions overlapped with the significant CpGs and DMRs in our main analysis described above **with the same direction of change**.

6) Re blood-brain correlation results: the authors report for example 39 of 3,751 probes showing strong correlations. Isn't that just even lower than the 5% level you would expect to observe by chance? I.e. evidence for cross-tissue correlation wasn't actually that convincing overall?

We agree with this reviewer that cross-tissue correlation wasn't convincing. We wish to clarify that although overall methylation levels did not correlate well between brain and blood, we did observe moderate to strong correlation at several individual CpGs in both London cohort (PMID: 25129077) and Edger et al. cohort (PMID: 28763057), which were all positive (Supplementary Table 8). We also mentioned in the Discussion section that “Future studies are needed to additionally evaluate and validate these potential biomarkers before their adoption in the clinics”.

In response to this reviewer's comment, we clarified in Results section that overall methylation levels in brain and blood did not correlate well.

Revision – in section “Correlation of AD-associated CpGs and DMRs methylation levels in blood and brain samples”

We performed both an adjusted correlation analysis based on methylation residuals (r_{resid}), in which we adjusted estimated neuron proportions for brain samples (or estimated blood cell-type proportions), array, age at death (for brain samples) or at blood draw (for blood samples), and sex, and an unadjusted correlation analysis based on beta values (r_{beta}) (Online Methods). The correlation between methylation levels in brain and blood were modest at the majority of CpG sites (mean Pearson $r = 0.069$, $SD = 0.165$), which is similar to those reported in Yu et al. (2016)⁷ for CD4+ lymphocytes and other previous reports^{7, 8}. Among CpGs mapped within the 119 significant DMRs (*n* = 728), only 11 showed moderate to strong association in brain and blood in both adjusted and unadjusted analyses (absolute $r_{beta} \geq 0.5$, $FDR_{beta} < 0.05$, absolute $r_{resid} \geq 0.5$, $FDR_{resid} < 0.05$). Similarly, among the 3,751 significant individual CpGs, only 39 showed moderate to strong associations (absolute $r_{beta} \geq 0.5$, $FDR_{beta} < 0.05$, absolute $r_{resid} \geq 0.5$, $FDR_{resid} < 0.05$) in brain and blood. Remarkably, all 50 CpGs showed significant positive correlations, corroborating previous analyses⁸ that also observed a significant negative correlation between brain and blood is relatively rare.

7) Re methylation – expression results: the authors argue that this was done separately in cases and controls as this association might be different in these groups. However, wouldn't a more sensible approach be a regression with group as a covariate?

We thank this reviewer for the helpful suggestion. As this reviewer suggested, we have re-analyzed methylation – gene expression data by including all samples and adjusting for Braak stage, which is similar to adjusting for case-control groups. We chose to adjust for Braak stage to leverage more information (actual Braak stage as opposed to cases vs. controls) in the samples.

Revision – updated Supplementary 9 and 10

(1) Supplementary Table 9 Association between methylation levels at Braak-associated DMRs with nearby genes. Each DMR is linked to genes located in the vicinity (± 250 kb). First, each DMR is summarized by the median of CpG methylation M-values over all CpGs mapped within the DMR. The median methylation M-values and normalized gene expression values are then adjusted for age at death, sex, cell type, and batch effects separately. Next, the residuals from these linear models are extracted. Finally, a separate linear model is used to test association between methylation residuals and gene expression residuals, adjusting for Braak stage. Shown are: the name of the most significant gene associated with each DMR (Gene), effect size (Estimate), p-value (pVal), and false discovery rate (FDR) for the regression model correlating methylation residuals and gene expression residuals. Annotations include the type of associated genomic feature (UCSC_RefGene_Group), location with respect to CpG islands (Relation_to_Island), and chromatin state (Chromatin state).

(2) Supplementary Table 10 Association between methylation levels at Braak-associated CpGs with nearby genes. Each CpG is linked to genes located in the vicinity (± 250 kb from the start or end of the DMR). The methylation M-values and normalized gene expression values are first adjusted for age at death, sex, cell type, and batch effects separately. Next, the residuals from these linear models are extracted. Finally, a separate linear model is used to test the association between methylation residuals and gene expression residuals, adjusting for Braak stage. Shown are: the name of the most significant gene associated with each CpG (Gene), effect size (Estimate), p-value (pVal), and false discovery rate (FDR) for the regression model correlating methylation residuals and gene expression residuals. Annotations include the type of associated genomic feature (UCSC_RefGene_Group), location with respect to CpG islands (Relation_to_Island), and chromatin state (Chromatin state).

8) Did I read this correctly that in the methylation analysis, Braak stage was included as an ordinal/continuous variable, but dichotomized in the gene expression analysis? Why was that?

In response, we have re-analyzed methylation – gene expression data by including both cases and controls and adjusting for Braak stage as this reviewer suggested.

Revision – updated Supplementary 9 and 10

(1) Supplementary Table 9 Association between methylation levels at Braak-associated DMRs with nearby genes. Each DMR is linked to genes located in the vicinity (± 250 kb). First, each DMR is summarized by the median of CpG methylation M-values over all CpGs mapped within the DMR. The median methylation M-values and normalized gene expression values are then adjusted for age at death, sex, cell type, and batch effects separately. Next, the residuals from these linear models are extracted. Finally, a separate linear

model is used to test association between methylation residuals and gene expression residuals, adjusting for Braak stage. Shown are: the name of the most significant gene associated with each DMR (Gene), effect size (Estimate), p-value (pVal), and false discovery rate (FDR) for the regression model correlating methylation residuals and gene expression residuals. Annotations include the type of associated genomic feature (UCSC_RefGene_Group), location with respect to CpG islands (Relation_to_Island), and chromatin state (Chromatin state).

(2) Supplementary Table 10 Association between methylation levels at Braak-associated CpGs with nearby genes. Each CpG is linked to genes located in the vicinity (± 250 kb from the start or end of the DMR). The methylation M-values and normalized gene expression values are first adjusted for age at death, sex, cell type, and batch effects separately. Next, the residuals from these linear models are extracted. Finally, a separate linear model is used to test the association between methylation residuals and gene expression residuals, adjusting for Braak stage. Shown are: the name of the most significant gene associated with each CpG (Gene), effect size (Estimate), p-value (pVal), and false discovery rate (FDR) for the regression model correlating methylation residuals and gene expression residuals. Annotations include the type of associated genomic feature (UCSC_RefGene_Group), location with respect to CpG islands (Relation_to_Island), and chromatin state (Chromatin state).

Minor comments:

1) In the blood-brain correlation section, the authors report on “Rresid” and “Rbeta”. Are these correlation coefficients? I.e. should this be a lower-case “r”?

In response, we changed R_{resid} and R_{beta} to r_{resid} and r_{beta} .

Revision – In section “Correlation of AD-associated CpGs and DMRs methylation levels in blood and brain samples”

We performed both an adjusted correlation analysis based on methylation residuals (r_{resid}), in which we adjusted estimated neuron proportions for brain samples (or estimated blood cell-type proportions), array, age at death (for brain samples) or at blood draw (for blood samples), and sex, and an unadjusted correlation analysis based on beta values (r_{beta}) (Online Methods). The correlation between methylation levels in brain and blood were modest at the majority of CpG sites (mean Pearson $r = 0.069$, $SD = 0.165$), which is similar to those reported in Yu et al. (2016)⁷ for CD4+ lymphocytes and other previous reports^{7,8}. Among CpGs mapped within the 119 significant DMRs ($n = 728$), only 11 showed moderate to strong association in brain and blood in both adjusted and unadjusted analyses (absolute $r_{beta} \geq 0.5$, $FDR_{beta} < 0.05$, absolute $r_{resid} \geq 0.5$, $FDR_{resid} < 0.05$). Similarly, among the 3,751 significant individual CpGs, only 39 showed moderate to strong associations (absolute $r_{beta} \geq 0.5$, $FDR_{beta} < 0.05$, absolute $r_{resid} \geq 0.5$, $FDR_{resid} < 0.05$) in brain and blood. Remarkably, all 50 CpGs showed significant positive correlations, corroborating previous analyses⁸ that also observed a significant negative correlation between brain and blood is relatively rare.

2) The quality threshold of bisulfite conversion efficiency was set to 88%. That seems to be rather arbitrary. What was the justification for doing so?

To clarify, we initially considered using 90% bisulfite conversion rate to select samples, but using this criterion we would only have 15 samples (out of 60 samples) for the Gasparoni dataset. Therefore, we lowered the criteria a little to 88%, so we only lose 3 samples and still have 57 samples for the Gasparoni dataset.

3) The authors equate a fixed-effect meta-analysis with an inverse-variance weighted one. I'm not sure that's correct (e.g. random effect MA could also be IVW). Can the authors explain and/or provide a reference?

This reviewer had a good point. Another reviewer also had a similar comment and suggested we use "inverse-variance weighted fixed effects model" instead. We have updated the manuscript accordingly.

Revision

(1) In Methods, under section "Meta-analysis", 2nd paragraph

The **inverse-variance weighted fixed effects model** was applied to synthesize statistical significance from individual cohorts.

(2) In Results, 3rd paragraph

Next, we combined the cohort-specific P-values for these genomic regions using an **inverse-variance fixed effects meta-analysis**.

4) In the methods section: "More specifically, we first matched each case with a control sample using matchControls function in e1071 R" – please add what the pairs were match by. E.g. "age-"matched each case..

In response, we clarified the samples were matched by age at death and sex.

Revision – In Methods, under section "Meta-analysis", 5th paragraph

More specifically, we first matched each case with control samples **with the same age at death (in years) and sex in the same cohort** using the matchControls function in the e1071 R package. **When there are multiple matched control samples, the control sample with the most similar age as the case sample is selected.** The **age and sex** matched samples were then analyzed in the same way as described above, except for removing **age at death and sex** effects in the linear models.

References

1. van Iterson, M., van Zwet, E.W., Consortium, B. & Heijmans, B.T. Controlling bias and inflation in epigenome- and transcriptome-wide association studies using the empirical null distribution. *Genome Biol* **18**, 19 (2017).
2. Devlin, B. & Roeder, K. Genomic control for association studies. *Biometrics* **55**, 997-1004 (1999).
3. Viana, J. et al. Schizophrenia-associated methylomic variation: molecular signatures of disease and polygenic risk burden across multiple brain regions. *Hum Mol Genet* **26**, 210-225 (2017).
4. Saffari, A. et al. Estimation of a significance threshold for epigenome-wide association studies. *Genet Epidemiol* **42**, 20-33 (2018).
5. Chen, Y.A. et al. Discovery of cross-reactive probes and polymorphic CpGs in the Illumina Infinium HumanMethylation450 microarray. *Epigenetics* **8**, 203-209 (2013).
6. Joehanes, R. et al. Epigenetic Signatures of Cigarette Smoking. *Circ Cardiovasc Genet* **9**, 436-447 (2016).
7. Yu, L. et al. Methylation profiles in peripheral blood CD4+ lymphocytes versus brain: The relation to Alzheimer's disease pathology. *Alzheimer's & dementia : the journal of the Alzheimer's Association* **12**, 942-951 (2016).
8. Hannon, E., Lunnon, K., Schalkwyk, L. & Mill, J. Interindividual methylomic variation across blood, cortex, and cerebellum: implications for epigenetic studies of neurological and neuropsychiatric phenotypes. *Epigenetics* **10**, 1024-1032 (2015).

Reviewer #2 (Remarks to the Author):

This paper analyses multiple brain-based DNA methylation datasets to identify epigenetic correlates of Alzheimer's disease in the prefrontal cortex. The study applies multiple approaches (EWAS, region-based methylation analyses, mQTL overlap and enrichment), identifying 3,751 significant CpG sites and 119 differentially methylated regions from 1,030 brain samples (from four cohorts). The analyses are thorough but I think could be improved with some minor adjustments. Most notably, there seems to be very high genomic inflation – the number of DMPs is extremely large for a complex trait EWAS – and I think sensitivity analyses and more conservative approaches are required to help determine the likely true positive signals.

We thank this reviewer for the thorough review and encouragement for our study. We have made substantial changes in response to this reviewer's helpful comments, which we will discuss in detail below.

Main comments:

R2.1. I don't think use of the FDR threshold for significance is the best approach here. The authors state that FDR is 'widely used in DNA methylation analysis' but this is not true in epigenome-wide association studies in which genome-wide significant thresholds are well established and employed (Saffari et al. Genetic Epidemiology 2018; Mansell et al. BMC Genomics 2019).

While the authors are trying to avoid missing true positives by using a less stringent significance threshold, it seems likely that use of the FDR threshold introduces false positives - this is reflected in the extremely high number of significant CpGs identified (n=3,751 – presumably correlated sites?) and evidence of inflation. Use of a conventional genome-wide significance threshold or additional sensitivity analyses would be more informative. Please report genomic inflation factors.

We appreciate this reviewer's concern about false positives and agree that the more stringent genome-wide significance threshold would help guard against false positives. In response, we have now added an additional column in Supplementary Table 15 indicating CpG sites that reached the 2.4×10^{-7} genome-wide significance level, as determined by Saffari et al. (2018) (PMID: 29034560) for Illumina 450k arrays. In addition, as this reviewer suggested, we also performed sensitivity analysis using the 339 CpGs that reached genome-wide significance level (after *bacon* correction for inflation, see more details below). Our results showed the functional enrichment results for genome-wide significant CpGs were largely congruent with enrichment results based on FDR significant CpGs. In particular, using genome-wide significant CpGs, we still observed significant enrichment in polycomb repressed regions ($P = 8.52 \times 10^{-32}$, OR = 7.84, Supplementary Table 17, Supplementary Figure 3) and immune system related pathways (GO:0002376 immune system process, $P = 9.63 \times 10^{-7}$, FDR = 0.022, Supplementary Table 20). See additional details in items (1) – (3) below.

With regard to this reviewer's concern about possible inflation and genomic inflation factors, we studied recent literature on genomic inflation factor. We found it was recently shown that genomic inflation factor (λ) is not suitable to measure inflation in epigenome-wide association studies (EWAS), because it often overestimates the amount of true inflation (see details in Iterson et al. (2017); PMID: 28129774). As shown by simulation studies (PMID: 28129774), real datasets (PMID: 28129774) as well as theory (PMID: 11315092), genomic inflation factor is dependent on the expected number of true associations. Because in a typically EWAS, it is expected small effects from many CpGs might be associated with the phenotype, genomic inflation factor would overestimate actual test-statistic

inflation. To estimate genomic inflations more accurately in EWAS, Iterson et al. (2017) (PMID: 28129774) developed a Bayesian method that estimates bias and inflation in EWAS based on empirical null distributions, which is implemented in the Bioconductor package *bacon*.

In response to this reviewer’s comment, we have now added Supplementary Figure 1 which includes Quantile-quantile (QQ) plots as well as genomic inflation factors estimated by both conventional and the *bacon* approach. Our results showed that the estimated genomic inflation factors for individual cohorts were modest and comparable to other recent EWAS profiling brain tissues (see for example Supp. Figure 10 in Viana et al. (2017) PMID: 28011714 shows lambdas for different brain regions: A) prefrontal cortex ($\lambda = 1.18$), B) striatum ($\lambda = 1.02$), C) hippocampus ($\lambda = 1.13$), and D) cerebellum ($\lambda = 1.23$)). In our study, lambdas by conventional approach ranged from 1.002 to 1.249, and lambdas based on *bacon* approach ranged from 0.987 to 1.082 (See item (4) below). In particular, for the ROSMAP cohort with the largest sample size, genomic inflation factors were close to 1 by both approaches ($\lambda = 1.016$, $\lambda_{bacon} = 1.002$). Genomic inflation factor for meta-analysis were $\lambda = 1.264$ and $\lambda_{bacon} = 1.086$. We have added discussions of these additional results to Discussion section. Please see items (4) – (5) below.

Revisions

Sensitivity analysis using genome-wide significant CpGs

(1) Supplementary Figure 3B – Enrichment of chromatin state for genome-wide significant CpGs after *bacon* correction

Compare to Figure 3c - Enrichment of chromatin state using FDR significant CpGs (presented in initial version of manuscript)

(2) Supplementary Table 17 – Enrichment of 339 genome-wide significant CpGs from meta-analysis in different chromatin states.

hyper-methylated differences in AD	After bacon correction			
	fdr significant CpGs		genome-wide significant CpGs	
	p-value	OR	p-value	OR
Active TSS	2.58E-01	0.93	2.72E-01	1.17
Bivalent Enhancer	5.60E-05	2.37	1.00E+00	0.93
Bivalent/Poised TSS	2.67E-02	1.37	9.12E-02	1.65
Enhancers	6.67E-05	1.50	2.77E-01	0.69
Flanking Active TSS	2.46E-61	3.91	4.11E-18	4.90
Flanking Bivalent TSS/Enh	4.17E-01	1.22	3.23E-01	1.61
Genic enhancers	3.59E-03	1.84	1.00E+00	0.75
Heterochromatin	2.66E-03	0.00	6.37E-01	0.00
Quiescent/Low	2.05E-63	0.23	2.38E-25	0.03
Repressed PolyComb	4.53E-65	4.37	8.52E-32	7.84
Strong transcription	3.51E-09	0.41	6.33E-03	0.34
Transcr. at gene 5' and 3'	1.34E-05	4.46	1.66E-02	5.73
Weak Repressed PolyComb	3.13E-05	0.64	2.63E-05	0.24
Weak transcription	2.14E-09	0.59	3.58E-05	0.37
ZNF genes & repeats	8.30E-02	0.00	1.00E+00	0.00

Compare to Supplementary Table 4 – enrichment of 3,751 FDR significant CpGs from previous submission

hyper-methylated differences in AD

Feature	DMR		CpG	
	p-value	OR	p-value	OR
Active TSS	8.87E-16	0.49	2.49E-02	0.90
Bivalent Enhancer	2.04E-01	1.55	2.71E-06	2.29
Bivalent/Poised TSS	2.49E-01	0.74	1.58E-01	1.18
Enhancers	4.74E-03	1.76	2.33E-09	1.62
Flanking Active TSS	5.36E-16	2.94	1.71E-88	3.84
Flanking Bivalent TSS/Enh	6.08E-03	0.13	9.96E-02	1.37
Genic enhancers	8.11E-01	1.09	2.66E-03	1.71
Heterochromatin	4.34E-01	0.00	7.52E-04	0.10
Quiescent/Low	5.00E-09	0.31	3.02E-72	0.30
Repressed PolyComb	2.32E-70	7.56	1.29E-66	3.60
Strong transcription	3.08E-10	0.07	1.59E-10	0.47
Transcr. at gene 5' and 3'	7.42E-07	8.26	4.73E-06	3.85
Weak Repressed PolyComb	1.94E-03	0.53	3.84E-03	0.79
Weak transcription	5.07E-02	0.70	2.56E-13	0.59
ZNF genes & repeats	6.43E-01	0.00	1.77E-02	0.00

(3) Supplementary Table 20 - Top 20 most significant pathways associated with AD Braak stage identified in meta-analysis. Significant single CpGs were selected as genome-wide significant CpGs at with p-value less than 2.4×10^{-7} .

	Description	P-value	FDR
GO:0002376	immune system process	9.63E-07	0.022
GO:0048705	skeletal system morphogenesis	9.17E-06	0.070
GO:0048704	embryonic skeletal system morphogenesis	9.25E-06	0.070
GO:0001775	cell activation	1.61E-05	0.091
GO:0045321	leukocyte activation	2.39E-05	0.098
GO:0048706	embryonic skeletal system development	2.94E-05	0.098
GO:0070661	leukocyte proliferation	3.04E-05	0.098
GO:0050670	regulation of lymphocyte proliferation	4.04E-05	0.104
GO:0032944	regulation of mononuclear cell proliferation	4.16E-05	0.104
GO:0046651	lymphocyte proliferation	5.22E-05	0.107
GO:0032943	mononuclear cell proliferation	5.61E-05	0.107
GO:0070663	regulation of leukocyte proliferation	6.20E-05	0.107
GO:0002684	positive regulation of immune system process	6.43E-05	0.107
GO:0002682	regulation of immune system process	6.66E-05	0.107
GO:0000302	response to reactive oxygen species	7.68E-05	0.116
GO:0051897	positive regulation of protein kinase B signaling	9.95E-05	0.140
GO:0002262	myeloid cell homeostasis	1.15E-04	0.152
GO:0032609	interferon-gamma production	1.52E-04	0.191
GO:0002520	immune system development	1.76E-04	0.201
GO:0048534	hematopoietic or lymphoid organ development	1.98E-04	0.201

Genomic inflation factors

(4) Supplementary Figure 1 Quantile-quantile (QQ) plots of observed and expected distributions of p-values in Gasparoni, London, Mount Sinai, and ROSMAP cohorts. λ is genomic inflation factor, λ_{bacon} is genomic inflation factor estimated using the method of Iterson et al. (2017) (PMID: 28129774), as implemented in bacon R package. Shading indicates 95% confidence intervals.

(5) In Results – under “Genomic inflation and sensitivity analysis”

To assess the potential inflation in our results, we estimated genomic inflation factors using both the conventional and the *bacon* method¹, specifically proposed for EWAS. As shown by simulation studies¹, real datasets¹, and theory², the conventional genomic inflation factor (lambda or λ used interchangeably below) is dependent on the expected number of true associations. Because in a typical EWAS it is expected that small effects from many CpGs might be associated with the phenotype, the genomic inflation factor would overestimate actual test-statistic inflation. To estimate genomic inflations more accurately in EWAS, Iterson et al. (2017)¹ developed a Bayesian method that estimates inflation in EWAS based on empirical null distributions, which is implemented in the Bioconductor package *bacon*. The estimated genomic inflation factors for individual cohorts in our study were modest and comparable to other recent EWAS profiling brain tissues (for example, Supplementary Fig. 10 in Viana et al. (2017)³ showed that lambdas for different brain regions ranged from 1.02 to 1.23). In our study, lambdas (λ) by conventional approach ranged from 1.002 to 1.249, and lambdas based on the *bacon* approach (λ_{bacon}) ranged from 0.986 to

1.082 (Supplementary Fig. 1). In particular, for the ROSMAP cohort with the largest sample size, genomic inflation factors were close to 1 by both approaches ($\lambda = 1.016$, $\lambda_{bacon} = 1.002$). Genomic inflation factors for the meta-analysis were $\lambda = 1.264$ and $\lambda_{bacon} = 1.086$.

In addition, we conducted sensitivity analyses to evaluate the impact of inflation on our enrichment analysis results. To this end, we performed inflation correction for single cohort effect sizes and standard errors, and then meta-analyzed the *bacon*-corrected effect sizes and standard errors. Enrichment analysis was then conducted for the 2,767 CpGs that reached FDR significance and the 339 CpGs that reached the 2.4×10^{-7} genome-wide significance level⁴ (Supplementary Table 15). Our results showed that the functional enrichment results in these sensitivity analyses were largely congruent with our main analysis results (Supplementary Fig. 2-3, Supplementary Tables 16-18). In particular, we still observed significant enrichment in polycomb repressed regions for hyper-methylated CpGs in AD, as well as in binding sites of polycomb repressive complex 2 subunits EZH2 and SUZ12. Moreover, pathway analysis still showed significant enrichment in immune system related pathways (Supplementary Table 19-20) for both FDR significant and genome-wide significant CpGs after *bacon* correction.

2. Linked to the point above, the use of analysis tools such as OSCA (Zhang et al. Genome Biology 2019) should be used to account for CpG correlation structure and hidden confounders that might drive inflation. This would allow for a more accurate identification of the main signals and aid in distinguishing the true- from the false-positives.

This reviewer raised an important point that utilizing software tools that account for inflation would allow more accurate identification of the main signals and aid in distinguishing the true- from the false-positives. In response, we studied the OSCA method proposed in (Zhang et al. Genome Biology 2019). The Mixed Linear Models proposed by Zhang et al. (2019) has the general form of $\mathbf{y} = C\boldsymbol{\beta} + W\mathbf{u} + \mathbf{e}$ where \mathbf{y} is the phenotype of interest. In our case, the phenotype of interest is Braak stage, which is an ordinal variable that ranges from 0 to 6. However, currently, OSCA does not support the analysis of ordinal phenotype variables, which would typically be modeled using proportional odds logistic regression model (see for example Section 6.2 in *An Introduction to Categorical Data Analysis* by Alan Agresti).

In response to this reviewer's concern about inflation and false positives, as described in our response to comment #1, we conducted additional sensitivity analyses by subjecting our individual cohort results to inflation correction by the *bacon* method (PMID: 28129774) and then meta-analyzing the *bacon*-corrected effect sizes and standard errors across cohorts. We found our main findings still hold in this sensitivity analysis. More specifically, the functional enrichment results based on inflation corrected FDR significant CpGs still showed hyper-methylated CpGs in AD were enriched in polycomb repressed regions and immune system related pathways. See details in items (1) – (4) below.

Revision

The results for the CpGs that reached genome-wide significance after *bacon*-correction were presented in our response to comment 1. Here we additionally present sensitivity analysis results for the CpGs that reached 5% FDR significance threshold after *bacon*-correction.

(1) Supplementary Figure 2B – Enrichment of chromatin state for *bacon* corrected FDR significant CpGs.

B Single cpG meta-analysis probes
ChromHMM: E073 – 15 coreMarks segments

Compare to Figure 3c - Enrichment of chromatin state using FDR significant CpGs (presented in initial version of manuscript)

C Single cpG meta-analysis probes
ChromHMM: E073 – 15 coreMarks segments

(2) Supplementary Table 17 – Enrichment of 2,767 *bacon*-corrected FDR significant CpGs from meta-analysis in different chromatin states.

hyper-methylated differences in AD	After bacon correction			
	fdr significant CpGs		genome-wide significant CpGs	
	p-value	OR	p-value	OR
Active TSS	2.58E-01	0.93	2.72E-01	1.17
Bivalent Enhancer	5.60E-05	2.37	1.00E+00	0.93
Bivalent/Poised TSS	2.67E-02	1.37	9.12E-02	1.65
Enhancers	6.67E-05	1.50	2.77E-01	0.69
Flanking Active TSS	2.46E-61	3.91	4.11E-18	4.90
Flanking Bivalent TSS/Enh	4.17E-01	1.22	3.23E-01	1.61
Genic enhancers	3.59E-03	1.84	1.00E+00	0.75
Heterochromatin	2.66E-03	0.00	6.37E-01	0.00
Quiescent/Low	2.05E-63	0.23	2.38E-25	0.03
Repressed PolyComb	4.53E-65	4.37	8.52E-32	7.84
Strong transcription	3.51E-09	0.41	6.33E-03	0.34
Transcr. at gene 5' and 3'	1.34E-05	4.46	1.66E-02	5.73
Weak Repressed PolyComb	3.13E-05	0.64	2.63E-05	0.24
Weak transcription	2.14E-09	0.59	3.58E-05	0.37
ZNF genes & repeats	8.30E-02	0.00	1.00E+00	0.00

Compare to Supplementary Table 4 – enrichment of 3,751 FDR significant CpGs from previous submission

hyper-methylated differences in AD	DMR		CpG	
	p-value	OR	p-value	OR
Active TSS	8.87E-16	0.49	2.49E-02	0.90
Bivalent Enhancer	2.04E-01	1.55	2.71E-06	2.29
Bivalent/Poised TSS	2.49E-01	0.74	1.58E-01	1.18
Enhancers	4.74E-03	1.76	2.33E-09	1.62
Flanking Active TSS	5.36E-16	2.94	1.71E-88	3.84
Flanking Bivalent TSS/Enh	6.08E-03	0.13	9.96E-02	1.37
Genic enhancers	8.11E-01	1.09	2.66E-03	1.71
Heterochromatin	4.34E-01	0.00	7.52E-04	0.10
Quiescent/Low	5.00E-09	0.31	3.02E-72	0.30
Repressed PolyComb	2.32E-70	7.56	1.29E-66	3.60
Strong transcription	3.08E-10	0.07	1.59E-10	0.47
Transcr. at gene 5' and 3'	7.42E-07	8.26	4.73E-06	3.85
Weak Repressed PolyComb	1.94E-03	0.53	3.84E-03	0.79
Weak transcription	5.07E-02	0.70	2.56E-13	0.59
ZNF genes & repeats	6.43E-01	0.00	1.77E-02	0.00

(3) Supplementary Table 18 After bacon correction, LOLA analysis of 2,767 FDR significant CpGs from meta-analysis

Antibody	CellType	Filename	OR	pValue	FDR
IKZF1_(IKN)_ (UCLA)	GM12878	wgEncodeAwgTfbsSy	4.28	6.66E-21	2.01E-18
BCL11A	GM12878	wgEncodeAwgTfbsH	3.15	2.85E-18	7.81E-16
EZH2_(39875)	H1-hESC	wgEncodeAwgTfbsB	1.90	9.27E-16	1.55E-13
PU.1	GM12891	wgEncodeAwgTfbsH	2.01	1.57E-14	2.15E-12
p300_(SC-584)	GM12878	wgEncodeAwgTfbsSy	4.59	7.75E-14	9.92E-12
BATF	GM12878	wgEncodeAwgTfbsH	2.55	1.41E-12	1.42E-10
PU.1	GM12878	wgEncodeAwgTfbsH	1.92	1.99E-12	1.94E-10
c-Fos	MCF10A-Er-Src	wgEncodeAwgTfbsSy	2.03	6.67E-11	5.05E-09
eGFP-FOS	K562	wgEncodeAwgTfbsU	2.80	1.02E-10	7.13E-09
c-Fos	MCF10A-Er-Src	wgEncodeAwgTfbsSy	1.89	5.48E-10	3.59E-08
EBF1_(SC-137065)	GM12878	wgEncodeAwgTfbsSy	1.87	6.29E-10	4.03E-08
p300_(SC-584)	HeLa-S3	wgEncodeAwgTfbsSy	2.39	1.96E-09	1.18E-07
SUZ12	H1-hESC	wgEncodeAwgTfbsSy	2.26	9.44E-08	4.99E-06
EZH2_(39875)	NH-A	wgEncodeAwgTfbsB	1.55	3.68E-07	1.85E-05
SUZ12	NT2-D1	wgEncodeAwgTfbsSy	1.77	4.61E-07	2.17E-05
p300	GM12878	wgEncodeAwgTfbsH	2.20	1.13E-06	4.81E-05
GR	ECC-1	wgEncodeAwgTfbsH	2.36	1.19E-06	4.99E-05
EBF1_(SC-137065)	GM12878	wgEncodeAwgTfbsH	1.53	2.13E-06	8.43E-05
CtBP2	H1-hESC	wgEncodeAwgTfbsSy	1.49	3.30E-06	1.27E-04
GR	A549	wgEncodeAwgTfbsH	1.90	3.49E-06	1.33E-04

Compare to Supplementary Table 5 from previous submission – enrichment of 3,751 FDR significant CpGs

Supplementary Table 5 Enrichment of FDR significant CpGs in binding sites of ENCODE transcription factors and chromatin proteins.

Antibody	CellType	Description	Filename	OR	pValue	FDR
EZH2_(39875)	H1-hESC	ChIP H1-hESC EZH2_(39875)	wgEncodeAwgTfbs	1.89	3.41E-20	7.34E-18
IKZF1_(IKN)_ (UCLA)	GM12878	ChIP GM12878 IKZF1_(IKN)	wgEncodeAwgTfbs	3.48	2.99E-18	5.64E-16
BCL11A	GM12878	ChIP GM12878 BCL11A	wgEncodeAwgTfbs	2.74	2.37E-17	3.57E-15
p300_(SC-584)	GM12878	ChIP GM12878 p300_(SC-58)	wgEncodeAwgTfbs	4.39	1.38E-16	1.89E-14
PU.1	GM12891	ChIP GM12891 PU.1	wgEncodeAwgTfbs	1.87	5.23E-15	6.06E-13
PU.1	GM12878	ChIP GM12878 PU.1	wgEncodeAwgTfbs	1.77	2.02E-12	1.65E-10
SUZ12	H1-hESC	ChIP H1-hESC SUZ12	wgEncodeAwgTfbs	2.40	1.23E-11	9.01E-10
BATF	GM12878	ChIP GM12878 BATF	wgEncodeAwgTfbs	2.22	2.26E-11	1.58E-09
c-Fos	MCF10A-Er-Src	ChIP MCF10A-Er-Src c-Fos	wgEncodeAwgTfbs	1.87	5.96E-11	3.90E-09
c-Fos	MCF10A-Er-Src	ChIP MCF10A-Er-Src c-Fos	wgEncodeAwgTfbs	1.77	2.53E-10	1.60E-08
EBF1_(SC-137065)	GM12878	ChIP GM12878 EBF1_(SC-13)	wgEncodeAwgTfbs	1.75	4.24E-10	2.55E-08
p300_(SC-584)	HeLa-S3	ChIP HeLa-S3 p300_(SC-584)	wgEncodeAwgTfbs	2.22	5.20E-10	3.07E-08
eGFP-FOS	K562	ChIP K562 eGFP-FOS	wgEncodeAwgTfbs	2.37	2.75E-09	1.51E-07
EZH2_(39875)	NH-A	ChIP NH-A EZH2_(39875)	wgEncodeAwgTfbs	1.54	8.35E-09	4.41E-07
GR	A549	ChIP A549 GR	wgEncodeAwgTfbs	1.91	5.02E-08	2.48E-06
GR	ECC-1	ChIP ECC-1 GR	wgEncodeAwgTfbs	2.21	3.60E-07	1.49E-05
GATA-2	HUVEC	ChIP HUVEC GATA-2	wgEncodeAwgTfbs	1.74	4.20E-07	1.71E-05
SUZ12	NT2-D1	ChIP NT2-D1 SUZ12	wgEncodeAwgTfbs	1.65	6.12E-07	2.46E-05
c-Fos	HUVEC	ChIP HUVEC c-Fos	wgEncodeAwgTfbs	1.49	1.50E-06	5.46E-05

(4) Supplementary Table 19 Top 20 most significant pathways associated with AD Braak stage identified in meta-analysis. Significant single CpGs were selected as FDR significant CpGs after bacon correction.

	Description	P-value	FDR
GO:0045321	leukocyte activation	7.91E-15	1.73E-10
GO:0001775	cell activation	1.53E-14	1.73E-10
GO:0007159	leukocyte cell-cell adhesion	1.63E-11	1.23E-07
GO:0002376	immune system process	4.14E-11	1.97E-07
GO:0022409	positive regulation of cell-cell adhesion	4.89E-11	1.97E-07
GO:0046649	lymphocyte activation	5.23E-11	1.97E-07
GO:1903037	regulation of leukocyte cell-cell adhesion	7.53E-11	2.43E-07
GO:0042110	T cell activation	2.97E-10	8.37E-07
GO:0022407	regulation of cell-cell adhesion	4.40E-10	1.10E-06
GO:0006955	immune response	4.94E-10	1.12E-06
GO:0002694	regulation of leukocyte activation	5.83E-10	1.20E-06
GO:0050865	regulation of cell activation	1.01E-09	1.89E-06
GO:0002682	regulation of immune system process	1.25E-09	2.17E-06
GO:0002521	leukocyte differentiation	1.93E-09	3.11E-06
GO:0051249	regulation of lymphocyte activation	2.38E-09	3.58E-06
GO:1903039	positive regulation of leukocyte cell-cell adhesion	3.24E-09	4.35E-06
GO:0002252	immune effector process	3.28E-09	4.35E-06
GO:0050867	positive regulation of cell activation	7.24E-09	9.09E-06
GO:0006954	inflammatory response	1.19E-08	1.41E-05
GO:0050863	regulation of T cell activation	1.54E-08	1.73E-05

3. The analyses for overlap with genetic loci (“Correlation and enrichment with genetic susceptibility loci”) are informal (as are the CAMTA1 analyses earlier in the paper). I think colocalization and Mendelian Randomisation approaches would offer greater insight.

In response to this reviewer’s comment, we additionally performed co-localization analysis of association signals in a recent AD meta-analysis (Kunkle et al. (2019) PMID: 30820047) and ROSMAP mQTL study using the method of Giambartolomei et al. (2014) (PMID: 24830394). Among the 24 LD blocks genetic variants reaching genome-wide significance in Kunkle et al. (2019), co-localization analysis provided strong support for 2 genomic regions to include a single causal variant common to both phenotypes (i.e. AD status and CpG methylation levels). These 2 regions and the predicted most likely causal SNPs are included in an additional supplementary table (Supplementary Table 14). See additional details under “Revision” below.

We also agree that mediation analysis such as Mendelian Randomization (MR) would offer greater insight by identifying CpG methylation that mediates genetic risk for AD. In the previous submission, we actually attempted mediation analysis using a similar approach, the Causal Inference Test (CIT). Ainsworth et al. (2017) (PMID: 28691305) commented that “The CIT... is more flexible than MR because it does not assume that the genetic variant is chosen specifically to be an instrument for the mediator. Due to the way the test is constructed, the CIT is also immune to problems of pleiotropy and reverse confounding.”

More specifically, to prioritize CpG methylations that mediate effects of mQTLs, let G = genotype, Y = AD status, M = residual methylation levels (after removing age, sex, cell type and batch effects), we computed p-values in linear models corresponding to each of the four components of CIT: (1) G and Y are associated: p-value for SNP in model $\text{Logit}(\text{pr}(\text{AD})) = \text{SNP} + \text{age} + \text{sex}$; (2) G is associated with M after adjusting for Y: p-value for SNP in model $\text{Methylation_resid} \sim \text{SNP} + \text{AD}$ (3) M is associated with Y after adjusting for G: p-value for AD status in model $\text{Methylation_resid} \sim \text{SNP} + \text{AD}$ and (4) G is independent of Y after adjusting for M: p-value for SNP in model $\text{Logit}(\text{Pr}(\text{AD})) = \text{SNP} + \text{methylation_resid} + \text{age} + \text{sex}$. The final CIT p-value was defined to be maximum of the four component test p-values, based on the intersection-union test principal. See Figure 2 in Liu et al. (2013) (PMID: 23334450) for a description of CIT analysis workflow.

However, using the CIT method, we did not detect mediation effects for any of the FDR-significant Braak associated CpGs or DMRs. The lowest p-value for CIT in our results was 0.0958, therefore, we did not include this mediation analysis results in the manuscript.

Revision

(1) In Methods – under “Correlation and co-localization with genetic susceptibility loci”

For co-localization analysis, we used the R package *coloc* to compare association signals in the AD GWAS meta-analysis⁵, which associated genetic variants with AD status (trait 1), with results from a mQTL analysis which associated genetic variants with methylation levels at CpGs within 500 kb of the 24 LD blocks (trait 2) computed using the ROSMAP dataset. Given observed data for trait 1 and trait 2, the co-localization analysis computes Bayesian posterior probabilities (PP_i) for each of the following five hypotheses⁶: (a) H_0 : No association with either trait; (b) H_1 : Association with trait 1, not with trait 2; (c) H_2 : Association with trait 2, not with trait 1; (d) H_3 : Association with trait 1 and trait 2, two independent SNPs; and (e) H_4 : Association with trait 1 and trait 2, one shared SNP. The summary statistics obtained in Kunkle et al. (2019)⁵ were downloaded from <https://www.niagads.org/igap-rv-summary-stats-kunkle-p-value-data> (file “Kunkle_etal_Stage1_results.txt”).

(2) In Results – 3rd paragraph under “Correlation and co-localization with genetic susceptibility loci”

Given the observed overlap between AD pathology associated CpGs and AD genetic risk loci, we next sought to determine whether the association signals at the GWAS loci (variant to AD status as determined by clinical consensus diagnosis of cognitive status, and variant to CpG methylation levels) are due to a single shared causal variant or to distinct causal variants close to each other. To this end, we performed a co-localization analysis using the method described in Giambartolomei et al. (2014)⁶. The results of this co-localization analysis strongly suggested⁷ (i.e. $PP_3+PP_4 > 0.90$, $PP_4 > 0.8$ and $PP_4/PP_3 > 5$, Online Methods) that 2 out of the 24 regions included a single causal variant common to both phenotypes (i.e. AD status and CpG methylation levels). The CpGs associated with these causal variants are located on the *SPI1*⁸ and *ADAMI0*⁹ genes (Supplementary Table 14).

(3) Supplementary Table 14

Supplementary Table 14 Bayesian co-localization of association signals from AD meta-analysis (Kunkle et al. (2019) PMID: 30820047) and ROSMAP mQTL study identified 2 GWAS nominated regions that included a single causal variant common to both traits (i.e. AD status and DNA methylation levels). Shown are GWAS nominated loci (AD_GWASloci), CpG and annotations (UCSC_RefGene_Name, UCSC_RefGene_Group), number of SNPs in the region (nsnps), estimated posterior probabilities for each of the co-localization hypotheses tested (PP3, PP4; Online Methods). In particular, PP3 quantifies support for the hypothesis of co-localization of two independent SNPs associated with the two traits, and PP4 quantifies support for the hypothesis of co-localization of one shared SNP associated with both traits. For each AD GWAS nominated region, shown are SNP with the highest estimated posterior probability for being the true causal variant for both traits (Best causal SNP), SNP with the lowest p-value for association with AD in Kunkle et al. (2019) (GWAS.SNP, GWAS.pval) and SNP with the lowest p-value for association with CpG methylation (mQTL.pval, mQTL.SNP).

Minor comments:

Introduction

4. ‘AD’ is used throughout the first paragraph of the introduction but only introduced - ‘Alzheimer’s

disease (AD)' - in the second paragraph

In response, we've now moved introduction for "Alzheimer's disease (AD)" to the first paragraph.

Revision – In Introduction, 1st paragraph

The causes of **Alzheimer's disease (AD)** are complex, with the disease likely resulting from a complicated interplay of genetic factors and environmental factors.

5. 'the use of a disease relevant tissue is critical for informative epigenetic studies' – while it may be preferable to investigate methylation changes in the target tissue, use of peripheral and accessible tissue can still be informative

In response, we rephrased this sentence as the reviewer suggested.

Revision

(1) In Introduction, 2nd paragraph

For neurological disorders such as AD, the use of disease-relevant tissue **is often preferred** for epigenetic studies. **However**, obtaining sufficient sample sizes for brain studies is challenging because of the difficulty in procuring post-mortem human brain tissue.

6. 'brain tissues provide the ideal source for study samples' – the authors should make it clear that there are still issues with using post-mortem brain tissue in such studies eg. it typically represents the end stages of disease and it's still unclear if there are post-mortem changes in methylation patterns and how post-mortem intervals (which aren't corrected for here) may effect these

We thank this reviewer for the good suggestions. In response, we have revised accordingly and incorporated additional comments on postmortem brain tissues in the Discussion section. Please see the details below.

Revision - In Discussion, 13th paragraph or 2nd paragraph from last

Also, **although using target tissues to study brain disorders is often preferred, there are still issues with using post-mortem brain tissue in such studies. For example, it typically represents the end stages of the disease, and it is still unclear if there are post-mortem changes in methylation patterns and how post-mortem intervals might affect them.**

7. 'the difference in corrected DNA methylation (%) between individuals with the lowest (score 0) and highest (score VI) Braak score ranged from 1% to 5%' – are these differences relative to the mean and standard deviation of the probe? For example, a difference of 1% is much greater if the mean is 10% (SD 0.1%) than if it is 50% (SD 15%)

To clarify, we believe the differences in corrected DNA methylation levels are absolute, and not relative to mean and standard deviations. Please see Table 1 from Lunnon et al. (2014) (PMID: 25129077).

In response, we clarified these differences are absolute differences. Please see revision below.

Revision – In Introduction, 2nd paragraph.

For example, in the Lunnon et al. epigenome-wide association study (EWAS)¹⁰, which examined postmortem brain tissues in **pathological** AD subjects and controls, the **absolute** difference in corrected DNA methylation between individuals with the lowest (score 0) and highest (score VI) Braak score ranged from 1% to 5% change even for the most significant CpGs in the prefrontal cortex region, a region that shows considerable vulnerability to AD.

8. The authors refer to methylation ‘changes’ (here and throughout the manuscript) – this should be changed to ‘differences’

In response, “methylation changes” were changed to “methylation differences” at a total of 41 places throughout the manuscript.

9. ‘in addition to corroborating previous findings...’ – it would help to clarify which findings are from basic re-analysis of the same data and which are novel

To clarify, Supplementary Table 1 (columns AD-AH) and Supplementary Table 2 (columns AV-AZ) indicate which findings are novel. In response, we’ve added additional clarifications by pointing the readers to Supplementary Tables 1 & 2.

Revision – In Discussion, 5th paragraph

In addition to corroborating previous findings in single-cohort analysis using the same methylation datasets (**Supplementary Tables 1 and 2**), our meta-analysis also uncovered a number of novel differentially methylated genes.

10. ‘suggesting their potential value as AD biomarkers’ – this is quite a bold statement and would need justification

In response, we removed this part of the sentence.

Results

11. Study cohort characteristics

o ‘women’ should be changed to ‘female’

This was corrected.

Revision – In Results section, under “Study cohort characteristics”

Among the four cohorts, the mean **age at death** ranged from 73.6 years to 86.3 years, and the percentage of **females** ranged from 51.8% to 63.5%.

- Meta-analysis identified previously reported as well as novel DNA methylation changes significantly associated with AD Braak stage at individual CpGs and co-methylated genomic regions

12. o ‘Adjusting for [estimated] cell type proportions’ – authors should also make it clear that it was the proportion of neurons that was controlled for

This was corrected.

Revision – In Results section, 2nd paragraph

Adjusting for **estimated** cell-type proportions (**i.e., the proportion of neurons**), **age at death**, sex, and batch effects, our meta-analysis of single CpGs in the four cohorts identified 3,979 statistically significant individual CpGs at 5% false discovery rate (FDR).

13. o ‘Therefore, methylation changes at individual CpGs and DMRs are located at different regions in the genome complementing each other, which provide a more complete picture of the Braak associated methylome.’ – this sentence is unclear

We re-wrote this sentence to make it more clear.

Revision - In Results section, 4th paragraph

Notably, 118 out of the 119 DMRs included FDR significant individual CpGs. On the other hand, only 421 out of the 3,751 FDR significant individual CpGs overlapped with the FDR significant DMRs. Therefore, **methylation differences at individual CpGs and DMRs did not completely overlap, so analyzing both individual CpGs and DMRs provided a more complete picture of the Braak-associated methylome in the prefrontal cortex.** Our final set of DNA **methylation differences** included these 3,751 individual CpGs along with the 119 DMRs (Fig. 2).

14. o Rather than removing probes associated with smoking it would be preferable to re-run the analysis with smoking as a covariate

This is a good suggestion, but unfortunately smoking information was not available for any of the four datasets we analyzed.

15. o ‘among which 24% are located in non-coding regions’ – how does this compare to the representation of non-coding regions on the arrays themselves?

Revision – In Results, 2nd paragraph

we obtained 3,751 significant CpGs (Supplementary Table 1), **of which 47.8% is located in non-coding regions. This proportion is lower than the proportion of non-coding probes (53.2%) on the array (P-value = 2.18×10^{-11}).**

16. o 'Next, we combined the cohort specific p-values for these genomic regions using meta-analysis' – I know this is mentioned in the methods but I would restate here that this is an inverse-variance fixed effects meta

This was revised as the reviewer suggested.

Revision – In Results, 3rd paragraph

Next, we combined the cohort-specific P-values for these genomic regions using an inverse-variance fixed effects meta-analysis.

17. o 'which provide a more complete picture of the Braak associated methylome [in the prefrontal cortex]'

This was revised as the reviewer suggested.

Revision - In Results section, 4th paragraph

Notably, 118 out of the 119 DMRs included FDR significant individual CpGs. On the other hand, only 421 out of the 3,751 FDR significant individual CpGs overlapped with the FDR significant DMRs. Therefore, methylation differences at individual CpGs and DMRs did not completely overlap, so analyzing both individual CpGs and DMRs provided a more complete picture of the Braak-associated methylome in the prefrontal cortex. Our final set of DNA methylation differences included these 3,751 individual CpGs along with the 119 DMRs (Fig. 2).

18. o 'Interestingly, this gene is observed to be expressed mainly in neurons' – needs a reference

We've added a reference to this sentence.

Revision – In Discussion, 3rd paragraph

Interestingly, this gene is observed to be expressed mainly in neurons¹¹.

Reference

46. Liu, Y., Wang, M., Marcora, E.M., Zhang, B. & Goate, A.M. Promoter DNA hypermethylation - Implications for Alzheimer's disease. *Neurosci Lett* **711**, 134403 (2019).

19. o 'In addition to corroborating previous findings, our meta-analysis also uncovered a number of novel differentially methylated genes' – the 'corroboration of previous findings' should be rephrased to reflect that results are from the same data

We edited this sentence as suggested. Also, another reviewer has suggested moving this paragraph to Discussion, please see below.

Revision – In Discussion section, 5th paragraph

In addition to corroborating previous findings in single-cohort analysis using the same methylation datasets (Supplementary Tables 1 and 2), our meta-analysis also uncovered a number of novel differentially methylated genes.

20. - Enrichment analysis of significant DNA methylation changes highlighted immune-related processes and Polycomb Repressive Complex 2 (PRC2) in AD

o I think this section would benefit from a more general introduction to the biological features being investigated, particularly for a general journal such as Nature Comms.

In response to this reviewer's comment, we have added additional introductions to the genomics and epigenomics features we studied in the enrichment analyses.

Revision – In Results section, under “Enrichment analysis of significant DNA methylation differences in pathological AD highlighted immune-related processes and polycomb repressive complex 2 (PRC2)”

1st paragraph: The probes on the Illumina 450k array are annotated according to their locations with respect to genes (TSS1500, TSS200, 5'UTR, 1stExon, gene body, 3'UTR, intergenic) or to CpG islands (island, shore, shelf, open sea).

2nd paragraph: In addition, we also compared our results with epigenomic annotations including chromatin states and transcription factor binding sites. Using combinations of histone modification marks, computational algorithms such as ChromHMM¹² segment and annotate the genome with different chromatin states (repressed, poised and active promoters, strong and weak enhancers, putative insulators, transcribed regions, and large-scale repressed and inactive domains), which were shown to vary across sex, tissue type, and developmental age¹³. For our analysis, we used the 15-state ChromHMM annotation of the Roadmap Epigenomics project¹⁴.

21. - Prioritizing significant DNA methylation changes with sample matching

o 'too few age-matched samples' – this should be quantified – how many is too few?

To clarify, only 12 samples were available for age and sex matched samples in the Gasparoni cohort, we therefore did not include this dataset for the matched samples analysis.

Revision – under “Prioritizing significant DNA methylation differences with sample matching”

We did not include the Gasparoni cohort for this analysis because too few age and sex matched samples (n = 12) were present in this dataset.

22. o 'note that the significant findings in this small sample analysis are still valid' – what is meant by 'valid'?

Another reviewer also had issues with this sentence, so we've removed it from the manuscript.

23. o 'Among them, 75% (n = 218) CpGs and 44% (n = 17) genomic regions overlapped with the significant CpGs and DMRs in our main analysis described above' –Was there any attenuation in the effect sizes?

This reviewer raised an important point. As another reviewer suggested, we re-ran the matched samples analysis by matching each AD case with a control subject with both the same sex and age at death in the same cohort. For the results, we identified 151 significant CpGs and 32 significant DMR with less than 5% *fdr*. Among them, 85% (n = 129) CpGs and 50% (n = 16) genomic regions overlapped with the significant CpGs and DMRs in our main analysis with the same direction of changes.

To clarify, compared to original estimated effect size for association between Braak stage and methylation values, we found both attenuation and strengthening of estimated effects in the matched samples analysis. In the figures below, we see that dots appear on both side of the regression line (above the line = strengthening of effects, below the line = attenuation of effects)

(a) comparison of estimated effects for CpGs in original analysis vs. matched samples analysis

(b) comparison of estimated effects for DMRs in original analysis vs. matched samples analysis

24. - Correlation of AD associated methylation changes in blood and brain samples

o Authors refer to 'adjusted correlation analysis' – it would be beneficial to include information about the covariates included in this analysis here and a signpost to the relevant methods section

We revised this sentence as suggested.

Revision – In Results, under "Correlation of AD-associated CpGs and DMRs methylation levels in blood and brain samples", 1st paragraph

We performed both an adjusted correlation analysis based on methylation residuals (r_{resid}), in which we adjusted estimated neuron proportions for brain samples (or estimated blood cell-type proportions), array,

age at death (for brain samples) or at blood draw (for blood samples), and sex, and an unadjusted correlation analysis based on beta values (r_{beta}) (Online Methods).

25. - Correlation of AD associated methylation changes with expressions of nearby genes
o 'we first removed age, sex...' – this would be clearer if it read 'adjusted for' rather than 'removed'

We revised this sentence as suggested.

Revision

To reduce the effect of potential confounding effects, when testing for methylation-gene expression associations, we first adjusted for age at death, sex, cell-type proportions, and batch effects in both DNA methylation and gene expression levels separately and extracted residuals from the linear models.

Discussion

26. - 'Our study illustrates the power of meta-analysis for epigenome-wide association studies' – it would be helpful if the authors added how the study illustrates this

In response, we added clarifications about how meta-analysis improves power.

Revision – In Discussion section, 1st paragraph

Our study illustrates the power of meta-analysis for EWAS. In individual cohort analysis, we obtained between 0 and 40 FDR significant CpGs per cohort. In comparison, we obtained 3,751 FDR significant CpGs in our meta-analysis.

27. - inconsistent use of PRC2/polycomb repressive complex 2

To clarify, PRC2 is acronym for polycomb repressive complex 2. We clarified in the first appearance of PRC2 that it is the same as polycomb repressive complex 2.

Section title

Enrichment analysis of significant DNA methylation differences in pathological AD highlights immune-related processes and polycomb repressive complex 2 (PRC2)

28. - unclear what is meant by 'substantial cross-talk has been observed between PRC2 and DNA methylation, two key epigenetics mechanisms for gene repression'. Again some more general comments about PRC2 would be very helpful here

In response, we added several sentences on the general comments about PRC2, along with two key review papers on PRC2.

Revision – In Discussion, 7th paragraph

PRC2 is a type of polycomb group (PcG) protein and plays important roles in multiple biological processes including proliferation and differentiation as well as maintenance of cellular identity through regulation of gene expression. The PRC2 protein complex is composed of three core component proteins: SUZ12, EED and either EZH1 or EZH2, which along with RBBP4 or RBBP7 forms distinct

subcomplexes by associating with different interaction partners. It is highly conserved and was originally discovered in *Drosophila* studies as a suppressor of Hox genes^{15, 16}.

29. - 'Plasticity nature' – plastic nature?

We revised this sentence as suggested.

Revision

On the other hand, epigenetic therapies that leverage the **plastic** nature of DNA methylation might also be an alternative strategy to modulate inflammation¹⁷.

30. - use of 'robust' in the first sentence of the final paragraph is perhaps not warranted seeing as a lenient significance threshold was used

In response, we removed "robust" from this sentence.

Revision – in Discussion section, last paragraph

In summary, we have identified numerous **methylation differences** at DMRs and CpGs consistently associated with AD Braak stage in multiple cohorts.

31. - 'a new computational framework' – this is a bit of an overstatement - the tools used for analysis are pre-existing so don't represent a new computational framework

In response, we removed this sentence.

32. - 'Facilitating development of effective therapeutic strategies for AD' – again this is a bold statement – how will it do this?

In response, we also removed this part of the sentence.

Revision

Methods

- Study cohorts

33. o '(late stage AD)' – it should be noted this is late stage tau pathology – there is no information about symptomatology or amyloid staging

We thank this reviewer for the good suggestion. In response, we have modified text to indicate late stage tau pathology.

Revision – In Methods, 1st paragraph

In all these datasets, brain samples were classified according to Braak stage¹⁸, with scores ranging from 0 (control) to VI (late stage **tau pathology** AD), indicating different levels of severity of the disease.

- Pre-processing of DNA methylation Data -

34. o misspelling of Illumina in the first sentence

This was corrected.

Revision – In Methods

All methylation datasets were measured by the same Illumina HumanMethylation 450K beadchip

- Meta-analysis

35. o Post-mortem interval should also be considered as a covariate in analyses

This reviewer raised an important point. In response, we re-run our analysis for the ROSMAP cohort, by additionally including PMI in model methylation M value \sim neuron.proportions + batch + array + ageAtDeath + sex + PMI. The Bland-Altman plot in the figure below shows the estimated effect sizes using a model that additionally adjust for PMI are very similar to those obtained from the model not adjusting for PMI. In particular, in individual CpGs analysis, the 95% confidence interval for the absolute difference in the effect sizes from these two models is $(-1.31 \times 10^{-3}, 1.50 \times 10^{-3})$.

In addition, in the ROSMAP cohort, PMI was not significantly associated with Braak stage ($r = -0.018$, p -value = 0.6156), so PMI is unlikely to be a confounder for Braak stage.

Also, we did not have PMI information in the other three public datasets (London, Mount Sinai, Gasparoni). Therefore, we decided not to include PMI in our final analysis.

In response to this reviewer's comment, we added clarifications on our rationale for not including PMI in the Discussion section.

Revision – In Discussion, 2nd from last paragraph

In this study, we did not adjust for post-mortem interval (PMI), because in the ROSMAP cohort PMI was not significantly associated with Braak stage (Spearman correlation $r = -0.018$, P -value = 0.6156), so is unlikely to be a confounder for it. Also, PMI was not available for the other three public datasets (London, Mount Sinai, Gasparoni) we analyzed.

36. o 'Fixed effects model (also referred to as inverse variance-weighted) meta-analysis was applied to synthesize statistical significance from individual cohorts' – I think you can simplify this to "An inverse variance-weighted fixed effects meta-analysis.."

This sentence was edited as the reviewer suggested.

Revision

The **inverse-variance weighted fixed effects model** was applied to synthesize statistical significance from individual cohorts.

37. o 'More specifically, we first matched each case with a control sample using matchControls function in e1071 R package' – more detail on how this package performs the matches and how closely cases and controls were matched would be beneficial

In response to this reviewer's comment, we added more clarifications about the matching process in the manuscript.

Revision – In Methods, 5th paragraph under "Meta-analysis"

More specifically, we first matched each case with control samples **with the same age at death (in years) and sex in the same cohort** using the matchControls function in the e1071 R package. **When there are multiple matched control samples, the control sample with the most similar age as the case sample is selected.**

38. - Functional annotation of significant methylation changes

o 'The enrichment of significant CpGs, or DMS (differentially methylated sites)...' – does this account for the correlation structure among CpGs?

This reviewer raised an important point. To clarify, we used Fisher's exact test to test for enrichment of significant CpGs in different types of genomic features which did not account for correlations between CpGs. In response to this reviewer's comment, we performed an additional enrichment analysis using a logistic mixed effects regression model that does account for correlations between CpGs. Our results showed enrichment analysis results based on the mixed-effects model are very similar to those based on Fisher's test (see Table below).

Revisions –

(1) In Methods, 2nd paragraph under "Functional Annotation of significant methylation changes"

In addition, we also explored an alternative enrichment analysis that accounts for correlations between CpGs using a logistic mixed effects regression model. More specifically, for each type of genomic feature (e.g., CpG island), we tested for an association between the type of genomic region (e.g., isCpGisland = "yes" or "no") and significance of the CpG (e.g., isSignificant = "yes" or "no"). Random effects for each chromosome were also included in this model to account for correlations between CpGs within the same chromosome. This analysis was performed using SAS procedures GLIMMIX and HPMIXED, which implemented specialized high-performance techniques designed to cope with estimation problems in mixed effects models with large datasets. The enrichment analysis results based on the mixed model were similar to those based on Fisher's test (Supplementary Table 24).

(2) Supplementary Table 24 Enrichment analysis results using mixed effects model (with random chromosomes effects to model correlations between CpGs) are similar to those from Fisher's test.

hyper-methylated CpGs in AD				hyper-methylated CpGs in AD					
Feature	Fisher's test		Mixed model		Feature	Fisher's test		Mixed Model	
	OR	p-value	OR	p-value		OR	p-value	OR	p-value
Island	1.26	3.84E-07	1.24	2.35E-06	Active TSS	0.90	2.49E-02	0.90	3.31E-02
OpenSea	0.79	1.76E-07	0.82	1.76E-05	Bivalent Enhancer	2.29	2.71E-06	2.29	1.72E-07
Shelf	0.68	1.63E-06	0.66	7.70E-07	Bivalent/Poised TSS	1.18	1.58E-01	1.19	1.51E-01
Shore	1.18	7.01E-04	1.16	1.95E-03	Enhancers	1.62	2.33E-09	1.61	4.46E-10
1stExon	1.37	1.40E-02	1.37	1.14E-02	Flanking Active TSS	3.84	1.71E-88	3.74	7.73E-113
3'UTR	0.97	9.03E-01	0.97	8.22E-01	Flanking Bivalent TSS/Enh	1.37	9.96E-02	1.37	9.14E-02
5'UTR	1.20	5.89E-03	1.22	2.32E-03	Genic enhancers	1.71	2.66E-03	1.71	1.12E-03
Body	1.26	4.18E-07	1.22	8.95E-06	Heterochromatin	0.10	7.52E-04	0.10	1.99E-02
Intergenic	0.87	7.65E-03	0.91	6.89E-02	Quiescent/Low	0.30	3.02E-72	0.31	1.29E-48
TSS1500	0.92	1.54E-01	0.91	1.35E-01	Repressed PolyComb	3.60	1.29E-66	3.56	1.86E-87
TSS200	0.65	2.69E-09	0.64	4.83E-09	Strong transcription	0.47	1.59E-10	0.43	2.79E-10
					Transcr. at gene 5' and 3'	3.85	4.73E-06	3.85	4.12E-08
					Weak Repressed PolyComb	0.79	3.84E-03	0.79	3.92E-03
					Weak transcription	0.59	2.56E-13	0.57	4.80E-13
					ZNF genes & repeats	0.00	1.77E-02		did not converge

hypo-methylated CpGs in AD				hypo-methylated CpGs in AD					
Feature	Fisher's test		Mixed model		Feature	Fisher's test		Mixed Model	
	OR	p-value	OR	p-value		OR	p-value	OR	p-value
Island	0.15	1.99E-127	0.15	3.56E-76	Active TSS	0.13	8.74E-126	0.13	3.99E-69
OpenSea	2.69	5.18E-83	2.84	5.90E-86	Bivalent Enhancer	0.54	1.20E-01	0.54	1.06E-01
Shelf	1.33	3.95E-04	1.32	3.73E-04	Bivalent/Poised TSS	0.26	1.11E-08	0.26	4.42E-06
Shore	0.89	5.06E-02	0.88	4.13E-02	Enhancers	1.77	1.38E-09	1.75	1.82E-10
1stExon	1.00	9.32E-01	1.00	9.82E-01	Flanking Active TSS	0.73	1.82E-02	0.71	1.19E-02
3'UTR	1.12	3.84E-01	1.12	4.05E-01	Flanking Bivalent TSS/Enh	0.26	1.60E-03	0.26	7.88E-03
5'UTR	0.91	3.14E-01	0.91	2.53E-01	Genic enhancers	1.27	3.09E-01	1.27	2.97E-01
Body	1.13	2.26E-02	1.14	1.79E-02	Heterochromatin	0.42	1.33E-01	0.43	1.40E-01
Intergenic	1.15	1.38E-02	1.18	4.55E-03	Quiescent/Low	1.92	2.90E-32	2.01	3.10E-38
TSS1500	0.89	1.19E-01	0.88	6.71E-02	Repressed PolyComb	0.68	9.60E-03	0.67	1.08E-02
TSS200	0.70	2.44E-05	0.69	3.01E-05	Strong transcription	0.94	6.56E-01	0.93	5.20E-01
					Transcr. at gene 5' and 3'	0.31	3.88E-01	0.31	2.47E-01
					Weak Repressed PolyComb	1.92	3.32E-18	1.89	1.27E-19
					Weak transcription	1.57	1.73E-11	1.57	1.88E-12
					ZNF genes & repeats	0.00	8.73E-02		did not converge

- Correlation and overlap of significant methylation changes with genetic susceptibility loci
 39. o 'We then fit the linear model median (Methylation_resid) ~ SNP to test association between methylation residuals in DMRs and CpGs described above and the imputed allele dosages for SNPs to identify mQTLs.' – There does not appear to be any adjustments for population stratification here (genetic PCs). I think these analyses should be rerun with PC adjustments.

We thank this reviewer for the good suggestion. Another reviewer also suggested to include PCs as well as batch variable in ROSMAP genetic data for the mQTL analysis. In response, we re-run our mQTL analysis additionally including batch and first three PCs from genetic data. Supplementary Tables 11 and 12 include the updated mQTL analysis results.

Revisions

(1) In Results, 1st paragraph under "Correlation and co-localization with genetic susceptibility loci"

To identify methylation quantitative trait loci (mQTLs) for the significant DMRs and CpGs, we tested associations between the methylation levels with nearby SNPs, using the ROSMAP study dataset (imputed to the Haplotype Reference Consortium r1.1 reference panel)¹⁹, which had matched genotype data and DNA methylation data for 688 samples. To reduce the number of tests, we focused on identifying *cis* mQTLs located within 500kb from the start or end of the DMR (or position of the significant CpG)²⁰. Among 166,797 SNPs that are associated with AD, 11,670 were also significantly associated with methylation levels, after correcting for confounding effects age, sex, cell type, batch effects and PCs in methylation data. Among the 119 DMRs and 3,751 CpGs significantly associated with Braak stage, 37 DMRs and 1,010 CpGs had at least one corresponding mQTL in brain samples, respectively (Supplementary Tables 11 and 12).

(2) In Methods, under "Correlation and co-localization with genetic susceptibility loci"

We then fit the linear model $\text{methylation residual} \sim \text{SNP dosage} + \text{batch} + \text{PC1} + \text{PC2} + \text{PC3}$, where PC1, PC2, and PC3 are the first three PCs estimated from genotype data, to test the association between methylation residuals in CpGs and the imputed allele dosages for SNPs to identify mQTLs.

(3) Supplementary Table 11 and 12 – updated results for mQTL analysis, additionally including batch and first three PCs estimated from genetic data.

Supplementary Table 11 Among the 3751 FDR significant Braak-associated CpGs, 1010 CpGs had at least one corresponding mQTL in the prefrontal cortex brain samples. The mQTL analysis was performed using the ROSMAP cohort samples with matched genotype data and DNA methylation data for 688 samples. *cis* mQTLs located within 500kb from the CpGs were considered. The genotype data was imputed to HRC r1.1 reference panel and tested against AD status using logistic regression adjusting for age, sex, and first three PCs estimated from genotype data. AD status of the samples was determined using clinical consensus diagnosis of cognitive status at the time of death (cases: variable cogdx = 4, 5, controls: others). Logistic regression results include reference allele (REF), alternative allele (ALT), minor and tested allele in the regression model (A1) as well as frequencies associated with it in cases and controls (A1_FREQS_cases, A1_FREQS_controls), odds ratio which is exp (beta estimate for A1 allele) (OR), standard error of beta estimate for A1 allele (SE), Z-value for test statistic (Z_STAT), and p-value (Pval). mQTLs were identified by fitting a linear model with methylation residuals (adjusted for neuron.proportions, DNAm.batch, array, ageAtDeath, and sex) as outcome variable, SNP dosage, batch, and the first three PCs from genotype data as independent variables. mQTL analysis results include estimated effect size (Estimate_SNP), standard error of the estimate (StdError_SNP), p-value (P_SNP), and FDR obtained from the linear model. Annotations include the location of the SNP (SNP), identifier and location of the CpG (CpG, chr, position), the distance between SNP and CpG (SNP-CpGdistance), and nearby gene based on Illumina annotation (UCSC_RefGene_Name).

Supplementary Table 12 Among the 119 FDR significant DMRs, 37 DMRs had at least one corresponding mQTL in the brain samples. The mQTL analysis was performed using the ROSMAP cohort samples with matched genotype data and DNA methylation data for 688 samples. *cis* mQTLs located within 500kb from the start or end of the DMRs were considered. The genotype data was imputed to HRC r1.1 reference panel and tested against AD status using logistic regression adjusting for age, sex, and first three PCs estimated from genotype data. AD status of the samples was determined using clinical consensus diagnosis of cognitive status at the time of death (cases: variable cogdx = 4, 5, controls: others). Logistic regression results include reference allele (REF), alternative allele (ALT), minor and tested allele in the regression model (A1) as well as frequencies associated with it in cases and controls (A1_FREQS_cases, A1_FREQS_controls), odds ratio which is exp (beta estimate for A1 allele) (OR), standard error of beta estimate for A1 allele (SE), Z-value for test statistic (Z_STAT), and p-value (Pval). Each DMR was summarized using median methylation residuals of all CpGs located within the DMR first, then mQTLs were identified by fitting a linear model with median methylation residuals (adjusted for neuron.proportions, DNAm.batch, array, ageAtDeath and sex) as outcome variable, SNP dosage, batch, and first three PCs from genotype data as independent variables. mQTL analysis results include estimated effect size (Estimate_SNP), standard error of the estimate (StdError_SNP), p-value (P_SNP), and FDR obtained from the linear model. Annotations include the location of the SNP (SNP), identifier and location of the CpG (CpG, chr, position), the distance between SNP and CpG (SNP-CpGdistance), and nearby gene based on Illumina annotation (UCSC_RefGene_Name).

References

1. van Iterson, M., van Zwet, E.W., Consortium, B. & Heijmans, B.T. Controlling bias and inflation in epigenome- and transcriptome-wide association studies using the empirical null distribution. *Genome Biol* **18**, 19 (2017).
2. Devlin, B. & Roeder, K. Genomic control for association studies. *Biometrics* **55**, 997-1004 (1999).
3. Viana, J. et al. Schizophrenia-associated methylomic variation: molecular signatures of disease and polygenic risk burden across multiple brain regions. *Hum Mol Genet* **26**, 210-225 (2017).
4. Saffari, A. et al. Estimation of a significance threshold for epigenome-wide association studies. *Genet Epidemiol* **42**, 20-33 (2018).
5. Kunkle, B.W. et al. Genetic meta-analysis of diagnosed Alzheimer's disease identifies new risk loci and implicates Abeta, tau, immunity and lipid processing. *Nature genetics* **51**, 414-430 (2019).
6. Giambartolomei, C. et al. Bayesian test for colocalisation between pairs of genetic association studies using summary statistics. *PLoS Genet* **10**, e1004383 (2014).
7. Guo, H. et al. Integration of disease association and eQTL data using a Bayesian colocalisation approach highlights six candidate causal genes in immune-mediated diseases. *Hum Mol Genet* **24**, 3305-3313 (2015).
8. Huang, K.L. et al. A common haplotype lowers PU.1 expression in myeloid cells and delays onset of Alzheimer's disease. *Nat Neurosci* **20**, 1052-1061 (2017).
9. Vassar, R. ADAM10 prodomain mutations cause late-onset Alzheimer's disease: not just the latest FAD. *Neuron* **80**, 250-253 (2013).
10. Lunnon, K. et al. Methylomic profiling implicates cortical deregulation of ANK1 in Alzheimer's disease. *Nat Neurosci* **17**, 1164-1170 (2014).
11. Liu, Y., Wang, M., Marcora, E.M., Zhang, B. & Goate, A.M. Promoter DNA hypermethylation - Implications for Alzheimer's disease. *Neurosci Lett* **711**, 134403 (2019).
12. Ernst, J. & Kellis, M. ChromHMM: automating chromatin-state discovery and characterization. *Nat Methods* **9**, 215-216 (2012).
13. Yen, A. & Kellis, M. Systematic chromatin state comparison of epigenomes associated with diverse properties including sex and tissue type. *Nat Commun* **6**, 7973 (2015).
14. Chadwick, L.H. The NIH Roadmap Epigenomics Program data resource. *Epigenomics* **4**, 317-324 (2012).
15. Margueron, R. & Reinberg, D. The Polycomb complex PRC2 and its mark in life. *Nature* **469**, 343-349 (2011).
16. Liu, P.P., Xu, Y.J., Teng, Z.Q. & Liu, C.M. Polycomb Repressive Complex 2: Emerging Roles in the Central Nervous System. *Neuroscientist* **24**, 208-220 (2018).
17. Ciechomska, M., Roszkowski, L. & Maslinski, W. DNA Methylation as a Future Therapeutic and Diagnostic Target in Rheumatoid Arthritis. *Cells* **8** (2019).
18. Braak, H. & Braak, E. Staging of Alzheimer's disease-related neurofibrillary changes. *Neurobiol Aging* **16**, 271-278; discussion 278-284 (1995).
19. McCarthy, S. et al. A reference panel of 64,976 haplotypes for genotype imputation. *Nature genetics* **48**, 1279-1283 (2016).
20. Hannon, E. et al. An integrated genetic-epigenetic analysis of schizophrenia: evidence for co-localization of genetic associations and differential DNA methylation. *Genome Biol* **17**, 176 (2016).

Reviewer #3 (Remarks to the Author):

In this manuscript, Zhang et al. conducted a meta-analysis using data from four cohorts with more than one thousand prefrontal cortex brain samples to examine DNA methylation changes in association with Braak stage. The authors did very extensive analyses. I have the following comments:

We thank this reviewer for the thorough review and encouragement for our study. We have made substantial changes in response to this reviewer's helpful comments, which we will discuss in detail below.

1. Throughout the manuscript, I think the authors need to differentiate between AD dementia, pathological AD, Braak and cases. They are different concepts, and seem to be used interchangeably. It is confusing sometimes. Please check the whole manuscript and make changes.

We appreciate this reviewer's comment. In response, we made many changes in the manuscript to clarify these different concepts. Please see details below.

Revision

(1) In Introduction – 2nd paragraph

However, obtaining sufficient sample sizes for brain studies is challenging because of the difficulty in procuring post-mortem human brain tissue. This makes it difficult to detect the DNA methylation differences observed in the brains of AD subjects, because these differences are often of small magnitude. For example, in the Lunnon et al. epigenome-wide association study (EWAS)¹, which examined postmortem brain tissues in pathological AD subjects and controls

(2) In Introduction – last paragraph

In addition to corroborating previous findings, our analysis also nominated a number of new differentially methylated genes. Enrichment analysis of differentially methylated genes highlighted multiple immune processes epigenetically associated with pathological AD as well as polycomb repressed regions.

(3) In Results – under “Enrichment analysis of significant DNA methylation differences in pathological AD highlights immune-related processes and polycomb repressive complex 2 (PRC2)”

We tested enrichment of the significant methylation differences associated with pathological AD in these different types of genomic features by analyzing individual CpGs and DMRs separately

As pathological AD-associated genes can harbor both significant individual CpGs and significant DMRs, we performed a pathway analysis by considering the significant CpGs and DMRs jointly. The test of KEGG pathways showed that hematopoietic cell lineage, phagosome, Cytokine-cytokine receptor interaction, and chemokine signaling pathways were significantly enriched with methylation differences in pathological AD at 5% FDR (Table 4).

(4) In Results – Under “Prioritizing significant DNA methylation differences with sample matching”

DNA methylation levels are known to be influenced by aging, which is also the strongest risk factor for AD. To prioritize significant methylation differences in pathological AD and minimize confounding

effects due to aging, we also explored an alternative strategy by matching each **pathological AD** case with a control subject of the same **sex and age at death in the same cohort**.

(5) In Discussion – 1st paragraph

We conducted a comprehensive meta-analysis of four cohorts of prefrontal cortex brain samples to prioritize consistent DNA **methylation differences** involved in **pathological AD**.

(6) In Discussion – 5th paragraph

In **pathological AD** samples, our meta-analysis identified significant hypo-methylation in the promoter region of the *AZU1* gene, also known as *CAP37*, which is a neutrophil granule protein that helps defend the host against microbial pathogens and regulate inflammation².

Taken together, these results demonstrated our meta-analysis replicated methylation differences in a number of genes previously implicated in **pathological AD**, as well as nominated novel genes likely to be involved in AD pathogenesis.

(7) In Discussion – 6th paragraph

As in previous single cohort studies^{1, 3-5}, we observed that the majority of the significant DNA **methylation differences** were hyper-methylated in **pathological AD**. Our enrichment analysis brought to light the potential roles of these hyper-methylations in regulating active and repressive elements in **pathological AD**.

(8) In Discussion – 8th paragraph

When we examined PRC2 target genes⁶, we found many of these genes had low normalized gene expression levels in both **pathological AD** cases (**Braak stage 3-6**) and controls (**Braak stage 0 -2**) in the ROSMAP samples (Supplementary Table 22), consistent with recent studies that have suggested DNA methylation and PRC2 often complement each other in gene silencing⁷⁻⁹.

(9) In Discussion – 9th paragraph

The results of our pathway analysis overwhelmingly point to immune system alterations in **pathological AD**.

2. Why did the authors only consider Braak? Why not consider other phenotypes such as CERAD?

To clarify, in this meta-analysis we analyzed four different DNA methylation datasets and Braak stage was the only Alzheimer's disease measurement that was available in all four of the public datasets (GEO: GSE80970, GEO: GSE59685, GEO: GSE66351, Synapse: syn3157275).

3. As the authors analyzed brain samples as well as blood sample, it is better to be specific about age. Please point out whether age means age at death or age at blood draw. Please check the main text and the tables.

This reviewer raised an important point. In response, we have made revisions throughout the manuscript to distinguish age at death and age at blood draw.

Revisions

(1) In Results, 1st paragraph

Among the four cohorts, the mean **age at death** ranged from 73.6 years to 86.3 years, and the percentage of **females** ranged from 51.8% to 63.5%.

(2) In Results, under “Meta-analysis identified **methylation differences** significantly associated with AD Braak stage at individual CpGs and co-methylated genomic regions”, 1st and 2nd paragraph

Adjusting for **estimated** cell-type proportions (i.e., the proportion of neurons), **age at death**, sex, and batch effects

In the coMethDMR approach, we tested 40,010 pre-defined genomic regions to identify co-methylated and differentially methylated regions associated with Braak stage, adjusting for estimated **neuron** proportions, **age at death**, sex, and batch effects for each cohort separately.

(3) In Results, under “Prioritizing significant DNA methylation differences with sample matching”

To prioritize significant **methylation differences** in **pathological** AD and minimize confounding effects due to aging, we also explored an alternative strategy by matching each **pathological** AD case with a control subject of the same **sex and age at death in the same cohort**.

We identified a total of **151** CpGs and **32** DMRs that were significantly different after matching cases and control samples by **sex and age at death** (Supplementary Tables 6 and 7).

(4) In Results, under “Correlation of methylation levels of significant CpGs and DMRs in AD with expressions of nearby genes”, 1st paragraph

To reduce the effect of potential confounding effects, when testing for methylation-gene expression associations, we **first adjusted for age at death**, sex, cell-type proportions, and batch effects in both DNA methylation and gene expression levels separately and extracted residuals from the linear models.

(5) In Discussion, 2nd paragraph

Fourth, as an alternative strategy to reduce confounding by **sex and age at death**, we also employed a matching design that matched each case with a control sample of the same **sex and age at death**.

(6) In Table 1

Table 1 Sample characteristics of the brain and blood cohorts included in the meta-analysis. Shown are numbers (and percentages) of samples after quality control.

Dataset	Tissue	Sample size	Women N (%)	Cases N (%)	Age at Death Mean (SD)	Accession
Brain samples cohort						
(1) ROSMAP cohort	PFC	726	461 (63.5%)	581 (80.0%)	86.3 (4.8)	Synapse: syn3157275
(2) Mt. Sinai cohort	PFC	141	88 (62.4%)	85 (60.3%)	85.8 (7.8)	GEO: GSE80970
(3) London cohort	PFC	107	64 (59.8%)	80 (74.8%)	84.6 (9.0)	GEO: GSE59685
(4) Gasparoni et al. (2018)	PFC	56	29 (51.8%)	36 (64.3%)	73.6 (14.7)	GEO: GSE66351
Blood samples cohort						
(5) London cohort	whole blood	69	44 (63.8%)	59 (85.5%)	83.6 (6.2)	GEO: GSE59685

PFC = prefrontal cortex

(7) Legend for Suppl. Table 7

Supplementary Table 7 In samples matched by sex and **age at death** within the same cohort, a total of 32 DMRs were significantly associated with AD Braak stage at 5% FDR in meta-analysis of three brain samples cohorts (London, Mt. Sinai, ROSMAP).

4. For the meta-analysis, I think it is better to use ‘fixed-effect’ and ‘random-effects’. Please check your results section. Also, why don’t you just use random-effects analysis, no matter whether there is heterogeneity or not because you lose little if there is no heterogeneity.

This reviewer had a good point. Another reviewer also had a similar comment and suggested we use the “inverse variance-weighted fixed effects model” instead. We have updated the manuscript accordingly. With regard to fixed effects vs. random effects models, because the association signals in these brain datasets were relatively weak, we wanted to use a model with as much power as is justifiable. As the figure on the right shows, we do gain a little more power using the fixed effects model for meta-analysis.

Revision – In Results, 3rd paragraph

Next, we combined the cohort-specific P-values for these genomic regions using an inverse-variance fixed effects meta-analysis.

(2) In Methods, under section “Meta-analysis”, 2nd paragraph

The **inverse-variance weighted fixed effects model** was applied to synthesize statistical significance from individual cohorts.

5. More information is needed about the blood sample of the London cohort. For example, when was the blood drawn? How long was it from the blood draw to death?

To clarify, the blood was drawn pre-mortem. The average time from blood draw to death was about 3.81 years, with a range of 0 to 10 years. In response, we additionally included these details in the manuscript.

Revision – In Results, under “Correlation of AD-associated CpGs and DMRs methylation levels in blood and brain samples”, 1st paragraph.

The difference between age at pre-mortem blood draw and age at death ranged from 0 to 10 years, with an average of 3.81 ± 2.61 years.

6. A relevant comment is that the relationship between methylation in blood and brain has already been studied using data from ROSMAP (Yu et al., Methylation profiles in peripheral blood CD4+ lymphocytes versus brain: the relation to Alzheimer's disease pathology, 2016). Can the author compare the findings?

In response to this reviewer's comment, we studied the paper “Methylation profiles in peripheral blood CD41 lymphocytes versus brain: The relation to Alzheimer's disease pathology” by Yu et al. (2016). Because of the small sample size (N = 41 sets of matched brain-blood samples), the authors reported global results on brain-blood correlations, but did not report brain-blood correlations for any specific CpG. More specifically, Yu et al. (2016) reported the correlations for T2 CD4+ lymphocyte and brain were modest with mean Pearson $r = 0.06$, $SD = 0.18$. ($SD =$ standard deviation)

In response to this reviewer's comment, we compared the global features of our blood-brain correlations with those in Yu et al. (2016). Our results showed the average brain-blood correlation in London samples is 0.069, with standard deviation of 0.165, which is remarkably close to those reported in Yu et al. (2016), even though the London cohort samples were whole blood samples and theirs were CD4+ lymphocytes. We added these results and comparison with Yu et al. (2016) in the manuscript.

Revision – in Results, under “Correlation of AD-associated CpGs and DMRs methylation levels in blood and brain samples”

The correlation between methylation levels in brain and blood were modest at the majority of CpG sites (mean Pearson $r = 0.069$, $SD = 0.165$), which is similar to those reported in Yu et al. (2016)^{10,11} for CD4+ lymphocytes and other previous reports^{10,11}.

7. Another question I have is why the authors didn't exclude CpGs that are associated with smoking or overlapping with cross-reactive probes before the analysis? How did you determine smoking-related CpGs?

To clarify, we determined smoking-related CpGs based on a recent meta-analysis of DNA methylation changes associated with smoking (Joehanes, R. et al. *Epigenetic Signatures of Cigarette Smoking*. *Circ Cardiovasc Genet* 9, 436-447 (2016) PMID: 27651444). In response to this reviewer's comment, we added more clarifications on how smoking-related CpGs were determined.

With regard to whether excluding CpGs before or after the analysis, they produce the same final results, but excluding CpGs after analysis provided us the opportunity to examine these additional CpGs.

Revision

(1) In Methods, under “Meta-analysis”, 4th paragraph

Finally, we excluded CpGs and DMRs that overlapped with any of the 2,623 CpGs associated with smoking identified in Joehanes et al. (2016)¹² or any of the cross-reactive probes identified in Chen et al. (2013)¹³.

(2) In Results, under “Meta-analysis identified methylation differences significantly associated with AD Braak stage at individual CpGs and co-methylated genomic regions”

After eliminating CpGs associated with smoking¹² or overlapping with cross-reactive probes¹³, we obtained 3,751 significant CpGs (Supplementary Table 1)

After eliminating those DMRs containing cross-reactive probes¹³ or smoking-associated probes¹², we obtained 119 co-methylated DMRs at 5% FDR (Supplementary Table 2).

8. It seems that the authors did not consider postmortem interval (PMI) in the analysis. PMI can affect DNA methylation.

This reviewer raised an important point. In response, we re-run our analysis for the ROSMAP cohort, by additionally including PMI in model methylation $M \text{ value} \sim \text{neuron.proportions} + \text{batch} + \text{array} + \text{ageAtDeath} + \text{sex} + \text{PMI}$. The Bland-Altman plot in the figure below shows the estimated effect sizes using a model that additionally adjust for PMI are very similar to those obtained from the model not adjusting for PMI. In particular, in individual CpGs analysis, the 95% confidence interval for the absolute difference in the effect sizes from these two models is $(-1.31 \times 10^{-3}, 1.50 \times 10^{-3})$.

In addition, in the ROSMAP cohort, PMI was not significantly associated with Braak stage ($r = -0.018$, $p\text{-value} = 0.6156$), so PMI is unlikely to be a confounder for Braak stage.

Also, we did not have PMI information in the other three public datasets (London, Mount Sinai, Gasporoni). Therefore, we decided not to include PMI in our final analysis.

In response to this reviewer's comment, we added clarifications on our rational for not including PMI in the Discussion section.

Revision – In Discussion, 2nd from last paragraph

In this study, we did not adjust for post-mortem interval (PMI), because in the ROSMAP cohort PMI was not significantly associated with Braak stage (Spearman correlation $r = -0.018$, P -value = 0.6156), so is unlikely to be a confounder for it. Also, PMI was not available for the other three public datasets (London, Mount Sinai, Gasporoni) we analyzed.

9. In the section 'Prioritizing significant DNA methylation changes with sample matching', the authors matched based on age (I think they mean age at death). Is this one-to-one matching? What if there are multiple matches for a person with a specific age at death in the case group? The authors need to be specific about how they handled matching. Also, matching based solely on age (at death) does not seem to be sufficient. I think sex and education should be at least considered.

We thank this reviewer for the good suggestion. We agree that it would be better to match on age at death as well as sex and education. In response, we first revised the matching algorithm to match case samples with control samples by age at death, sex and education in the ROSMAP cohort. But we encountered several difficulties (see details below), therefore, in the end we decided to only match samples by age at death and sex.

There were several issues when matching samples by education, in addition to age at death and sex:

(1) We would lose a significant number of samples ($n = 68$) when additionally match by education (in years). For ROSMAP cohort, when matching by age at death and sex, we obtained a sample size of 244. When matching by age at death, sex and education, we obtained a sample size of 176.

(2) In the ROSMAP cohort, education was not associated with Braak stage ($r = -0.063$, p -value = 0.0882). Therefore, education was unlikely to be a confounder for Braak stage.

(3) For the four public datasets we analyzed in this study, the other three datasets (London, Mount Sinai, Gasporoni) did not have education information.

In response to this reviewer's comment, we re-run our matched samples analysis by matching each AD case with a control subject with both the same sex and age at death in the same cohort. We also added clarifications on the details of the matching algorithm in the Methods section. See details below.

Revision:

(1) Supplementary Table 6 and 7 - Updated analysis results after matching samples by both sex and age.

Supplementary Table 6 In samples matched by sex and age at death within the same cohort, a total of 151 differentially methylated CpGs were significantly associated with the AD Braak stage at 5% FDR in the meta-analysis of three brain samples cohorts (London, Mt. Sinai, ROSMAP). For each CpG, annotations

include the location of the CpG based on hg19/GRCh37 genomic annotation (Chr, Position), nearby genes based on Illumina gene annotations (UCSC_RefGene_Accession, UCSC_RefGene_Name), the type of associated genomic feature (UCSC_RefGene_Group), and location with respect to CpG islands (Relation_to_Island). Meta-analysis results include estimated effect size (estimate), where CpGs that are hyper-methylated in AD have positive values; standard error of the estimates (se); p-value from inverse-variance fixed effects model (pVal.fixed); p-value from inverse-variance random effects model (pVal.random); p-value for heterogeneity (pVal.Q); the direction of effects in individual cohorts of London, Mount Sinai, and ROSMAP (Estimate_Direction), where + indicates hyper-methylation in AD and - indicates hypo-methylation in AD in an individual cohort; the final meta-analysis p-value (pVal.final), which is based on pVal.fixed if pVal.Q < 0.05 and otherwise on pVal.random, and false discovery rate (fdr). Analysis results for individual cohorts include estimated effect size (estimate), standard error (se), and p-value (pValue) for each cohort.

Supplementary Table 7 In samples matched by sex and age at death within the same cohort, a total of 32 DMRs were significantly associated with AD Braak stage at 5% FDR in meta-analysis of three brain samples cohorts (London, Mt. Sinai, ROSMAP). For each DMR, annotations include location of the DMR based on hg19/GRCh37 genomic annotation (DMR), nearby genes based on Illumina gene annotations (UCSC_RefGene_Accession, UCSC_RefGene_Name), the type of associated genomic feature (UCSC_RefGene_Group), and location with respect to CpG islands (Relation_to_Island). Meta-analysis results include estimated effect size (estimate), where DMRs that are hyper-methylated in AD have positive values; standard error of the estimates (se); p-value from inverse-variance fixed effects model (pVal.fixed); p-value from inverse-variance random effects model (pVal.random); p-value for heterogeneity (pVal.Q); direction of effects in individual cohorts of London, Mount Sinai, and ROSMAP (Estimate_Direction), where + indicates hyper-methylation in AD and - indicates hypo-methylation in AD in an individual cohort; the final meta-analysis p-value (pVal.final), which is based on pVal.fixed if pVal.Q < 0.05 and otherwise on pVal.random; and false discovery rate (fdr). CoMethDMR analysis results for individual cohorts include the number of CpGs in co-methylated cluster (nCpGs), estimated effect size (estimate), standard error (se), and p-value (pValue) for each cohort.

(2) In Methods, 5th paragraph under “Meta-analysis”

To prioritize methylation changes most likely to be affected by the AD pathogenesis process, we also performed an analysis using an alternative strategy to control for confounding effects for age and sex. More specifically, we first matched each case with control samples with the same age at death (in years) and sex in the same cohort using the matchControls function in the e1071 R package. When there are multiple matched control samples, the control sample with the most similar age as the case sample is selected. The age and sex matched samples were then analyzed in the same way as described above, except for removing age at death and sex effects in the linear models.

10. I think some of the results should not be included in the results section. They are more appropriate for the discussions section. For example, from paragraph ‘Our meta-analysis of individual CpGs identified many of the loci’... to the paragraph ‘In addition to corroborating previous findings, ...’ Please consider revision.

We appreciate this reviewer’s suggestion for moving some of the text in the Results to the Discussion section. In response, we have moved these paragraphs to the Discussion section.

Revision – In Discussion, 3rd, 4th and 5th paragraphs were moved from Results section

11. The paragraph before the discussion section. More information about genetic data from ROSMAP is needed. For example, are the genetic data imputed data? If so, were they imputed based on 1K genome or HRC? There are two batches of genetic data from ROSMAP. Did you use both batches? If so, you should control for the batch effect (of the genetic data). Also, did you control for any principal components?

To clarify, the genetic data from ROSMAP was imputed based on HRC. We agree with this reviewer that PCs and batch effects should also be modeled in our mQTL analysis. In response, we re-ran our mQTL analysis additionally including batch and first three PCs from genetic data. We also added clarification about imputation for ROSMAP genetic data in the Results section.

Revisions

(1) In Results, 2nd paragraph under “Correlation and co-localization with genetic susceptibility loci”

To identify methylation quantitative trait loci (mQTLs) for the significant DMRs and CpGs, we tested associations between the methylation levels with nearby SNPs, using the ROSMAP study dataset (imputed to the Haplotype Reference Consortium r1.1 reference panel)¹⁴, which had matched genotype data and DNA methylation data for 688 samples. To reduce the number of tests, we focused on identifying *cis* mQTLs located within 500kb from the start or end of the DMR (or position of the significant CpG)¹⁵. Among 166,797 SNPs that are associated with AD, 11,670 were also significantly associated with methylation levels, after correcting for confounding effects age, sex, cell type, **batch effects and PCs in methylation data**. Among the 119 DMRs and 3,751 CpGs significantly associated with Braak stage, 37 DMRs and 1,010 CpGs had at least one corresponding mQTL in brain samples, respectively (Supplementary Tables 11 and 12).

(2) In Methods, under “Correlation and co-localization with genetic susceptibility loci”

We then fit the linear model **methylation residual ~ SNP dosage + batch + PC1 + PC2 + PC3**, where **PC1, PC2, and PC3 are the first three PCs estimated from genotype data**, to test the association between methylation residuals in CpGs and the imputed allele dosages for SNPs to identify mQTLs.

(3) Supplementary Table 11 and 12 – updated results for mQTL analysis, additionally including batch and first three PCs estimated from genetic data.

Supplementary Table 11 Among the 3751 FDR significant Braak-associated CpGs, 1010 CpGs had at least one corresponding mQTL in the prefrontal cortex brain samples. The mQTL analysis was performed using the ROSMAP cohort samples with matched genotype data and DNA methylation data for 688 samples. *cis* mQTLs located within 500kb from the CpGs were considered. The genotype data was imputed to HRC r1.1 reference panel and tested against AD status using logistic regression adjusting for age, sex, and first three PCs estimated from genotype data. AD status of the samples was determined using clinical consensus diagnosis of cognitive status at the time of death (cases: variable cogdx = 4, 5, controls: others). Logistic regression results include reference allele (REF), alternative allele (ALT), minor and tested allele in the regression model (A1) as well as frequencies associated with it in cases and controls (A1_FREQS_cases, A1_FREQS_controls), odds ratio which is exp (beta estimate for A1 allele) (OR), standard error of beta estimate for A1 allele (SE), Z-value for test statistic (Z_STAT), and p-value (Pval). mQTLs were identified by fitting a linear model with methylation residuals (adjusted for neuron.proportions, DNAm.batch, array, ageAtDeath, and sex) as outcome variable, SNP dosage, batch, and the first three PCs from genotype data as independent variables. mQTL analysis results include estimated effect size (Estimate_SNP), standard error of the estimate (StdError_SNP), p-value (P_SNP), and

FDR obtained from the linear model. Annotations include the location of the SNP (SNP), identifier and location of the CpG (CpG, chr, position), the distance between SNP and CpG (SNP-CpGdistance), and nearby gene based on Illumina annotation (UCSC_RefGene_Name).

Supplementary Table 12 Among the 119 FDR significant DMRs, 37 DMRs had at least one corresponding mQTL in the brain samples. The mQTL analysis was performed using the ROSMAP cohort samples with matched genotype data and DNA methylation data for 688 samples. *cis* mQTLs located within 500kb from the start or end of the DMRs were considered. The genotype data was imputed to HRC r1.1 reference panel and tested against AD status using logistic regression adjusting for age, sex, and first three PCs estimated from genotype data. AD status of the samples was determined using clinical consensus diagnosis of cognitive status at the time of death (cases: variable cogdx = 4, 5, controls: others). Logistic regression results include reference allele (REF), alternative allele (ALT), minor and tested allele in the regression model (A1) as well as frequencies associated with it in cases and controls (A1_FREQS_cases, A1_FREQS_controls), odds ratio which is exp(beta estimate for A1 allele) (OR), standard error of beta estimate for A1 allele (SE), Z-value for test statistic (Z_STAT), and p-value (Pval). Each DMR was summarized using median methylation residuals of all CpGs located within the DMR first, then mQTLs were identified by fitting a linear model with median methylation residuals (adjusted for neuron.proportions, DNAm.batch, array, ageAtDeath and sex) as outcome variable, SNP dosage, batch, and first three PCs from genotype data as independent variables. mQTL analysis results include estimated effect size (Estimate_SNP), standard error of the estimate (StdError_SNP), p-value (P_SNP), and FDR obtained from the linear model. Annotations include the location of the SNP (SNP), identifier and location of the CpG (CpG, chr, position), the distance between SNP and CpG (SNP-CpGdistance), and nearby gene based on Illumina annotation (UCSC_RefGene_Name).

12. For many tables, more tables notes are needed. For example, what does the '++++' and '----' symbol mean in the column 'direction'?

In response to this reviewer's comment, we have added extensive descriptions for the tables and figures legends. Please see details below.

Revision –

(1) Table 2 The last column (Estimate_Direction) indicates the direction of effects in Gasporoni, London, Mount Sinai, and ROSMAP cohorts where + indicates hyper-methylation in AD and - indicates hypo-methylation in AD in an individual cohort.

(2) Supplementary Table 1 direction of effects in individual cohorts of Gasporoni, London, Mount Sinai, and ROSMAP (Estimate_Direction), where + indicates hyper-methylation in AD and - indicates hypo-methylation in AD in an individual cohort

(3) Table and Figure legends were completely updated to include more descriptions.

Table 1 Sample characteristics of the brain and blood cohorts included in the meta-analysis. Shown are numbers (and percentages) of samples after quality control.

Table 2 Top 20 most significant differentially methylated CpGs associated with Braak stage in meta-analysis. For each CpG, annotations include the location of the CpG based on hg19/GRCh37 genomic annotation (Chr, Position), nearby genes based on GREAT, and Illumina gene annotations. Meta-analysis results include estimated effect size (Estimate) where CpGs that are hyper-methylated in AD have positive values, P-value, and false discovery rate (FDR). The last column (Estimate_Direction) indicates the

direction of effects in Gasporoni, London, Mount Sinai, and ROSMAP cohorts where + indicates hyper-methylation in AD and - indicates hypo-methylation in AD in an individual cohort.

Table 3 Top 20 most significant differentially methylated regions (DMRs) associated with Braak stage identified by both coMethDMR and comb-p in meta-analysis. For each DMR, annotations include location of the DMR based on hg19/GRCh37 genomic annotation (Region), nearby genes based on GREAT, and Illumina gene annotations. Meta-analysis results based on coMethDMR include the number of probes in the DMR (No. Probes), estimated effect size (Estimate) where DMRs that are hyper-methylated in AD have positive values, P-value, and false discovery rate (FDR). Meta-analysis results based on comb-p include multiple comparison corrected P-value based on Sidak method (Sidak P-value). The last column (Estimate_Direction) indicates the direction of DMR effects estimated by coMethDMR in Gasporoni, London, Mount Sinai, and ROSMAP cohorts where + indicates hyper-methylation in AD and - indicates hypo-methylation in AD in an individual cohort.

Table 4 Gene set enrichment analysis of significant methylation differences associated with AD Braak stage identified in meta-analysis. Shown are gene ontology or KEGG database ID (Gene Set), a description of the pathway (Description), and significance assessment (P-value, FDR).

Figure 1 Workflow of meta-analysis for individual CpGs and DMRs.

Figure 2 Manhattan plot of significant methylation differences in individual CpGs and DMRs identified in meta-analysis. The X-axis indicates chromosomes 1-22 and the Y-axis indicates $-\log_{10}$ (P-value), with the horizontal red line indicating a 5% FDR (false discovery rate) significance level.

Figure 3 Enrichment of CpGs significantly associated with AD Braak stage in meta-analysis of individual CpGs and DMRs at 5% FDR. A two-sided Fisher's test was used to determine over or under-representation of the significant CpGs in individual CpGs analysis and CpGs mapped within significant DMRs in various (a) (b) genomic features and (c) (d) chromatin states. *** indicates P-value < 0.001, ** indicates P-value < 0.01 and * indicates P-value < 0.05.

Supplementary Tables and Figures

Supplementary Table 1 A total of 3751 differentially methylated CpGs were significantly associated with the AD Braak stage at 5% FDR in the meta-analysis of four brain samples cohorts (Gasparoni, London, Mt. Sinai, ROSMAP). For each CpG, annotations include the location of the CpG based on hg19/GRCh37 genomic annotation (chr, position), nearby genes based on GREAT (GREAT_annotation) and Illumina gene annotations (UCSC_RefGene_Accession, UCSC_RefGene_Name), the type of associated genomic feature (UCSC_RefGene_Group), location with respect to CpG islands (Relation_to_Island), and chromatin states (state). Meta-analysis results include estimated effect size (estimate), where CpGs that are hyper-methylated in AD have positive values; standard error of the estimates (se); p-value from inverse-variance fixed effects model (pVal.fixed); p-value from inverse-variance random effects model (pVal.random); p-value for heterogeneity (pValQ); direction of effects in individual cohorts of Gasporoni, London, Mount Sinai, and ROSMAP (Estimate_Direction), where + indicates hyper-methylation in AD and - indicates hypo-methylation in AD in an individual cohort; the final meta-analysis p-value (pVal.final), which is based on pVal.fixed if pValQ < 0.05 and otherwise on pVal.random; and false discovery rate (fdr). Analysis results for individual cohorts include estimated effect size (estimate), standard error (se) and p-value (pValue) for each cohort. Comparison with literature includes variables that indicate (1 = "yes", 0 = "no") if a CpG was previously described in primary publications of each individual cohort (references PMIDs: 25129075, 25129077, 29550519, 30045751) that contributed to this meta-analysis, with the last column (isAnyPreviousStudy) indicating if a CpG was described in any previous single cohort publications.

Supplementary Table 2 A total of 119 co-methylated DMRs were significantly associated with AD Braak stage, which were identified by both coMethDMR (at 5% FDR) and comb-p (at 5% Sidak adjusted p-value) methods in meta-analysis of four brain samples cohorts (Gasparoni, London, Mount Sinai, ROSMAP). For each DMR, annotations include location of the DMR based on hg19/GRCh37 genomic annotation (DMR), nearby genes based on GREAT (GREAT_annotation) and Illumina gene annotations (UCSC_RefGene_Accession, UCSC_RefGene_Name), the type of associated genomic feature (UCSC_RefGene_Group), location with respect to CpG islands (Relation_to_Island), and chromatin states (state). Meta-analysis results include estimated effect size (estimate), where DMRs that are hyper-methylated in AD have positive values; standard error of the estimates (se); p-value from inverse-variance fixed effects model (pVal.fixed); p-value from inverse-variance random effects model (pVal.random); p-value for heterogeneity (pValQ); direction of effects in individual cohorts of Gasparoni, London, Mount Sinai, and ROSMAP (Estimate_Direction), where + indicates hyper-methylation in AD and - indicates hypo-methylation in AD in an individual cohort; the final meta-analysis p-value (pVal.final), which is based on pVal.fixed if pValQ < 0.05 and otherwise on pVal.random; and false discovery rate (fdr). CoMethDMR analysis results for individual cohorts include the number of co-methylated clusters found in the input region (nCoMethRegion), most significant co-methylated cluster (coMethRegion), number of CpGs in this co-methylated cluster (nCpGs), estimated effect size (estimate), standard error (se), and p-value (pValue) for each cohort. Comb-p analysis results include location of the DMR (combp_chrom, combp_start, combp_end), minimum p-value of individual CpGs within the DMR (combp_min_p), number of probes in DMR (combp_n_probes), p-value (combp_z_p), and Sidak adjusted p-value (combp_z_sidak_p). Column AU (Probes_single_cpg_analysis_fdr_0_05) includes differentially methylated CpGs with 5% or less FDR within each co-methylated DMR. Comparison with literature includes variables that indicate (1 = “yes”, 0 = “no”) if a DMR was previously described in primary publications of each individual cohort (references PMIDs: 25129075, 25129077, 29550519, 30045751) that contributed to this meta-analysis, with the last column (isAnyPreviousStudy) indicating if a DMR was described in any previous single cohort publications.

Supplementary Table 3 Enrichment analysis of 119 FDR significant DMRs and 3751 FDR significant differentially methylated CpGs identified in meta-analysis in different types of genomic features. A two-sided Fisher’s exact test was used to determine over- or under-representation of the significant CpGs or regions in each type of genomic feature. Shown is the type of genomic feature (Feature), odds ratio that compares the odds that a significant CpG (or DMR) is mapped to a particular type of genomic feature with the odds that a background CpG (or DMR) is associated with the particular type of genomic feature (OR), and p-value from Fisher’s exact test (p-value). Shown in bold are significant enrichment results.

Supplementary Table 4 Enrichment analysis of 119 FDR significant DMRs and 3751 FDR significant differentially methylated CpGs identified in meta-analysis in different types of chromatin states. A two-sided Fisher’s exact test was used to determine over- or under-representation of the significant CpGs or regions in each type of chromatin state. Shown is the type of chromatin state (Feature), odds ratio that compares the odds that a significant CpG (or DMR) is mapped to a particular type of chromatin state with the odds that a background CpG (or DMR) is associated with the particular type of chromatin state (OR), and p-value from Fisher’s exact test (p-value). Shown in bold are significant enrichment results.

Supplementary Table 5 Enrichment of FDR significant differentially methylated CpGs in meta-analysis in binding sites of transcription factors and chromatin proteins assayed by ENCODE project. This analysis was performed using LOLA (Locus Overlap Analysis) software. Annotations include the antibody for the protein (Antibody), cell type for the ChIPseq experiment (CellType), Description, and name of the file from ENCODE database used for analysis. Results include the odds ratio (OR), p-value from Fisher’s exact test (pValue), and false discovery rate (FDR).

Supplementary Table 6 In samples matched by sex and age at death within the same cohort, a total of 151 differentially methylated CpGs were significantly associated with the AD Braak stage at 5% FDR in the meta-analysis of three brain samples cohorts (London, Mt. Sinai, ROSMAP). For each CpG, annotations include the location of the CpG based on hg19/GRCh37 genomic annotation (Chr, Position), nearby genes based on Illumina gene annotations (UCSC_RefGene_Accession, UCSC_RefGene_Name), the type of associated genomic feature (UCSC_RefGene_Group), and location with respect to CpG islands (Relation_to_Island). Meta-analysis results include estimated effect size (estimate), where CpGs that are hyper-methylated in AD have positive values; standard error of the estimates (se); p-value from inverse-variance fixed effects model (pVal.fixed); p-value from inverse-variance random effects model (pVal.random); p-value for heterogeneity (pValQ); the direction of effects in individual cohorts of London, Mount Sinai, and ROSMAP (Estimate_Direction), where + indicates hyper-methylation in AD and - indicates hypo-methylation in AD in an individual cohort; the final meta-analysis p-value (pVal.final), which is based on pVal.fixed if pValQ < 0.05 and otherwise on pVal.random, and false discovery rate (fdr). Analysis results for individual cohorts include estimated effect size (estimate), standard error (se), and p-value (pValue) for each cohort.

Supplementary Table 7 In samples matched by sex and age at death within the same cohort, a total of 32 DMRs were significantly associated with AD Braak stage at 5% FDR in meta-analysis of three brain samples cohorts (London, Mt. Sinai, ROSMAP). For each DMR, annotations include location of the DMR based on hg19/GRCh37 genomic annotation (DMR), nearby genes based on Illumina gene annotations (UCSC_RefGene_Accession, UCSC_RefGene_Name), the type of associated genomic feature (UCSC_RefGene_Group), and location with respect to CpG islands (Relation_to_Island). Meta-analysis results include estimated effect size (estimate), where DMRs that are hyper-methylated in AD have positive values; standard error of the estimates (se); p-value from inverse-variance fixed effects model (pVal.fixed); p-value from inverse-variance random effects model (pVal.random); p-value for heterogeneity (pValQ); direction of effects in individual cohorts of London, Mount Sinai, and ROSMAP (Estimate_Direction), where + indicates hyper-methylation in AD and - indicates hypo-methylation in AD in an individual cohort; the final meta-analysis p-value (pVal.final), which is based on pVal.fixed if pValQ < 0.05 and otherwise on pVal.random; and false discovery rate (fdr). CoMethDMR analysis results for individual cohorts include the number of CpGs in co-methylated cluster (nCpGs), estimated effect size (estimate), standard error (se), and p-value (pValue) for each cohort.

Supplementary Table 8 Comparison of the brain and blood DNA methylation levels using London cohort samples and BeCon software. For the London cohort with 69 pairs of matched brain and blood samples, analysis results using beta values include Spearman correlation between brain and blood beta values (Spearman_cor), p-value for the correlation (pVal), and false discovery rate (fdr). The analysis using residuals was performed by first adjusting methylation M-values in brain and blood samples separately for estimated neuron proportions for brain samples (or estimated blood cell type proportions), array, age at death (for brain samples) or age at blood draw (for blood samples), and sex; extracting residuals from the linear models; and then computing Spearman correlation (spearman_cor), p-values (pVal), and false discovery rate (fdr) on the residuals obtained for brain and blood samples. BeCon analysis results include correlation coefficients (cor_BA10) computed on 16 subjects with a matched brain (Broadmann area 10) sample and blood samples, as described in Edger et al. (2017). Annotations include nearby genes based on Illumina gene annotations (UCSC_RefGene_Accession, UCSC_RefGene_Name), the type of associated genomic feature (UCSC_RefGene_Group), location with respect to CpG islands (Relation_to_Island) and chromatin state (state).

Supplementary Table 9 Association between methylation levels at Braak-associated DMRs with nearby genes. Each DMR is linked to genes located in the vicinity (\pm 250 kb). First, each DMR is summarized by the median of CpG methylation M-values over all CpGs mapped within the DMR. The median methylation M-values and normalized gene expression values are then adjusted for age at death, sex, cell type, and batch

effects separately. Next, the residuals from these linear models are extracted. Finally, a separate linear model is used to test association between methylation residuals and gene expression residuals, adjusting for Braak stage. Shown are: the name of the most significant gene associated with each DMR (Gene), effect size (Estimate), p-value (pVal), and false discovery rate (FDR) for the regression model correlating methylation residuals and gene expression residuals. Annotations include the type of associated genomic feature (UCSC_RefGene_Group), location with respect to CpG islands (Relation_to_Island), and chromatin state (Chromatin state).

Supplementary Table 10 Association between methylation levels at Braak-associated CpGs with nearby genes. Each CpG is linked to genes located in the vicinity (± 250 kb from the start or end of the DMR). The methylation M-values and normalized gene expression values are first adjusted for age at death, sex, cell type, and batch effects separately. Next, the residuals from these linear models are extracted. Finally, a separate linear model is used to test the association between methylation residuals and gene expression residuals, adjusting for Braak stage. Shown are: the name of the most significant gene associated with each CpG (Gene), effect size (Estimate), p-value (pVal), and false discovery rate (FDR) for the regression model correlating methylation residuals and gene expression residuals. Annotations include the type of associated genomic feature (UCSC_RefGene_Group), location with respect to CpG islands (Relation_to_Island), and chromatin state (Chromatin state).

Supplementary Table 11 Among the 3751 FDR significant Braak-associated CpGs, 1010 CpGs had at least one corresponding mQTL in the prefrontal cortex brain samples. The mQTL analysis was performed using the ROSMAP cohort samples with matched genotype data and DNA methylation data for 688 samples. *cis* mQTLs located within 500kb from the CpGs were considered. The genotype data was imputed to HRC r1.1 reference panel and tested against AD status using logistic regression adjusting for age, sex, and first three PCs estimated from genotype data. AD status of the samples was determined using clinical consensus diagnosis of cognitive status at the time of death (cases: variable cogdx = 4, 5, controls: others). Logistic regression results include reference allele (REF), alternative allele (ALT), minor and tested allele in the regression model (A1) as well as frequencies associated with it in cases and controls (A1_FREQS_cases, A1_FREQS_controls), odds ratio which is exp (beta estimate for A1 allele) (OR), standard error of beta estimate for A1 allele (SE), Z-value for test statistic (Z_STAT), and p-value (Pval). mQTLs were identified by fitting a linear model with methylation residuals (adjusted for neuron.proportions, DNAm.batch, array, ageAtDeath, and sex) as outcome variable, SNP dosage, batch, and the first three PCs from genotype data as independent variables. mQTL analysis results include estimated effect size (Estimate_SNP), standard error of the estimate (StdError_SNP), p-value (P_SNP), and FDR obtained from the linear model. Annotations include the location of the SNP (SNP), identifier and location of the CpG (CpG, chr, position), the distance between SNP and CpG (SNP-CpGdistance), and nearby gene based on Illumina annotation (UCSC_RefGene_Name).

Supplementary Table 12 Among the 119 FDR significant DMRs, 37 DMRs had at least one corresponding mQTL in the brain samples. The mQTL analysis was performed using the ROSMAP cohort samples with matched genotype data and DNA methylation data for 688 samples. *cis* mQTLs located within 500kb from the start or end of the DMRs were considered. The genotype data was imputed to HRC r1.1 reference panel and tested against AD status using logistic regression adjusting for age, sex, and first three PCs estimated from genotype data. AD status of the samples was determined using clinical consensus diagnosis of cognitive status at the time of death (cases: variable cogdx = 4, 5, controls: others). Logistic regression results include reference allele (REF), alternative allele (ALT), minor and tested allele in the regression model (A1) as well as frequencies associated with it in cases and controls (A1_FREQS_cases, A1_FREQS_controls), odds ratio which is exp (beta estimate for A1 allele) (OR), standard error of beta estimate for A1 allele (SE), Z-value for test statistic (Z_STAT), and p-value (Pval). Each DMR was summarized using median methylation residuals of all CpGs located within the DMR first, then mQTLs were identified by fitting a linear model with median methylation residuals (adjusted for

neuron.proportions, DNAm.batch, array, ageAtDeath and sex) as outcome variable, SNP dosage, batch, and first three PCs from genotype data as independent variables. mQTL analysis results include estimated effect size (Estimate_SNP), standard error of the estimate (StdError_SNP), p-value (P_SNP), and FDR obtained from the linear model. Annotations include the location of the SNP (SNP), identifier and location of the CpG (CpG, chr, position), the distance between SNP and CpG (SNP-CpGdistance), and nearby gene based on Illumina annotation (UCSC_RefGene_Name).

Supplementary Table 13 Overlap of the 3751 FDR significant Braak-associated CpGs with AD GWAS loci (LDblockGRCh37) reported in Kunkle et al. (2019). Annotations for CpGs include location of the CpG based on hg19/GRCh37 genomic annotation (Chr, Position), nearby genes based on Illumina gene annotations (UCSC_RefGene_Accession, UCSC_RefGene_Name), the type of associated genomic feature (UCSC_RefGene_Group), and location with respect to CpG islands (Relation_to_Island).

Supplementary Table 14 Bayesian co-localization of association signals from AD meta-analysis (Kunkle et al. (2019) PMID: 30820047) and ROSMAP mQTL study identified 2 GWAS nominated regions that included a single causal variant common to both traits (i.e. AD status and DNA methylation levels). Shown are GWAS nominated loci (AD_GWASloci), CpG and annotations (UCSC_RefGene_Name, UCSC_RefGene_Group), number of SNPs in the region (nsnps), estimated posterior probabilities for each of the co-localization hypotheses tested (PP3, PP4; Online Methods). In particular, PP3 quantifies support for the hypothesis of co-localization of two independent SNPs associated with the two traits, and PP4 quantifies support for the hypothesis of co-localization of one shared SNP associated with both traits. For each AD GWAS nominated region, shown are SNP with the highest estimated posterior probability for being the true causal variant for both traits (Best causal SNP), SNP with the lowest p-value for association with AD in Kunkle et al. (2019) (GWAS.SNP, GWAS.pval) and SNP with the lowest p-value for association with CpG methylation (mQTL.pval, mQTL.SNP).

Supplementary Table 15 After *bacon*-correction, a total of 2767 differentially methylated CpGs were significantly associated with the AD Braak stage at 5% FDR in meta-analysis of four brain samples cohorts (Gasparoni, London, Mt. Sinai, ROSMAP). For each CpG, annotations include the location of the CpG based on hg19/GRCh37 genomic annotation (chr, position), nearby genes based on GREAT (GREAT_annotation) and Illumina gene annotations (UCSC_RefGene_Accession, UCSC_RefGene_Name), the type of associated genomic feature (UCSC_RefGene_Group), location with respect to CpG islands (Relation_to_Island), and chromatin states (state). Meta-analysis results include estimated effect size (estimate), where CpGs that are hyper-methylated in AD have positive values; standard error of the estimates (se); p-value from inverse-variance fixed effects model (pVal.fixed); p-value from inverse-variance random effects model (pVal.random); p-value for heterogeneity (pValQ); the direction of effects in individual cohorts of Gasporoni, London, Mount Sinai, and ROSMAP (Estimate_Direction), where + indicates hyper-methylation in AD and - indicates hypo-methylation in AD in an individual cohort; the final meta-analysis p-value (pVal.final), which is based on pVal.fixed if pValQ < 0.05 and otherwise on pVal.random; and false discovery rate (fdr). Analysis results for individual cohorts include estimated effect size (Estimate), standard error (StdErr), and p-value (pValue) for each cohort. The last column (genome.wide.sig) indicates if genome-wide significance level at 2.4×10^{-7} was achieved (1= yes, 0 = no).

Supplementary Table 16 Sensitivity analysis results for testing enrichment of *bacon*-corrected differentially methylated CpGs significantly associated with AD Braak stage in different genomic features. The p-values for single CpGs were subjected to *bacon* correction first, then the significant CpGs were identified as those that reached 5% FDR significance (*FDR significant CpGs*; n = 2767) or those that reached genome-wide significance at 2.4×10^{-7} (*genome-wide significant CpGs*; n = 339). A two-sided Fisher's exact test was used to determine over- or under-representation of the significant CpGs in each type of genomic feature. Shown is the type of genomic feature (Feature), odds ratio that compares the odds that a significant CpG is mapped to a particular type of genomic feature with the odds that a background CpG

is associated with the particular type of genomic feature (OR), and p-value from Fisher's exact test (p-value). Shown in bold are significant enrichment results.

Supplementary Table 17 Sensitivity analysis results for testing enrichment of *bacon*-corrected differentially methylated CpGs significantly associated with AD Braak stage in different chromatin states. The p-values for single CpGs were subjected to *bacon* correction first, then the significant CpGs were identified as those that reached 5% FDR significance (*FDR significant CpGs*; n = 2767) or those that reached genome-wide significance at 2.4×10^{-7} (*genome-wide significant CpGs*; n = 339). A two-sided Fisher's exact test was used to determine over- or under-representation of the significant CpGs in each type of chromatin state. Shown is the type of genomic feature (Feature), odds ratio that compares the odds that a significant CpG is mapped to a particular type of genomic feature with the odds that a background CpG is associated with the particular type of genomic feature (OR), and p-value from Fisher's exact test (p-value). Shown in bold are significant enrichment results.

Supplementary Table 18 Sensitivity analysis results for testing enrichment of *bacon*-corrected differentially methylated CpGs significantly associated with AD Braak stage in binding sites of ENCODE transcription factors and chromatin proteins. The p-values for single CpGs were subjected to *bacon* correction first, then the significant CpGs were identified as those that reached 5% FDR significance (*fdr significant CpGs*; n = 2767). This analysis was performed using LOLA (Locus Overlap Analysis) software. Annotations include the antibody for the protein (Antibody), cell type for the ChIPseq experiment (CellType), Description, and name of the file from ENCODE database used for analysis (Filename). Results include the odds ratio (OR), p-value from Fisher's test (pValue), and false discovery rate (FDR).

Supplementary Table 19 Sensitivity analysis results for gene set enrichment analysis of significant methylation differences associated with the AD Braak stage identified in meta-analysis. The p-values for single CpGs were subjected to *bacon* correction first, then the significant CpGs were identified as those that reached 5% FDR significance after *bacon* correction. Shown are the top 20 most significant gene ontology terms (Gene Set), a description of the pathway (Description), and significance assessment (P-value, FDR).

Supplementary Table 20 Sensitivity analysis results for gene set enrichment analysis of significant methylation differences associated with the AD Braak stage identified in meta-analysis. The p-values for single CpGs were subjected to *bacon* correction first, then the significant CpGs were identified as those that reached genome-wide significance after *bacon* correction. Shown are the top 20 most significant gene ontology terms (Gene Set), a description of the pathway (Description), and significance assessment (P-value, FDR).

Supplementary Table 21 Single CpG analysis results using QN.BMIQ pre-processing pipeline. Shown are p-values for testing association between methylation levels and Braak stage, adjusting for age, sex, and neuron proportions for London cohort samples (London_pval) and Mount Sinai cohort samples (MtSinai_pval). These results are similar to those in Table 1 of Smith et al. (2018) (PMID: 29550519), analyzed using *dasen* analysis pipeline (PMID: 23631413).

Supplementary Table 22 Average gene expression levels for PRC2 target genes by AD stages in ROSMAP samples. The PRC2 target genes were identified in Supplementary Tables 3 and 6 of Schimmelmänn et al. (2016) (PMID: 27526204). Shown are the direction of changes in gene expression in mouse experiment upon PRC2 deletion (Regulation) and the average RNA expression levels in ROSMAP cohort for samples with different Braak stages (Control: 0-2, Stage 3, 4, 5, 6), as measured by normalized FPKM values.

Supplementary Table 23 Quality control (QC) information on DNA methylation samples and probes for each cohort contributing to this meta-analysis. Under *Probes QC*, shown are the number of probes

remaining after each QC procedure. Under *Samples QC*, shown are the number of samples remaining after each QC procedure.

Supplementary Table 24 Enrichment analysis results using mixed effects model (with random chromosomes effects to model correlations between CpGs) are similar to those from Fisher's exact test. The significant CpGs were identified as those reaching 5% FDR significance. A logistic mixed effects regression model was used to test the association between the type of genomic region (e.g. isCpGisland = "yes" or "no") and the significance of the CpG (e.g. isSignificant = "yes" or "no"). Random effects for each chromosome were also included in this model to account for correlations between CpGs within the same chromosome. For comparison, enrichment analysis results using Fisher's exact test from Supplementary Tables 3 and 4 are also included. Shown are odds ratios (OR) and p-values (p-value) obtained by each method. Shown in bold are significant enrichment results.

Supplementary Tables and Figures

Supplementary Figure 1 Quantile-quantile (QQ) plots of observed and expected distributions of p-values in Gasparoni, London, Mount Sinai, and ROSMAP cohorts. λ is the genomic inflation factor, and λ_{bacon} is the genomic inflation factor estimated using the method of Iterson et al. (2017) (PMID: 28129774), as implemented in the *bacon* R package. Shading indicates 95% confidence intervals.

Supplementary Figure 2 Enrichment of CpGs significantly associated with the AD Braak stage in meta-analysis of individual CpGs at 5% FDR, after inflation correction by *bacon* method. A two-sided Fisher's test was used to determine over- or under-representation of the significant CpGs in various (A) genomic features and (B) chromatin states. *** indicates P-value < 0.001, ** indicates P-value < 0.01 and * indicates P-value < 0.05.

Supplementary Figure 3 Enrichment of CpGs significantly associated with AD Braak stage in meta-analysis of individual CpGs reaching genome-wide significance (2.4×10^{-7}), after inflation correction by *bacon* method. A two-sided Fisher's test was used to determine over- or under-representation of the significant CpGs in various (A) genomic features and (B) chromatin states. *** indicates P-value < 0.001, ** indicates P-value < 0.01 and * indicates P-value < 0.05.

13. There are occasionally grammar issues throughout the manuscript. Please carefully check the whole manuscript.

In response to this reviewer's comment, we carefully checked the entire manuscript. In addition, we also worked with a staff at the writing center, who edited the entire manuscript for grammar issues.

References

1. Lunnon, K. et al. Methylomic profiling implicates cortical deregulation of ANK1 in Alzheimer's disease. *Nat Neurosci* **17**, 1164-1170 (2014).
2. Stock, A.J., Kasus-Jacobi, A. & Pereira, H.A. The role of neutrophil granule proteins in neuroinflammation and Alzheimer's disease. *J Neuroinflammation* **15**, 240 (2018).
3. De Jager, P.L. et al. Alzheimer's disease: early alterations in brain DNA methylation at ANK1, BIN1, RHBDF2 and other loci. *Nat Neurosci* **17**, 1156-1163 (2014).
4. Watson, C.T. et al. Genome-wide DNA methylation profiling in the superior temporal gyrus reveals epigenetic signatures associated with Alzheimer's disease. *Genome Med* **8**, 5 (2016).
5. Bakulski, K.M. et al. Genome-wide DNA methylation differences between late-onset Alzheimer's disease and cognitively normal controls in human frontal cortex. *J Alzheimers Dis* **29**, 571-588 (2012).
6. von Schimmelmann, M. et al. Polycomb repressive complex 2 (PRC2) silences genes responsible for neurodegeneration. *Nat Neurosci* **19**, 1321-1330 (2016).
7. Schlesinger, Y. et al. Polycomb-mediated methylation on Lys27 of histone H3 pre-marks genes for de novo methylation in cancer. *Nature genetics* **39**, 232-236 (2007).
8. Reddington, J.P. et al. Redistribution of H3K27me3 upon DNA hypomethylation results in de-repression of Polycomb target genes. *Genome Biol* **14**, R25 (2013).
9. Brinkman, A.B. et al. Sequential ChIP-bisulfite sequencing enables direct genome-scale investigation of chromatin and DNA methylation cross-talk. *Genome Res* **22**, 1128-1138 (2012).
10. Yu, L. et al. Methylation profiles in peripheral blood CD4+ lymphocytes versus brain: The relation to Alzheimer's disease pathology. *Alzheimer's & dementia : the journal of the Alzheimer's Association* **12**, 942-951 (2016).
11. Hannon, E., Lunnon, K., Schalkwyk, L. & Mill, J. Interindividual methylomic variation across blood, cortex, and cerebellum: implications for epigenetic studies of neurological and neuropsychiatric phenotypes. *Epigenetics* **10**, 1024-1032 (2015).
12. Joehanes, R. et al. Epigenetic Signatures of Cigarette Smoking. *Circ Cardiovasc Genet* **9**, 436-447 (2016).
13. Chen, Y.A. et al. Discovery of cross-reactive probes and polymorphic CpGs in the Illumina Infinium HumanMethylation450 microarray. *Epigenetics* **8**, 203-209 (2013).
14. McCarthy, S. et al. A reference panel of 64,976 haplotypes for genotype imputation. *Nature genetics* **48**, 1279-1283 (2016).
15. Hannon, E. et al. An integrated genetic-epigenetic analysis of schizophrenia: evidence for co-localization of genetic associations and differential DNA methylation. *Genome Biol* **17**, 176 (2016).

REVIEWERS' COMMENTS

Reviewer #1 (Remarks to the Author):

I thank the authors for their comprehensive revision. All my comments and suggestions were addressed.

I just have one final suggestion: could the authors please add more prominently somewhere (e.g. to the results 'meta-analysis' section or possibly to the abstract) that 339 CpG sites of the original 3.7k also passed a genome-wide threshold?

This information does not get mentioned until the 'sensitivity analysis' section and is hidden away in SM Table 15, but I believe the difference in results based on different multiple correction methods needs to be more visible.

This issue aside, I believe the current version is suitable for publication.

Reviewer #2 (Remarks to the Author):

I thank the authors for preparing such a detailed and well presented set of responses. I have no further comments.

Reviewer #3 (Remarks to the Author):

I appreciate the authors' efforts in revising the manuscript. The authors answered all of my comments.

REVIEWERS' COMMENTS

Reviewer #1 (Remarks to the Author):

I thank the authors for their comprehensive revision. All my comments and suggestions were addressed. I just have one final suggestion: could the authors please add more prominently somewhere (e.g. to the results 'meta analysis' section or possibly to the abstract) that 339 CpG sites of the original 3.7k also passed a genome-wide threshold?

This information does not get mentioned until the 'sensitivity analysis' section and is hidden away in SM Table 15, but I believe the difference in results based on different multiple correction methods needs to be more visible. This issue aside, I believe the current version is suitable for publication.

Response

We were glad that our comprehensive revision addressed the reviewer's comments well. Because the abstract is limited to only 150 words, we have now added information on the 339 CpG sites that passed the genome-wide significance threshold to the Results section in response to this reviewer's comment.

Revision

In Results, under "Meta-analysis identified methylation differences significantly associated with AD Braak stage at individual CpGs and co-methylated genomic regions"

Among the 3751 CpGs, 339 also reached genome-wide significance level (see details in Section "Genomic inflation and sensitivity analysis" below).